# MULTI-PLAY COMBINATORIAL SEMI-BANDIT PROBLEM

## ABSTRACT

In the combinatorial semi-bandit (CSB) problem, a player selects an action from a combinatorial action set and observes feedback from the base arms included in the action. While CSB is widely applicable to combinatorial optimization problems, its restriction to binary decision spaces excludes important cases involving non-negative integer flows or allocations, such as the optimal transport and knapsack problems. To overcome this limitation, we propose a general form of CSB, the multi-play combinatorial semi-bandit (MP-CSB) problem, where a player can select a non-negative integer action and observe multiple feedbacks from a single arm in each round. We propose two algorithms for the MP-CSB. One is a Thompson-sampling-based algorithm that is computationally feasible even when the action space is exponentially large with respect to the number of arms, and attains $O(\log T)$ distribution-dependent regret in the stochastic regime, where $T$ is the time horizon. We show that this algorithm can be much more memory-efficient than naively expanding existing algorithms. The other is a best-of-both-worlds algorithm, which achieves $O(\log T)$ variance-dependent regret in the stochastic regime and the worst-case $\tilde{\mathcal{O}}(\sqrt{T})$ regret in the adversarial regime. Moreover, its regret in adversarial one is data-dependent, adapting to the cumulative loss of the optimal action, the total quadratic variation, and the path-length of the loss sequence. Finally, we numerically show that the proposed algorithms outperform existing methods in the CSB literature.

## 1 INTRODUCTION

The multi-armed bandit (MAB) problem is a fundamental framework to investigate online decision-making problems, where we study the tradeoff between *exploitation* and *exploration* problem (Auer et al., 2002; Audibert & Bubeck, 2009). One of the most important subfields of MAB is the combinatorial bandit problem (Audibert et al., 2014; Cesa-Bianchi & Lugosi, 2012; Combes et al., 2015; Wang & Chen, 2018; Kveton et al., 2015; Chen et al., 2016b; Wang & Chen, 2017; Chen et al., 2016a; Wen et al., 2014), which has many practical applications such as the shortest path problem (Sniedovich, 2006), crowdsourcing (ul Hassan & Curry, 2016), matching (Gibbons, 1985), the spanning tree problem (Pettie & Ramachandran, 2002), recommender systems (Qin et al., 2014), and learning spectrum allocations (Gai et al., 2012). In the combinatorial semi-bandit (CSB) problem, a player sequentially interacts with an unknown environment over $T$ rounds. At each round, the player selects a combinatorial action to play and observes the losses for each selected component. The goal is to minimize (expected) cumulative regret, defined as the difference between the loss of the player's actions and the loss of the optimal action.

The CSB problem has been widely applied to modeling combinatorial optimization problems under uncertainty. However, its restriction to *binary* decision spaces limits its applicability to problems involving integer flows or allocations, such as the knapsack problem (Dantzig & Mazur, 2007), the optimal transport (OT) problem (Villani, 2008), and numerous others. These problems require a more general action space where decisions can take *non-negative integer values*. In these problems, each element of an action often represents some quantity. For example, in the OT problem, each element of an action can be seen as the number of trucks used to transport goods. In real-world applications, it is reasonable to assume that each truck is equipped with sensors and that feedback can be obtained from each truck.

Table 1: Regret upper bounds of algorithms for linear objectives. $\Delta_{\mathrm{LB,min}}$ and $\sigma_{\mathrm{LB}}$ are the minimum sub-optimality gap and maximum variance of the feedback loss, respectively. $c(\mathcal{A}, \ell)$ is a quantity that appears in the analysis of an asymptotic lower bound satisfying $\liminf_{T \to \infty} \frac{\mathbb{E}[R_T]}{\log T} \geq \Omega(c(\mathcal{A}, \ell))$. $b_i$ is the number of duplicates of arm $i$. Other quantities are introduced in Section 2.

| Reference | Stochastic | Adversarial | Complexity |
|---|---|---|---|
| Ito & Takemura (2023a) | $\mathcal{O}\left(\frac{d^3 \sigma_{\mathrm{LB}}^2 \log T}{\Delta_{\mathrm{LB,min}}}\right)$ | $\mathcal{O}\left(d^2 \sqrt{Z \log T}\right)$ | $\mathrm{Exp}(d)$ |
| Ito & Takemura (2023b) | $\mathcal{O}\left(\frac{d^2}{\Delta_{\mathrm{LB,min}}} \log T\right)$ | $\mathcal{O}(d\sqrt{T \log T})$ | $\mathrm{Exp}(d)$ |
| Lee et al. (2021) | $\mathcal{O}\left(c(\mathcal{A}, \ell)(\log T)^2\right)$ | $\mathcal{O}(\sqrt{dT} \log T)$ | $\mathrm{Exp}(d)$ |
| CTS (Wang & Chen, 2018) | $\mathcal{O}\left(\sum_{i=1}^d b_i \frac{\log M}{\Delta_i} \log T\right)$ | - | $\mathrm{Poly}\left(\sum_{i=1}^d b_i\right)$ |
| LBINFV (Tsuchiya et al., 2023) | $\mathcal{O}\left(\sum_{i \in J^*} b_i \frac{M\sigma_i^2}{\Delta_i} \log T\right)$ | $\mathcal{O}(\sqrt{\sum_{i=1}^d b_i Z \log T})$ | $\mathrm{Poly}\left(\sum_{i=1}^d b_i\right)$ |
| **GenCTS** | $\mathcal{O}\left(\sum_{i=1}^d \frac{\log m}{\Delta_i} \log T\right)$ | - | $\mathrm{Poly}(d)$ |
| **GenLBINFV** | $\mathcal{O}\left(\sum_{i \in J^*} \frac{M\sigma_i^2}{\Delta_i} \log T\right)$ | $\mathcal{O}(\sqrt{\sum_{i=1}^d n_i^2 Z \log T})$ | $\mathrm{Poly}(d)$ |

In this paper, we propose a novel framework, multi-play CSB (MP-CSB), where the set of actions is a subset of the non-integer vector space, and multiple losses can be observed from a single arm. MP-CSB can be seen as a natural generalization of the ordinary CSB problem.

**Related Works**    In Table 1, we show existing works that are related to MP-CSB.

The top three are state-of-the-art methods (SOTA) for the linear bandit (LB) (Ito & Takemura, 2023a;b; Lee et al., 2021). In the ordinary CSB, if the objective is linear, these algorithms can be applied. However, there are several reasons why LB algorithms are not recommended for (MP-)CSB. First, to apply LB algorithms, we need to enumerate all the feasible actions, which is unrealistic since the time complexity to enumerate them is exponential with respect to the number of arms in general. Secondly, since LB algorithms assume full-bandit feedback, in which the agent observes only the sum of rewards, they are not able to take advantage of the benefit of obtaining samples from every single arm.

The middle two are existing works that show SOTA methods proposed for the ordinary CSB. One may apply existing CSB algorithms to MP-CSB by *duplicating* base arms so that the action space becomes binary. However, this duplicating technique has two major shortcomings. First, duplicating base arms greatly increases the number of base arms, making it computationally infeasible to maintain statistics for each base arm (e.g., sample mean) in some cases. We show this in detail in Section 2. The second issue is the sample efficiency in the stochastic regime. Even with the duplicating technique, existing algorithms maintain separate statistics on each base arm (Wang & Chen, 2018; Neu, 2015; Chen et al., 2021; Tsuchiya et al., 2023), even though all of the duplicated base arms follow the same distribution. Such a lack of distinction between identical distributions leads to poor sample efficiency and may force the player to choose suboptimal actions frequently.

**Contributions**    We first formulate MP-CSB, a general form of the conventional CSB that does not restrict the action set to binary vectors. We show that it can be applied to combinatorial optimization problems whose action sets are collections of nonnegative integer vectors—such as the optimal transport problem and the knapsack problem—which have long been overlooked in the CSB literature.

Then, we introduce two algorithms. First, we introduce the Generalized Combinatorial Thompson Sampling (GenCTS) algorithm. Notably, this algorithm remains computationally feasible even when the action set grows exponentially with the number of arms. Furthermore, we establish that GenCTS can be significantly more memory-efficient than naively applying conventional CSB algo-

rithms to the MP-CSB setting with the duplicating technique. While the algorithm may be viewed as a straightforward extension of the CTS algorithm proposed by Wang & Chen (2018), our primary contribution lies in the accompanying regret analysis. In particular, we demonstrate that GenCTS achieves $\mathcal{O}(\log T)$ distribution-dependent regret in the stochastic regime and further improves upon the case of naively applying existing methods by a margin of $\mathcal{O}(\sum_{i=1}^d b_i/d)$, where $b_i$ represents the number of duplicates of arm $i$.

Then, we introduce a *best-of-both-worlds* (BOBW) algorithm named GenLBINFV (Generalized Logarithmic Barrier Implicit Normalized Forecaster considering Variances for semi-bandits), which achieves $\mathcal{O}(\log T)$ variance-dependent regret in the stochastic regime and $\tilde{\mathcal{O}}(\sqrt{T})$ regret in the adversarial regime. We extended the LBNINFV (Tsuchiya et al., 2023) algorithm for MP-CSB by effectively leveraging $n_i$, the maximum number of times arm $i$ can be pulled in one round. Moreover, its regret in adversarial one is data-dependent, adapting to the cumulative loss of the optimal action, the total quadratic variation, and the path-length of the loss sequence.

Finally, we show that our algorithms outperform existing methods in the ordinary CSB literature by conducting numerical experiments with synthetic data.

## 2 PRELIMINARIES

In this section, we formally define the MP-CSB problem and introduce the three regimes depending on how the loss is generated. Then, we discuss the optimality in MP-CSB, and show some typical applications of MP-CSB.

**Multi-Play Combinatorial Semi-Bandit (MP-CSB) Problem**  We formalize the MP-CSB problem as follows. Consider a set of $d$ base arms, indexed by $i \in \{1, \ldots, d\}$. In each round $t \in [T]$, the environment generates a set of losses $\{L_{i,j}(t)\}_{j=1,\ldots,n_i} \subset [0,1]$ for each base arm $i$, where $n_i$ is the maximum number of times arm $i$ can be sampled in a single round. Then, the player chooses an action $\boldsymbol{a}(t)$ from the action set $\mathcal{A} \subset \mathbb{Z}_{\geq 0}^d$, observes a set of losses, $\{L_{i,j}(t)\}_{j=1,\ldots,a_i(t)}$, for each $i$ where $a_i(t) \geq 1$, and incurs a loss of $f(\boldsymbol{a}(t), \boldsymbol{L}^1(t), \ldots, \boldsymbol{L}^d(t))$, where $f : \mathcal{A} \times \mathbb{R}^{n_1} \times \cdots \mathbb{R}^{n_d} \to \mathbb{R}$ and $\boldsymbol{L}^i = (L_{i,1}(t), \ldots, L_{i,n_i}(t)) \in \mathbb{R}^{n_i}$. The performance of the player is evaluated by the expected regret $R_T$ defined as the difference between the cumulative losses of the player and the single optimal action $\boldsymbol{a}^*$ fixed in terms of the expected cumulative loss, i.e., $\boldsymbol{a}^* = \arg\min_{\boldsymbol{a} \in \mathcal{A}} \mathbb{E}\left[\sum_{t=1}^T f(\boldsymbol{a}, \boldsymbol{L}^1(t), \ldots, \boldsymbol{L}^d(t))\right]$ and

$$R_T = \mathbb{E}\left[\sum_{t=1}^T (f(\boldsymbol{a}(t), \boldsymbol{L}^1(t), \ldots, \boldsymbol{L}^d(t)) - f(\boldsymbol{a}^*, \boldsymbol{L}^1(t), \ldots, \boldsymbol{L}^d(t)))\right],$$

where the expectation is taken w.r.t. to the randomness of losses and the internal randomness of the algorithm.

We define $I_{\boldsymbol{a}} = \{i \in [d] \mid a_i \geq 1\}$, representing the set of indices of arms from which one or more samples are obtained. We define $J^* = [d] \setminus I_{\boldsymbol{a}^*}$, $m = \max_{\boldsymbol{a} \in \mathcal{A}} |I_{\boldsymbol{a}}|$, and $M = \max_{\boldsymbol{a} \in \mathcal{A}} \|\boldsymbol{a}\|_1$. Note that $M \geq m$. Also, we define an action set dependent constant $\lambda_{\mathcal{A}} = \min\{M, W_{J^*}\}$, where $W_{J^*} = \sum_{i \in J^*} n_i$. We assume that for all $i \in [d]$, there exists $\boldsymbol{a} \in \mathcal{A}$ such that $a_i \geq 1$.

**Considered Regimes**  We consider three regimes as the assumptions for the losses.

In the *stochastic regime*, the losses are generated by unknown but fixed distributions. Before the game starts, the environment chooses an arbitrary distribution $\mathcal{D}_i$ for each base arm $i \in [d]$. In each round $t$, for each $i \in [d]$, the environment samples a set of $n_i$ random variables, $\{L_{i,j}(t)\}_{j=1,\ldots,n_i}$, from $\mathcal{D}_i$. We denote $\ell_i = \mathbb{E}_{\xi \sim \mathcal{D}_i}[\xi]$ and $\sigma_i^2 \in [0, 1/4]$ as the expected outcome and variance of base arm $i$, respectively. Also, we assume that the expected loss of an action $\boldsymbol{a} \in \mathcal{A}$ only depends on the mean outcomes of base arms in $I_{\boldsymbol{a}}$. That is, there exists a function $r$ such that $\mathbb{E}[f(\boldsymbol{a}, \boldsymbol{L}^1(t), \ldots, \boldsymbol{L}^d(t))] = r(\boldsymbol{a}, \{\ell_i\}_{i \in [I_{\boldsymbol{a}}]})$.

By contrast, in the *adversarial regime*, we do not assume any stochastic structure for the losses, and the losses can be chosen arbitrarily. In this regime, for each $i \in [d]$ and $j \in [n_i]$,

the environment can choose $L_{i,j}(t)$ depending on the past history until $(t-1)$-th round, i.e., $\{(\boldsymbol{L}^1(s), \ldots, \boldsymbol{L}^d(s), \boldsymbol{a}(s))\}_{s=1}^{t-1}$.

We also consider the *stochastic regime with adversarial corruptions* (Ito, 2021; Zimmert & Seldin, 2021), which is an intermediate regime between the stochastic and adversarial regimes. In this regime, for each $i \in [d]$, after a set of temporary loss $\{L'_{i,j}(t)\}_{j=1,\ldots,n_i}$ is sampled from $\mathcal{D}_i$, the adversary corrupts $\{L'_{i,j}(t)\}_{j=1,\ldots,n_i}$ to $\{L_{i,j}(t)\}_{j=1,\ldots,n_i}$. We define the corruption level by $C = \mathbb{E}\left[\sum_{t=1}^T \max_{i \in [d]} \max_{j \in [n_i]} |L_{i,j}(t) - L'_{i,j}(t)|\right] \geq 0$. If $C = 0$, this regime coincides with the stochastic regime.

**Discussion on the Optimality**  Next, we discuss lower bounds of MP-CSB for stochastic and adversarial regimes. For the stochastic regime, if the objective is linear, i.e., $r(\boldsymbol{a}, \{\ell_i\}_{i \in [I_{\boldsymbol{a}}]}) = \boldsymbol{a}^\top \boldsymbol{\ell}$, any consistent [1] algorithm with the duplicating technique suffers a regret of $\Omega(\frac{\sum_{i=1}^d b_i M}{\Delta} \log T)$ asymptotically, where $\Delta = \min_{\boldsymbol{a} \in \mathcal{A} \setminus \{\boldsymbol{a}^*\}} \boldsymbol{a}^\top \boldsymbol{\ell} - \boldsymbol{a}^{*\top} \boldsymbol{\ell}$ and $b_i$ is the number of duplicates of arm $i$ (Kveton et al., 2015; Merlis & Mannor, 2020). In Sections 3 and 4, we show that the upper bounds of our proposed algorithms are tighter than this lower bound. We leave the derivation of a regret lower bound of consistent algorithms without using the duplicating technique in MP-CSB as a future work.

On the other hand, in the adversarial regime, since the adversary is setting $\sum_{i=1}^d n_i$ losses in total, from Audibert et al. (2014), we can directly obtain a worst case lower bound of MP-CSB of $\sqrt{M(\sum_{i=1}^d n_i)T}$.

**Typical Applications of MP-CSB**  Here, we show two typical applications, the optimal transport and knapsack problems, where MP-CSB can be applied.

The optimal transport (OT) problem (Villani, 2008) models resource allocation from $N_{\text{sup}}$ suppliers to $N_{\text{dem}}$ demanders. It is defined on a complete bipartite graph, where each supplier $x \in S := \{1, \ldots, N_{\text{sup}}\}$ has $u_x \in \mathbb{Z}_{\geq 0}$ trucks to deliver items, and each demander $y \in D := \{1, \ldots, N_{\text{dem}}\}$ requires $v_y \in \mathbb{Z}_{\geq 0}$ units of items. The goal is to find the most efficient transportation plan, minimizing the total transportation cost:

$$
\begin{aligned}
\text{min.} \quad & \sum_{x=1}^{N_{\text{sup}}} \sum_{y=1}^{N_{\text{dem}}} a_{xy} c_{xy} \\
\text{s.t.} \quad & \boldsymbol{a} \in \{\boldsymbol{\pi} \in \mathbb{Z}_{\geq 0}^{N_{\text{sup}} \times N_{\text{dem}}} \mid \boldsymbol{\pi} \mathbf{1} = \boldsymbol{u}, \boldsymbol{\pi}^\top \mathbf{1} = \boldsymbol{v}\},
\end{aligned}
\tag{1}
$$

where $a_{xy}$ represents the number of trucks transported from supplier $x$ to demander $y$, and $c_{xy}$ is the transportation cost of edge $(x, y)$.

In some scenarios, the transportation cost $c_{xy}$ is unknown and must be estimated. As a real-world application, each truck may have a sensor to measure the cost of edges it passes through, and feedback can be obtained from each truck. In such a case, we can apply the MP-CSB problem. Here, the number of arms $d$ is the number of edges in the bipartite graph, i.e., $d = N_{\text{sup}} \times N_{\text{dem}}$. Also, $n_{xy}$ is the maximum number of trucks that can pass through edge $(x, y)$, i.e., $n_{xy} = \min\{u_x, v_y\}$.

Next, we introduce an example of the knapsack problem (Dantzig & Mazur, 2007). In the knapsack problem, we have $d$ items. Each item $i \in [d]$ has a weight $w_i$ and value $\mu_i$. Also, there is a knapsack whose capacity is $W$ in which we put items. Our goal is to maximize the total value of the items in the knapsack, not letting the total weight of the items exceed the capacity of the knapsack. Formally, the optimization problem is given as follows:

$$
\begin{aligned}
\text{maximize}_{\boldsymbol{a}} \quad & \boldsymbol{a}^\top \boldsymbol{\mu} \\
\text{subject to} \quad & \boldsymbol{a}^\top \boldsymbol{w} \leq W \\
\text{and} \quad & \boldsymbol{a} \in \mathbb{Z}_{\geq 0}^d,
\end{aligned}
$$

where $\boldsymbol{w} = (w_1, \ldots, w_d)$.

Then, let us consider online advertising. Suppose an advertiser considers placing different types of ads in a frame of size $W$ on a website. The advertiser is allowed to place multiple ads of the same

---

[1] We say that an algorithm is consistent if for any stochastic CSB instance problem instance, any suboptimal $\boldsymbol{a}$, and any $0 < \alpha < 1$, $\mathbb{E}[T_n(\boldsymbol{a})] = o(n^\alpha)$, where $T_n(\boldsymbol{a})$ is the number of times that action $\boldsymbol{a}$ is chosen in $n$ steps by the algorithm.

type. The size of each ad $i$ is $w_i$, and it is assumed that the total size of all ads must not exceed $W$. As feedback to advertisers, they observe the profits generated from each ad. In this example, we can apply the MP-CSB problem. The number of arms is the number of types of ads, and $n_i$ is the maximum number of ad $i$ that can be put on a website, i.e., $n_i = \left\lfloor \frac{W}{w_i} \right\rfloor$.

## 3 GENERALIZED CTS ALGORITHM

In this section, we introduce the generalized combinatorial Thompson sampling (GenCTS) algorithm, which is computationally feasible even when the action set $\mathcal{A}$ is exponentially large in $d$. We show that the GenCTS algorithm achieves $O(\log T)$ distribution-dependent regret in the stochastic regime. Although GenCTS can be viewed as a straightforward extension of ordinary CTS (Wang & Chen, 2018), the primary contribution is to quantitatively analyze how much tighter a regret upper bound GenCTS can attain compared with CTS with the duplication technique. We also show that GenCTS can be substantially more efficient than CTS in both memory and computation.

**Technical Assumptions**  To allow the GenCTS to handle not only linear loss functions but also a broader class of nonlinear loss functions, we assume that the function $r$ is Lipschitz continuous. Specifically, there exists a constant $\kappa_r$, such that for every action $\boldsymbol{a}$ and every pair of mean vectors $\boldsymbol{\mu}$ and $\boldsymbol{\mu}'$, $\left| r(\boldsymbol{a}, \{\mu_i\}_{i \in I_{\boldsymbol{a}}}) - r(\boldsymbol{a}, \{\mu_i'\}_{i \in I_{\boldsymbol{a}}}) \right| \le \kappa_r \sum_{i \in I_{\boldsymbol{a}}} |\mu_i - \mu_i'|$. In the OT and knapsack problems, $r(\boldsymbol{a}, \{\mu_i\}_{i \in [I_{\boldsymbol{a}}]}) = \sum_{i \in I_{\boldsymbol{a}}} a_i \mu_i$, and we can easily confirm that $\kappa_r = \max_{i \in [d]} n_i$.

Also, GenCTS assumes that we have an *oracle* that takes a vector $\boldsymbol{\rho} = (\rho_1, \ldots, \rho_d)$ as input and output an action $\text{Oracle}(\boldsymbol{\rho}) = \arg\min_{\boldsymbol{a} \in \mathcal{A}} r(\boldsymbol{a}, \{\rho_i\}_{i \in I_{\boldsymbol{a}}})$. We assume that the time complexity of the oracle is polynomial or pseudo-polynomial [2] in $d$. For instance, since it is known that the OT problem can be solved in $O(d^3 \log d)$ time (Cuturi, 2013) using a linear programming solver, it can be the oracle. For the knapsack problem, there is a dynamic programming-based algorithm that runs in pseudo-polynomial time (Kellerer et al., 2004; Fujimoto, 2016), and therefore it can be the oracle.

**Algorithm**  GenCTS is shown in Algorithms 1 and 2. Initially, we set a prior distribution of all the base arms as the beta distribution $\text{Beta}(1, 1)$, which is the uniform distribution on $[0, 1]$. In each round $t$, we choose an action by drawing independent samples, $\{\theta_i(t)\}_{i \in [d]}$, from each base arm's prior distribution, and use the output from the oracle, $\boldsymbol{a}(t) = \text{Oracle}(\boldsymbol{\theta}(t))$, as the action to play. After we obtain losses, we update the prior distributions of each base arm $i$ using the procedure Update (Algorithm 2). In the Update procedure, we update the prior beta distribution of each base arm as follows. For each $i \in I_{\boldsymbol{a}(t)}$ and $j \in [a_i(t)]$, we generate a Bernoulli random variable $Y_{i,j}(t)$ with mean $L_{i,j}(t)$, and update the prior beta distribution of base arm $i$ using $Y_{i,j}(t)$ as the new observation. Let $p_i(t)$ and $q_i(t)$ denote the values of $p_i$ and $q_i$ at the beginning of round $t$, respectively. Here $p_i(t) - 1$ and $q_i(t) - 1$ represent the number of 1s and 0s in $\bigcup_{s=1}^{t-1} \bigcup_{j=1}^{a_i(s)} \{Y_{i,j}(s)\}$, respectively. Then, following Bayes' rule, the posterior distribution of arm $i$ after round $t$ is $\text{Beta}(p_i(t) + \sum_{j=1}^{a_i(t)} Y_{i,j}(t), q_i(t) + \sum_{j=1}^{a_i(t)} (1 - Y_{i,j}(t)))$, which is what the Update procedure does for $p_i$ and $q_i$.

One key advantage of the GenCTS algorithm is that it does not require enumerating all the possible actions in $\mathcal{A}$ in the beginning of the game. The total computation time of GenCTS is $O(\text{poly}(d)T)$ or $O(\text{pseudo-poly}(d)T)$.

**Regret Analysis**  Here, we show a regret upper bound of the GenCTS algorithm.

**Theorem 3.1.** *The GenCTS algorithm achieves* $R_T = \mathcal{O}(\sum_{i=1}^{d} \frac{\kappa_r^2 \log m}{\Delta_i} \log T)$, *where* $\Delta_i = \min_{\boldsymbol{a} \in \mathcal{A} \setminus \{\boldsymbol{a}^*\} : a_i \ge 1} r(\boldsymbol{a}, \{\ell_i\}_{i \in I_{\boldsymbol{a}}}) - r(\boldsymbol{a}^*, \{\ell_i\}_{i \in I_{\boldsymbol{a}^*}})$

We can see that the upper bound of GenCTS is tighter than the asymptotic lower bound of consistent algorithms with the duplicating technique, $\mathcal{O}(\frac{\sum_{i=1}^{d} b_i M}{\Delta} \log T)$, since $b_i \ge 1$ and $M \ge m$. From the

---

[2]In computational complexity theory, a numeric algorithm runs in pseudo-polynomial time if its running time is a polynomial in the *numeric value* of the input (the largest integer present in the input)—but not necessarily in the length (dimension) of the input, which is the case for polynomial time algorithms.

**Algorithm 1: GenCTS: Generalized Combinatorial Thompson Sampling**

1: $\boldsymbol{p}(1), \boldsymbol{q}(1) \leftarrow \mathbf{1}_d, \mathbf{1}_d$
2: **for** $t = 1, 2, \ldots$ **do**
3:     $\text{Beta}(p_i(t), q_i(t)) \leftarrow \frac{x^{p_i(t)-1}(1-x)^{q_i(t)-1}}{\int_0^1 u^{p_i(t)-1}(1-u)^{q_i(t)-1} du}$
4:     For all arm $i \in [d]$, draw a sample $\theta_i(t)$ from $\text{Beta}(p_i(t), q_i(t))$
5:     $\boldsymbol{\theta}(t) \leftarrow (\theta_1(t), \ldots, \theta_d(t))$
6:     /\* Play an Action /\*
7:     Play action $\boldsymbol{a}(t) = \text{Oracle}(\boldsymbol{\theta}(t))$
8:     /\* Collect Losses /\*
9:     $Q(t) = \{\}$
10:     **for** $i \in I_{\boldsymbol{a}(t)}$ **do**
11:       **for** $j = 1, \ldots, a_i(t)$ **do**
12:        Observe loss $L_{i,j}(t)$
13:        $Q(t) \leftarrow (i, j, L_{i,j}(t))$
14:       **end for**
15:     **end for**
16:     /\* Update the beta distribution /\*
17:     $\boldsymbol{p}(t+1), \boldsymbol{q}(t+1) \leftarrow \text{Update}(\boldsymbol{p}(t), \boldsymbol{q}(t), Q(t))$
18: **end for**

**Algorithm 2: Procedure Update**

1: **Input**: $\boldsymbol{p}(t), \boldsymbol{q}(t), Q(t)$
2: **Output**: Updated $\boldsymbol{p}(t+1)$ and $\boldsymbol{q}(t+1)$
3: **for** $(i, j, L_{i,j}(t)) \in Q(t)$ **do**
4:     $Y_{i,j}(t) \leftarrow 1$ with probability $L_{i,j}(t)$, 0 with probability $1 - L_{i,j}(t)$
5:     $p_i(t+1) \leftarrow p_i(t) + Y_{i,j}(t)$
6:     $q_i(t+1) \leftarrow q_i(t) + 1 - Y_{i,j}(t)$
7: **end for**
8: **Return** $\boldsymbol{p}(t+1)$ and $\boldsymbol{q}(t+1)$

**Algorithm 3: Generalized LBINFV**

**Input:** Action set $\mathcal{A}$, time horizon $T$
1: **for** $t = 1, 2, \ldots T$ **do**
2:     Compute $\boldsymbol{x}(t) \in \mathcal{X}$ by (2)
3:     Sample $\boldsymbol{a}(t)$ such that $\mathbb{E}[\boldsymbol{a}(t)|\boldsymbol{x}(t)] = \boldsymbol{x}(t)$
4:     Take action $\boldsymbol{a}(t)$ and observe feedback $\{L_{i,1}, \ldots, L_{i,a_i(t)}\}$ for $i$ such that $a_i(t) \geq 1$.
5:     Update the regularization parameters $\beta_i(t)$ in (6) and optimistic prediction $q_i(t)$ using (3) or (4).
6: **end for**

result in Wang & Chen (2018), if we use the ordinary CTS algorithm with the duplicating technique for MP-CSB, the regret upper bound is $\mathcal{O}(\sum_{i=1}^d b_i \frac{\kappa_r^2 \log M}{\Delta_i} \log T)$. Therefore, we can see that the upper bound of GenCTS is tighter than that of the ordinary CTS algorithm with the duplicating technique.

**Time and Memory Complexity**   Here we briefly show that GenCTS enjoys clear advantages over ordinary CTS with the duplication technique, in both time and memory. The per-round time complexity of GenCTS, $\mathcal{O}(\text{Poly}(d))$, is no greater than that of CTS with duplication, $\mathcal{O}(\text{Poly}(\sum_{i=1}^d b_i))$. For example, in OT, the per-round time complexity of ordinary CTS is $\mathcal{O}((\sum_{i=1}^d b_i)^3 \log(\sum_{i=1}^d b_i))$, which can be much larger than that of GenLBINFV, $\mathcal{O}(d^3 \log d)$.

On the memory side, GenCTS maintains only $\mathcal{O}(\text{Poly}(d))$-sized statistics (e.g., one posterior for each arm), whereas CTS with the duplicating technique must keep a separate set for every duplicated arm—$(\sum_{x=1}^{N_{\text{sup}}} u_x) \times N_{\text{dem}}$ in total—leading to a space requirement on the order of $\mathcal{O}(\text{Poly}(\sum_{i=1}^d b_i))$. When $(\sum_{x=1}^{N_{\text{sup}}} u_x) \times N_{\text{dem}} \sim \mathcal{O}(10^{10})$, storing and updating such a vast number of base-arm posteriors becomes computationally and memory-wise impractical. See Appendix A for details.

## 4   GENERALIZED LBINFV ALGORITHM

In this section, we introduce the GenLBINFV algorithm (Algorithm 3), which is a BOBW algorithm for MP-CSB. GenLBINFV is inspired by the LBINFV algorithm; however, extending the original method to MP-CSB is not straightforward. We show that, by carefully leveraging $n_i$'s, LBINFV can be generalized to MP-CSB.

**Technical Assumption for GenLBINFV**   For the GenLBINFV algorithm, we need an assumption that the loss function is linear, i.e., $f(\boldsymbol{a}, \boldsymbol{L}^1(t), \ldots, \boldsymbol{L}^d(t)) = \sum_{i=1}^d \sum_{j=1}^{a_i} L_{i,j}(t)$. This assumption holds for many combinatorial optimization problems with a linear objective, such as the OT and knapsack problems.

**Algorithm** We construct the algorithm based on the *optimistic-follow-the-regularized-leader* (OFTRL) framework, which has occasionally been used in the development of the BOBW algorithms (Wei & Luo, 2018; Ito, 2021). In each round $t$, we choose $\boldsymbol{a}(t) \in \mathcal{A}$ so that $\mathbb{E}\left[\boldsymbol{a}(t) \mid \boldsymbol{x}(t)\right] = \boldsymbol{x}(t)$, where

$$\boldsymbol{x}(t) \in \min_{\boldsymbol{x} \in \mathcal{X}} \left\{ \left\langle \boldsymbol{q}(t) + \sum_{s=1}^{t-1} \hat{\boldsymbol{\ell}}(s), \boldsymbol{x} \right\rangle + \psi_t(\boldsymbol{x}) \right\}. \tag{2}$$

Here, $\mathcal{X} = \mathrm{conv}(\mathcal{A})$ is a convex hull of the action set $\mathcal{A}$. Below, we define $\boldsymbol{q}(t)$, $\hat{\boldsymbol{\ell}}(t)$, and $\psi_t(\boldsymbol{x})$.

$q(t)$ is called the optimistic prediction; intuitively, it estimates the loss of the arms in round $t$. For the choice of optimistic prediction $\boldsymbol{q}(t)$, we introduce two methods: the least squares (LS) and gradient descent (GD) methods. LS defines $\boldsymbol{q}(t) = (q_1(t), \ldots, q_d(t))^\top \in [0,1]^d$ by

$$q_i(t) = \frac{1}{N_i(t-1)} \left( \frac{1}{2} + \sum_{s=1}^{t-1} \sum_{j=1}^{a_i(s)} L_{i,j}(s) \right), \tag{3}$$

and GD defines $\boldsymbol{q}(t)$ by $q_i(1) = \frac{1}{2}$ and

$$q_i(t+1) = \begin{cases} (1-\eta)q_i(t) + \eta \frac{1}{a_i(t)} \sum_{j=1}^{a_i(t)} L_{i,j}(t) & \text{if } a_i(t) \geq 1, \\ q_i(t) & \text{otherwise}, \end{cases} \tag{4}$$

for all $i \in [d]$ with a step size $\eta \in (0, \frac{1}{2})$. The design of LS is to reduce the leading constant $\frac{1}{1-2\eta}$ in the regret, and GD is to derive a path-length bound.

Next, we define $\hat{\boldsymbol{\ell}}(t) = (\hat{\ell}_1(t), \ldots, \hat{\ell}_d(t)) \in \mathbb{R}^d$ as $\hat{\ell}_i(t) = q_i(t) + \frac{a_i(t)}{x_i(t)}(k_i(t) - q_i(t))$ for $i \in [d]$, where $\boldsymbol{k}(t) = (\frac{1}{a_1(t)} \sum_{j=1}^{a_1(t)} L_{1,j}(t), \ldots, \frac{1}{a_d(t)} \sum_{j=1}^{a_d(t)} L_{d,j}(t))$. From basic calculation, we can confirm that $\hat{\ell}_i(t)$ is an unbiased estimator of $\mathbb{E}\left[\sum_{j=1}^{a_i(t)} L_{i,j}(t) \mid \boldsymbol{x}(t)\right]/x_i$, which can be seen as the average of the losses occurred by pulling arm $i$. The optimistic prediction $\boldsymbol{q}(t)$ plays a role in reducing the variance of $\hat{\ell}(t)$; the better $\boldsymbol{q}(t)$ predicts $\boldsymbol{k}(t)$, the smaller the variance of $\hat{\ell}(t)$ becomes.

$\psi_t : \mathbb{R}^d \to \mathbb{R}$ is a convex regularizer function given by $\psi_t(\boldsymbol{x}) = \sum_{i=1}^d \beta_i(t)\varphi_i(x_i)$, where $\varphi_i : \mathbb{R} \to \mathbb{R}$ is defined as

$$\varphi_i(z) = n_i \left( \frac{z}{n_i} - 1 - \log \frac{z}{n_i} + \log T \left( \frac{z}{n_i} + \left( 1 - \frac{z}{n_i} \log \left( 1 - \frac{z}{n_i} \right) \right) \right) \right), \tag{5}$$

and regularization parameters $\{\beta_i(t)\}_{i=1,\ldots,d}$ are defined as

$$\beta_i(t) = \sqrt{\left( 1 + \frac{\epsilon_i}{n_i} \right)^2 + \frac{1}{\log T} \sum_{s=1}^{t-1} \left( \frac{a_i(s)}{n_i} \right)^2 (k_i(s) - q_i(s))^2 \min \left\{ 1, \frac{2(1 - \frac{x_i(s)}{n_i})}{(\frac{x_i(s)}{n_i})^2 \log T} \right\}}. \tag{6}$$

Here, $\epsilon_i \in (0, \frac{n_i}{2}]$ is a hyperparameter. Our regularizer function $\varphi_i$ consists of a logarithmic barrier term $-\log \frac{z}{n_i}$ and an entropy term $(1 - \frac{z}{n_i}) \log \left( 1 - \frac{z}{n_i} \right)$. This type of regularizer is called a *hybrid* regularizer and was employed in existing studies for bounding a component of the regret (Zimmert et al., 2019; Ito et al., 2022a;b). Our regularizer function can be seen as a generalization of that of LBINFV (Tsuchiya et al., 2023) since $n_i = 1$ for all $i \in [d]$ in the ordinary CSB. $\beta_i(t)$ determines the strength of the regularization. When $a_i(s) \geq 1$, $(k_i(s) - q_i(s))^2$ in (6) can be seen as the squared error of the optimistic prediction, and the algorithm becomes more explorative when the loss is unpredictable or has a high variance.

Overall, intuitively, $\boldsymbol{q}(t)$ and $\sum_{s=1}^{t-1} \hat{\boldsymbol{\ell}}(t)$ are values determined based on past information and are responsible for the *exploitation*. On the other hand, the regularizer $\psi_t(\boldsymbol{x})$ prevents overfitting to past data and encourages moderate *exploration*. This is intended to minimize regret even in adversarial environments.

**Computational Complexity** OFTRL in equation 2 can be solved in polynomial time in $d$ as long as the convex hull $\mathcal{X} = \text{conv}(\mathcal{A})$ is represented by a polynomial number of constraints or admits a polynomial-time separation oracle (Schrijver, 1998). Given the solution $\boldsymbol{x}(t) \in \mathcal{X}$, sampling a combinatorial action $\boldsymbol{a}(t) \in \mathcal{A}$ such that $\mathbb{E}[\boldsymbol{a}(t) \mid \boldsymbol{x}(t)] = \boldsymbol{x}(t)$ requires a convex decomposition of $\boldsymbol{x}(t)$ (Wei & Luo, 2018; Ito, 2021). By Carathéodory's theorem, any point $\boldsymbol{x}(t) \in \mathcal{X}$ can be expressed as a convex combination $\boldsymbol{x}(t) = \sum_{k=1}^{m} \lambda_k \boldsymbol{a}^{(k)}$, where $\boldsymbol{a}^{(k)} \in \mathcal{A}$, $\lambda_k \in [0, 1]$, $\sum_{k=1}^{m} \lambda_k = 1$, and $m \leq d+1$. When the linear optimization oracle over $\mathcal{A}$ is efficient, the Frank-Wolfe (FW) algorithm can construct such a decomposition iteratively (Combettes & Pokutta, 2021). This holds, for instance, in OT problems, where the feasible set forms a transportation polytope and each FW step reduces to a tractable linear program. In contrast, for NP-hard domains such as knapsack problems, solving OFTRL exactly is generally intractable. In practice, this can be addressed either by exploiting problem structure to keep the action set small enough to enumerate, or by using approximate optimization and sampling methods.

**Regret Analysis** Here, we show regret upper bounds of the GenLBINFV algorithm. First, we show regret upper bounds of the GenLBINFV algorithm for each optimistic prediction method under the stochastic regime.

**Theorem 4.1.** *Regret upper bounds of GenLBINFV using LS and GD methods in the stochastic regime are $\mathcal{O}(\sum_{i \in J^*} \frac{\lambda_{\mathcal{A}} \sigma_i^2}{\Delta_i} \log T)$ and $\mathcal{O}(\frac{1}{1-2\eta} \sum_{i \in J^*} \frac{\lambda_{\mathcal{A}} \sigma_i^2}{\Delta_i} \log T)$, respectively, where $\Delta_i = \min_{\boldsymbol{a} \in \mathcal{A}: a_i \geq 1} \boldsymbol{a}^\top \boldsymbol{\ell} - \boldsymbol{a}^{*\top} \boldsymbol{\ell}$.*

We can see that both optimistic prediction methods achieve $\mathcal{O}(\log T)$ variance-dependent regret bound. Variance dependency is a clear advantage since the variances of losses for each base arm are extremely small in many real-world applications (Tsuchiya et al., 2023; Komiyama et al., 2017; György et al., 2006). The upper bound of the ordinary LBINFV algorithm with the duplicating technique is $\mathcal{O}(\sum_{i \in J^*} b_i \frac{\lambda'_{\mathcal{A}} \sigma_i^2}{\Delta_i} \log T)$, where $\lambda'_{\mathcal{A}} = \min\{M, \sum_{i=1}^{d} b_i - \|\boldsymbol{a}^*\|_1\}$. In the example of OT, when $u_x$'s are large, $\sum_{i=1}^{d} b_i - \|\boldsymbol{a}^*\|_1 = (\sum_{x=1}^{N_{\text{sup}}} u_x) \times N_{\text{dem}} - \|\boldsymbol{a}^*\|_1$ is much larger than $M$, and we have $\lambda'_{\mathcal{A}} = M$. Therefore, $\lambda_{\mathcal{A}} = \min\{M, W_{J^*}\} \leq M = \lambda'_{\mathcal{A}}$, which implies that the upper bound of GenLBINFV is no looser than that of the LBINFV since $b_i \geq 1$.

Next, we show regret upper bounds of the GenLBINFV algorithm for each optimistic prediction method under the adversarial regime. Let us denote the cumulative loss of the optimal action, total quadratic variation in loss sequence, and path-length of loss sequence by $L^* = \min_{\boldsymbol{a} \in \mathcal{A}} \mathbb{E}[\sum_{t=1}^{T} \sum_{i=1}^{d} \sum_{j=1}^{a_i(t)} L_{i,j}(t)]$, $Q_2 = \mathbb{E}[\sum_{t=1}^{T} \|\boldsymbol{k}(t) - \frac{1}{T} \sum_{s=1}^{T} \boldsymbol{k}(s)\|_2^2]$, and $V_1 = \mathbb{E}[\sum_{t=1}^{T-1} \|\boldsymbol{k}(t) - \boldsymbol{k}(t+1)\|_1]$, respectively, to introduce the data-dependent bound of the GenLBINFV algorithm.

**Theorem 4.2.** *Regret upper bounds of GenLBINFV using the LS and GD methods in the adversarial regime are $\mathcal{O}(\sqrt{(\sum_{i=1}^{d} n_i^2) Z^{\text{LS}} \log T})$ and $\mathcal{O}(\sqrt{\frac{1}{1-2\eta} (\sum_{i=1}^{d} n_i^2) Z^{\text{GD}} \log T})$, respectively, where $Z^{\text{LS}} = \min\{L^*, MT - L^*, Q_2\}$ and $Z^{\text{GD}} = \min\{L^*, MT - L^*, Q_2, \frac{V_1}{\eta}\}$.*

$Z^{\text{LS}}$ and $Z^{\text{GS}}$ can be seen as indicators of the problem. If the problem is relatively easy and can be assumed to be $\mathcal{O}(Z^{\text{LS}}) = \mathcal{O}(Z^{\text{GD}}) = o(T)$, the GenLBINFV can achieve a much smaller bound than the the worst case bound $\mathcal{O}(\sqrt{M(\sum_{i=1}^{d} b_i)T})$. The upper bound of the ordinary LBINFV using LS and GD methods are $\mathcal{O}(\sqrt{(\sum_{i=1}^{d} b_i) Z^{\text{LS}} \log T})$ and $\mathcal{O}(\sqrt{\frac{1}{1-2\eta} (\sum_{i=1}^{d} b_i) Z^{\text{GD}} \log T})$, respectively. In general, we do not know whether $\sum_{i=1}^{d} n_i^2$ is smaller than $\sum_{i=1}^{d} b_i$ or not.

On the other hand, GenLBINFV is computationally friendlier than LBINFV with the duplicating technique, since when calling the oracle to compute (2), the time complexity of GenLBINFV in each round is $\mathcal{O}(\text{Poly}(d))$, which can be much smaller than the time complexity of LBINFV with the duplicating technique, $\mathcal{O}(\text{Poly}(\sum_{i=1}^{d} b_i))$.

**Regret Upper Bound of the Intermediate Regime** We have the following corollary for the intermediate regime.

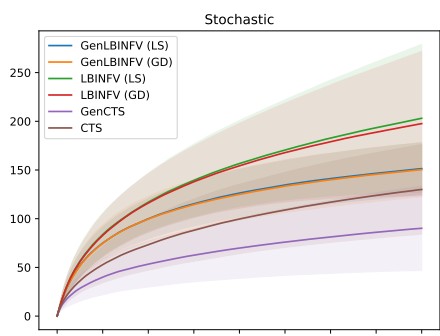 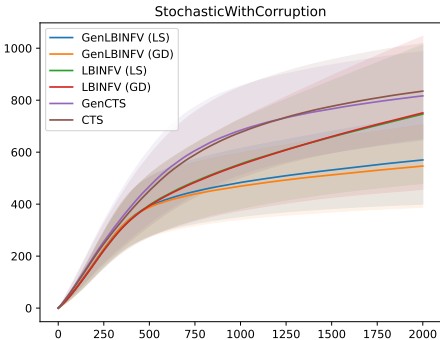

Figure 1: The result of the experiment under the stochastic regime.

Figure 2: The result of the experiment under the stochastic regime with adversarial corruption.

**Corollary 4.3.** *In the stochastic regime with adversarial corruptions, upper bounds of GenLBINFV using the LS and GD methods are $\mathcal{O}(R^{\mathrm{LS}} + \sqrt{CMR^{\mathrm{LS}}})$ and $\mathcal{O}(R^{\mathrm{GD}} + \sqrt{CMR^{\mathrm{GD}}})$, respectively. Here, $R^{\mathrm{LS}} = \mathcal{O}(\sum_{i \in J^*} \frac{\lambda_{\mathcal{A}} \sigma_i^2}{\Delta_i} \log T)$ and $R^{\mathrm{GD}} = \mathcal{O}(\frac{1}{1-2\eta} \sum_{i \in J^*} \frac{\lambda_{\mathcal{A}} \sigma_i^2}{\Delta_i} \log T)$.*

## 5 EXPERIMENTS

In this section, we compare the GenLBINFV and GenCTS algorithms with existing algorithms in the CSB literature with the duplicating technique, and numerically illustrate their behavior with synthetic data.

We use the same notation as that used in (1). We consider a case where $\boldsymbol{u} = (1,2,3,4)^{\top}$ and $\boldsymbol{v} = (1,3,6)^{\top}$. In each round $t$, the environment sets a loss $\{c_{xy,j}(t)\}_{j=1,\ldots,n}$ for each edge $(x,y)$. Then, the player's objective is to minimize the regret defined as follows: $R_T = \mathbb{E}\left[\sum_{t=1}^{T} \sum_{x=1}^{m} \sum_{y=1}^{n} (\sum_{j=1}^{a_{xy}(t)} c_{xy,j}(t) - \sum_{j=1}^{a_{xy}^*} c_{xy,j}(t))\right]$, where $\boldsymbol{a}^* = \arg\min_{\boldsymbol{a} \in \mathcal{A}} \mathbb{E}\left[\sum_{t=1}^{T} \sum_{x=1}^{m} \sum_{y=1}^{n} \sum_{j=1}^{a_{xy}} c_{xy,j}(t)\right]$.

We generate each element of the cost matrix $\boldsymbol{c}$ uniformly from $[0.10, 0.50]$. The time horizon $T$ is set to 2000. For the stochastic regime, each sample from edge $(x,y)$ is from $U(0, 2c_{xy})$, where $U(a,b)$ denotes the uniform distribution on $[a,b]$. For the stochastic regime with adversarial corruption, until $t \leq 200$, each sample from edge $(x,y)$ is drawn from $U(0, 2c_{xy})$, but when $t > 200$, it is drawn from $U(1 - 2c_{xy}, 1)$. We compare our algorithm with the LBINFV and CTS algorithms. To apply these two methods, we use the *duplicating* technique.

We show the results in Figures 1 and 2. The lines indicate the average over 100 independent trials. In the stochastic regime, we can see that GenCTS and GenLBINFV algorithms outperform the CTS and LBINFV algorithms, respectively. In the stochastic regime with adversarial corruptions, GenLBINFV outperforms Thompson sampling-based algorithms and converges faster than the LBINFV algorithm.

## 6 CONCLUSION

In this study, we proposed the MP-CSB framework, where a player can select a non-negative integer action and observe multiple feedbacks from a single arm in each round. We proposed two algorithms for MP-CSB: GenCTS and GenLBINFV. GenCTS is computationally feasible even when the action set $\mathcal{A}$ is exponentially large in $d$, and achieves a $\mathcal{O}(\log T)$ distribution-dependent regret. GenLBINFV is a BOBW algorithm, which achieves $\mathcal{O}(\log T)$ variance-dependent regret in the stochastic regime and $\tilde{\mathcal{O}}(\sqrt{T})$ regret in the adversarial regime. We numerically showed that the proposed algorithms outperform existing methods in the CSB literature.

## REPRODUCIBILITY STATEMENT

To ensure reproducibility, Section 2 provides a precise formulation; Sections 3 and 4 present the theoretical analyses of each algorithm; and Section 5 describes the experimental setup in detail. The proofs for the theoretical analyses are included in the Appendix, and the experimental code is submitted as supplementary materials.

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

## A    EXAMPLE OF THE DUPLICATING TECHNIQUE

In Figure 3, we show an example of the OT problem. Here, $S = \{1, 2, 3\}$ and $D = \{1, 2, 3, 4\}$. We have $d = n_s \times n_d = 12$ base arms. One candidate of action $\boldsymbol{a}$ can be $\boldsymbol{a} = \begin{pmatrix} 1 & 0 & 0 & 0 \\ 0 & 1 & 2 & 0 \\ 0 & 0 & 0 & 4 \end{pmatrix}$. The player observes a set of losses, $\{L_{(1,1),1}\} \cup \{L_{(2,2),1}\} \cup \{L_{(2,3),1}, L_{(2,3),2}\} \cup \{L_{(3,4),1}, L_{(3,4),2}, L_{(3,4),3}, L_{(3,4),4}\}$. Here, $L_{(x,y),j}$ is the loss of edge $(x, y)$ observed by the $j$-th truck departed from supplier $x$ to demander $y$. Then, she incurs a loss of

$$f(\boldsymbol{a}(t), \boldsymbol{L}^{(1,1)}, \ldots, \boldsymbol{L}^{(3,4)}) = \sum_{x=1}^{n_s} \sum_{y=1}^{n_d} \sum_{j=1}^{a_{xy}} L_{(x,y),j}.$$

To apply existing algorithms in the ordinary CSB algorithms to MP-CSB, one may use the duplicating technique, which is shown in Figure 4. Here, we treat each truck independently so that the action set becomes binary. However, the duplicating technique makes the sample efficiency in the stochastic regime worse. For instance, in the stochastic regime, orange, green, and purple arms (edges) in Figure 4 follow the same distribution as edges $(1, 2)$, $(2, 2)$, and $(3, 2)$, in Figure 3, respectively. Such a lack of distinction between identical distributions leads to poor sample efficiency and may force the player to choose suboptimal actions frequently.

## B    PROOF OF THEOREM 3.1

Here, we prove Theorem 3.1.

### B.1    CHERNOFF-HOEFFDING INEQUALITY

We first introduce the Chernoff-Hoeffding inequality, which is useful in the analysis.

**Fact B.1** (Chernoff-Hoeffding Inequality (Hoeffding, 1994))**.** *When $X_1, X_2, \ldots, X_N$ are identical independent random variables such that $X_i \in [0, 1]$ and $\mathbb{E}[X_i] = \mu_i$, we have the following*

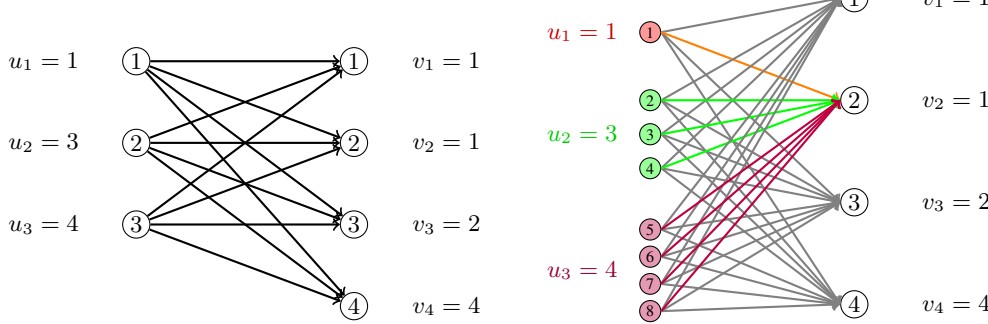

Figure 3: A simple sketch of the MP-CSB problem applied to the OT problem.

Figure 4: An illustration of the duplicating technique. The total number of the duplicated base arms is $d' = \left( \sum_{x=1}^{n_s} u_x \right) \cdot n_d = 32$.

*inequalities:*

$$\Pr\left[ \frac{\sum_{i=1}^{N} X_i}{N} \geq \mu_i + \epsilon \right] \leq \exp\left( -2\epsilon^2 N \right), \tag{7}$$

$$\Pr\left[ \frac{\sum_{i=1}^{N} X_i}{N} \leq \mu_i + \epsilon \right] \leq \exp\left( -2\epsilon^2 N \right). \tag{8}$$

### B.2 NOTATIONS

We use $p_i(t)$ and $q_i(t)$ to denote the value of $p_i$ and $q_i$ at the beginning of time $t$. Let

$$\hat{\mu}_i(t) = \frac{p_i(t) - 1}{N_i(t)} = \frac{1}{N_i(t)} \sum_{\tau:\tau < t, i \in I_{\boldsymbol{a}(t)}} \sum_{j=1}^{a_i(t)} Y_{i,j}(t) \tag{9}$$

be the empirical mean of arm $i$ at the beginning of time $t$, where $N_i(t) = p_i(t) + q_i(t) - 2$ is the number of observations of arm $i$ at the beginning of time $t$. Notice that for fixed arm $i$, in different time $t$ with $i \in I_{\boldsymbol{a}(t)}$ and $j \in [a_i(t)]$, $X_{i,j}(t)$'s are i.i.d with mean $\ell_i$, and $Y_{i,j}(t)$ is a Bernoulli random variable with mean $X_{i,j}$ thus the Bernoulli random variables $Y_{i,j}(t)$'s are also i.i.d. with mean $\ell_i$.

Let us define $M^* = \|\boldsymbol{a}^*\|_1$. Also, let $\epsilon$ be an arbitrary real number that satisfies   Based on $\hat{\mu}_i(t)$, we can define the following five events :

- $\mathcal{P}(t) = \{\boldsymbol{a}(t) \neq \boldsymbol{a}^*\}$

- $\mathcal{Q}(t) = \left\{ \exists i \in I_{\boldsymbol{a}(t)}, |\hat{\mu}_i(t) - \ell_i| > \frac{\epsilon}{\|\boldsymbol{a}(t)\|_1} \right\}$

- $\mathcal{R}(t) = \left\{ \sum_{i \in I_{\boldsymbol{a}(t)}} a_i(t)|\theta_i(t) - \ell_i| > \frac{\Delta_{\boldsymbol{a}(t)}}{\kappa_r} - \left( M^{*2} + 1 \right)\epsilon \right\}$

- $\mathcal{S}(t) = \left\{ \sum_{i \in I_{\boldsymbol{a}(t)}} a_i(t)|\theta_i(t) - \hat{\mu}_i(t)| > \frac{\Delta_{\boldsymbol{a}(t)}}{\kappa_r} - \left( M^{*2} + 2 \right)\epsilon \right\}$

- $\mathcal{T}(t) = \left\{ \sum_{i \in I_{\boldsymbol{a}(t)}} \frac{1}{N_i(t)} \leq \frac{2\left( \frac{\Delta_{\boldsymbol{a}(t)}}{\kappa_r} - \left( M^{*2} + 2 \right)\epsilon \right)^2}{\log(2^d |\mathcal{A}| T)} \right\}$

## B.3 PROOF OF THEOREM 3.1

The total regret can be written as follows:

$$\sum_{t=1}^{T} \mathbb{E}\left[\mathbb{1}\left[\mathcal{P}(t)\right] \times \Delta_{\boldsymbol{a}(t)}\right]$$

$$\leq \sum_{t=1}^{T} \mathbb{E}\left[\mathbb{1}\left[\mathcal{Q}(t) \wedge \mathcal{P}(t)\right] \times \Delta_{\boldsymbol{a}(t)}\right] + \sum_{t=1}^{T} \mathbb{E}\left[\mathbb{1}\left[\neg\mathcal{Q}(t) \wedge \mathcal{R}(t) \wedge \mathcal{P}(t)\right] \times \Delta_{\boldsymbol{a}(t)}\right]$$

$$+ \sum_{t=1}^{T} \mathbb{E}\left[\mathbb{1}\left[\neg\mathcal{R}(t) \wedge \mathcal{P}(t)\right] \times \Delta_{\boldsymbol{a}(t)}\right]. \tag{10}$$

We analyze each term in the RHS of (10).

### B.3.1 THE FIRST TERM OF THE RHS OF (10)

We can use the following lemma to bound the first term. Below, we denote $\hat{\mu}_i(t)$ by the sample mean of arm $i$ at the beginning of round $t$.

**Lemma B.1.** *In Algorithm 1, we have*

$$\mathbb{E}\left[\left\{t \in [T] \mid i \in I_{\boldsymbol{a}(t)}, |\hat{\mu}_i(t) - \ell_i| > \epsilon\right\}\right] \leq 1 + \frac{1}{\epsilon^2}$$

*for any $1 \leq i \leq d$.*

*Proof.* Let $\tau_1, \tau_2, \ldots$ be the time slots such that $i \in I_{\boldsymbol{a}(t)}$ and define $\tau_0 = 0$, then

$$\mathbb{E}\left[\left|\left\{t \in [T] \mid i \in I_{\boldsymbol{a}(t)}, |\hat{\mu}_i(t) - \ell_i| > \epsilon\right\}\right|\right]$$

$$= \mathbb{E}\left[\sum_{t=1}^{T} \mathbb{1}\left[i \in I_{\boldsymbol{a}(t)}, |\hat{\mu}_i(t) - \ell_i| > \epsilon\right]\right]$$

$$\leq \mathbb{E}\left[\sum_{w=0}^{T} \mathbb{E}\left[\sum_{t=\tau_w}^{\tau_{w+1}-1} \mathbb{1}\left[i \in I_{\boldsymbol{a}(t)}, |\hat{\mu}_i(t) - \ell_i| > \epsilon\right]\right]\right]$$

$$\leq \mathbb{E}\left[\sum_{w=0}^{n_i T} \Pr\left[|\hat{\mu}_i(t) - \ell_i| > \epsilon, N_i = w\right]\right]$$

$$\leq 1 + \sum_{w=1}^{n_i T} \Pr\left[|\hat{\mu}_i(t) - \ell_i| > \epsilon, N_i = w\right]$$

$$\leq 1 + \sum_{w=1}^{T} \exp\left(-2w\epsilon^2\right) + \sum_{w=1}^{T} \exp\left(-2w\epsilon^2\right) \tag{11}$$

$$\leq 1 + 2 \sum_{w=1}^{\infty} \left(\exp\left(-2w\epsilon^2\right)\right)^w$$

$$\leq 1 + 2 \frac{\exp\left(-2\epsilon^2\right)}{1 + \exp\left(-2\epsilon^2\right)}$$

$$\leq 1 + \frac{2}{2\epsilon^2}$$

$$= 1 + \frac{1}{\epsilon^2}$$

where Eq (11) is because of the Chernoff-Hoeffding's inequality (Fact B.1). $\square$

By Lemma B.1, we know that the first term is upper bounded by $\left(\frac{dM^2}{\epsilon^2} + d\right) \Delta_{\max}$, where $M = \max_{\boldsymbol{a} \in \mathcal{A}} \|\boldsymbol{a}\|_1$.

B.3.2   THE SECOND TERM OF THE RHS OF (10)

Under $\neg\mathcal{Q}(t) \wedge \mathcal{R}(t)$, we must have that

$$
\begin{aligned}
\sum_{i=1}^{d} a_i(t)|\theta_i(t) - \hat{\mu}_i(t)| &\geq \sum_{i=1}^{d} a_i(t)|\theta_i(t) - \ell_i| - \sum_{i=1}^{d} a_i(t)|\ell_i - \hat{\mu}_i(t)| \\
&> \frac{\Delta_{\boldsymbol{a}(t)}}{\kappa_r} - (M^{*2} + 1)\epsilon^2 - \epsilon, \\
&= \frac{\Delta_{\boldsymbol{a}(t)}}{\kappa_r} - (M^{*2} + 2)\epsilon^2
\end{aligned}
\tag{12}
$$

i.e., event $\mathcal{S}(t)$ must happen.

Then, the second term of the RHS of (10) can be bounded by

$$
\sum_{t=1}^{T} \mathbb{E}\left[\mathbb{1}\left[\neg\mathcal{Q}(t) \wedge \mathcal{R}(t) \wedge \mathcal{P}(t)\right] \times \Delta_{\boldsymbol{a}(t)}\right]
$$

$$
\leq \sum_{t=1}^{T} \mathbb{E}\left[\mathbb{1}\left[\mathcal{S}(t) \wedge \mathcal{T}(t) \wedge \mathcal{P}(t)\right] \times \Delta_{\boldsymbol{a}(t)}\right] + \sum_{t=1}^{T} \mathbb{E}\left[\mathbb{1}\left[\mathcal{S}(t) \wedge \neg\mathcal{T}(t) \wedge \mathcal{P}(t)\right] \times \Delta_{\boldsymbol{a}(t)}\right].
$$

Following the same discussion in Perrault et al. (2020), we can obtain $\Pr\left[\mathcal{S}(t) \wedge \mathcal{T}(t)\right] \leq \mathcal{O}\left(\frac{1}{T}\right)$, and therefore, $\mathbb{E}\left[\mathbb{1}\left[\mathcal{S}(t) \wedge \mathcal{T}(t) \wedge \mathcal{P}(t)\right] \times \Delta_{\boldsymbol{a}(t)}\right]$ is $\mathcal{O}(1)$.

Now, we bound the regret term $\mathbb{E}\left[\mathbb{1}\left[\mathcal{S}(t) \wedge \neg\mathcal{T}(t) \wedge \mathcal{P}(t)\right]\right]$. Here, we use the regret allocation method to count this regret term. That is, for any time step $t$ such that $\mathcal{S}(t) \wedge \neg\mathcal{T}(t) \wedge \mathcal{P}(t)$ happens, we allocate regret $g_i(N_i(t))$ to each base arm $i \in I_{\boldsymbol{a}(t)}$. We say the allocation function $g_i$'s are correct if the sum of allocated regret in this step is larger than $\Delta_{\boldsymbol{a}(t)}$, i.e., $\sum_{i=1}^{d} g_i(N_i(t)) \geq \Delta_{\boldsymbol{a}(t)}$.

Then, we describe our allocation function $g_i$'s. Here, we define

$$
L_{i,1} = \frac{m \log\left(2^d |\mathcal{A}| T\right)}{\min_{\boldsymbol{a}: i \in I_{\boldsymbol{a}}}\left(\frac{\Delta_{\boldsymbol{a}}}{\kappa_r} - (M^{*2} + 2)\epsilon\right)^2}
\tag{13}
$$

and

$$
L_{i,2} = \frac{\log\left(2^d |\mathcal{A}| T\right)}{\min_{\boldsymbol{a}: i \in I_{\boldsymbol{a}}}\left(\frac{\Delta_{\boldsymbol{a}}}{\kappa_r} - (M^{*2} + 2)\epsilon\right)^2}.
\tag{14}
$$

Also, we define $g_i(w)$ as follows:

$$
g_i(w) = \begin{cases}
\Delta_{\max} & (w = 0) \\
2\kappa_r \sqrt{\frac{\log(2^d |\mathcal{A}| T)}{w}} & 0 < w < L_{i,2} \\
\frac{2\kappa_r \log\left(2^d |\mathcal{A}| T\right)}{w \min_{\boldsymbol{a}: i \in I_{\boldsymbol{a}}}\left(\frac{\Delta_{\boldsymbol{a}}}{\kappa_r} - (M^{*2}+2)\epsilon\right)} & L_{i,2} < w \leq L_{i,1} \\
0 & w > L_{i,1}
\end{cases}
\tag{15}
$$

Now, we prove that these allocation function $g_i$'s satisfy the correctness condition when $\epsilon \leq \frac{\Delta_{\min}}{2\kappa_r(M^{*2}+2)}$, i.e., if event $\mathcal{S}(t) \wedge \neg\mathcal{T}(t) \wedge \mathcal{P}(t)$ happens, then $\sum_{i=1}^{d} g_i(N_i(t)) \geq \Delta_{\boldsymbol{a}(t)}$.

If there exists $i \in I_{\boldsymbol{a}(t)}$ such that $N_i(t) = 0$, then $g_i(N_i(t)) = \Delta_{\max} \geq \Delta_{\boldsymbol{a}(t)}$. Since $g_i(w)$ is always non-negative, we know that $\sum_{i \in I_{\boldsymbol{a}(t)}} g_i(w) \geq \Delta_{\boldsymbol{a}(t)}$.

If there exists $i \in I_{\boldsymbol{a}(t)}$ such that $1 \le N_i(t) \le \frac{\log\left(2^d |\mathcal{A}|T\right)}{\min\limits_{\boldsymbol{a}:i \in I_{\boldsymbol{a}}} \left(\frac{\Delta_{\boldsymbol{a}}}{\kappa_r} - (M^{*2}+2)\epsilon\right)^2}$, then $N_i(t) \le L_{i,2}$, and

therefore,

$$g_i(t) = 2\kappa_r \sqrt{\frac{\log\left(2^d |\mathcal{A}|T\right)}{N_i(t)}} \ge 2\kappa_r \sqrt{\frac{\log\left(2^d |\mathcal{A}|T\right)}{\frac{\log\left(2^d |\mathcal{A}|T\right)}{\left(\frac{\Delta_{\boldsymbol{a}(t)}}{\kappa_r} - (M^{*2}+2)\epsilon\right)^2}}} = 2\kappa_r \left(\frac{\Delta_{\boldsymbol{a}(t)}}{\kappa_r} - (M^{*2}+2)\epsilon\right) \ge \Delta_{\boldsymbol{a}(t)},$$

where the last inequality is because that $\epsilon \le \frac{\Delta_{\min}}{2\kappa_r(M^{*2}+2)}$ and $a_i(t) \ge 1$. From the above inequalities, we know that $\sum\limits_{i \in I_{\boldsymbol{a}(t)}} g_i(t) \ge \Delta_{\boldsymbol{a}(t)}$.

If for all $i \in I_{\boldsymbol{a}(t)}$, $N_i(t) > \frac{\log\left(2^d |\mathcal{A}|T\right)}{\left(\frac{\Delta_{\boldsymbol{a}(t)}}{\kappa_r} - (M^{*2}+2)\epsilon\right)^2}$, then we use $S^1_{\boldsymbol{a}(t)}$ to denote the set of arms $i \in I_{\boldsymbol{a}(t)}$ such that $N_i(t) > L_{i,1}$, $S^2_{\boldsymbol{a}(t)}$ to denote the set of arms $i \in I_{\boldsymbol{a}(t)}$ such that $L_{i,2} < N_i(t) < L_{i,1}$, $S^3_{\boldsymbol{a}(t)}(t)$ to denote the set of arms $i \in I_{\boldsymbol{a}(t)}$ such that $N_i(t) \le L_{i,2}$. By the definition of allocation

functions $g_i$'s, we have that

$$\sum_{ii \in I_{\boldsymbol{a}(t)}} g_i(N_i(t))$$

$$= \sum_{i \in I^3_{\boldsymbol{a}(t)}} 2\kappa_r \sqrt{\frac{\log\left(2^d|\mathcal{A}|T\right)}{N_i(t)}} + \sum_{i \in I^2_{\boldsymbol{a}(t)}} \frac{2\kappa_r \log\left(2^d|\mathcal{A}|T\right)}{N_i(t) \min\limits_{\boldsymbol{a}:i \in I_{\boldsymbol{a}}}\left(\frac{\Delta_{\boldsymbol{a}(t)}}{\kappa_r} - \left(M^{*2}+2\right)\epsilon\right)}$$

$$\geq \sum_{i \in I^3_{\boldsymbol{a}(t)}} 2\kappa_r \sqrt{\frac{\log\left(2^d|\mathcal{A}|T\right)}{N_i(t)}} + \sum_{i \in I^2_{\boldsymbol{a}(t)}} \frac{2\kappa_r \log\left(2^d|\mathcal{A}|T\right)}{N_i(t)\left(\frac{\Delta_{\boldsymbol{a}(t)}}{\kappa_r} - \left(M^{*2}+2\right)\epsilon\right)}$$

$$= \sum_{i \in I^3_{\boldsymbol{a}(t)}} 2\kappa_r \frac{\log\left(2^d|\mathcal{A}|T\right)}{N_i(t)\left(\frac{\Delta_{\boldsymbol{a}(t)}}{\kappa_r} - \left(M^{*2}+2\right)\epsilon\right)} \cdot \sqrt{\frac{N_i(t)\left(\frac{\Delta_{\boldsymbol{a}(t)}}{\kappa_r} - \left(M^{*2}+2\right)\epsilon\right)^2}{\log\left(2^d|\mathcal{A}|T\right)}}$$

$$+ \sum_{i \in I^2_{\boldsymbol{a}(t)}} \frac{2\kappa_r \log\left(2^d|\mathcal{A}|T\right)}{N_i(t)\left(\frac{\Delta_{\boldsymbol{a}(t)}}{\kappa_r} - \left(M^{*2}+2\right)\epsilon\right)}$$

$$\geq \sum_{i \in I^3_{\boldsymbol{a}(t)}} \frac{2\kappa_r \log\left(2^d|\mathcal{A}|T\right)}{N_i(t)\left(\frac{\Delta_{\boldsymbol{a}(t)}}{\kappa_r} - \left(M^{*2}+2\right)\epsilon\right)} + \sum_{i \in I^2_{\boldsymbol{a}(t)}(t)} \frac{2\kappa_r \log\left(2^d|\mathcal{A}|T\right)}{N_i(t)\left(\frac{\Delta_{\boldsymbol{a}(t)}}{\kappa_r} - \left(M^{*2}+2\right)\epsilon\right)} \qquad (16)$$

$$= \sum_{i \in I_{\boldsymbol{a}(t)} \setminus I^1_{\boldsymbol{a}(t)}(t)} \frac{2\kappa_r \log\left(2^d|\mathcal{A}|T\right)}{N_i(t)\left(\frac{\Delta_{\boldsymbol{a}(t)}}{\kappa_r} - \left(M^{*2}+2\right)\epsilon\right)}$$

$$= \frac{2\kappa_r \log\left(2^d|\mathcal{A}|T\right)}{\left(\frac{\Delta_{\boldsymbol{a}(t)}}{\kappa_r} - \left(M^{*2}+2\right)\epsilon\right)}\left(\sum_{i \in I_{\boldsymbol{a}(t)}} \frac{1}{N_i(t)} - \sum_{i \in I^1_{\boldsymbol{a}(t)}} \frac{1}{N_i(t)}\right)$$

$$\geq \frac{2\kappa_r \log\left(2^d|\mathcal{A}|T\right)}{\left(\frac{\Delta_{\boldsymbol{a}(t)}}{\kappa_r} - \left(M^{*2}+2\right)\epsilon\right)}\left(\frac{2\left(\frac{\Delta_{\boldsymbol{a}(t)}}{\kappa_r} - \left(M^{*2}+2\right)\epsilon\right)^2}{\log\left(2^d|\mathcal{A}|T\right)} - \sum_{i \in I^1_{\boldsymbol{a}(t)}} \frac{\left(\frac{\Delta_{\boldsymbol{a}(t)}}{\kappa_r} - \left(M^{*2}+2\right)\epsilon\right)^2}{m\log\left(2^d|\mathcal{A}|T\right)}\right)$$

$$\qquad (17)$$

$$\geq \frac{2\kappa_r \log\left(2^d|\mathcal{A}|T\right)}{\left(\frac{\Delta_{\boldsymbol{a}(t)}}{\kappa_r} - \left(M^{*2}+2\right)\epsilon\right)}\left(\frac{2\left(\frac{\Delta_{\boldsymbol{a}(t)}}{\kappa_r} - \left(M^{*2}+2\right)\epsilon\right)^2}{\log\left(2^d|\mathcal{A}|T\right)} - m\frac{\left(\frac{\Delta_{\boldsymbol{a}(t)}}{\kappa_r} - \left(M^{*2}+2\right)\epsilon\right)^2}{m\log\left(2^d|\mathcal{A}|T\right)}\right)$$

$$= \frac{2\kappa_r \log\left(2^d|\mathcal{A}|T\right)}{\left(\frac{\Delta_{\boldsymbol{a}(t)}}{\kappa_r} - \left(M^{*2}+2\right)\epsilon\right)} \frac{\left(\frac{\Delta_{\boldsymbol{a}(t)}}{\kappa_r} - \left(M^{*2}+2\right)\epsilon\right)^2}{\log\left(2^d|\mathcal{A}|T\right)}$$

$$= 2\kappa_r\left(\frac{\Delta_{\boldsymbol{a}(t)}}{\kappa_r} - \left(M^{*2}+2\right)\epsilon\right)$$

$$\geq \Delta_{\boldsymbol{a}(t)}.$$

Here, Eq(16) is because that $N_i(t) > \frac{\log\left(2^d|\mathcal{A}|T\right)}{\left(\frac{\Delta_{\boldsymbol{a}(t)}}{\kappa_r} - (M^{*2}+2)\epsilon\right)^2}$ (as we assumed in the beginning of the paragraph), Eq (17) comes from the definition of $\neg\mathcal{T}(t)$ (the first term) and the definition of $S^1_{\boldsymbol{a}(t)}$ (the second term). This finishes the proof that the allocation functions $g_i$'s satisfy the correctness condition when $\epsilon \leq \frac{\Delta_{\min}}{2\kappa_r(M^{*2}+2)}$.

Because of this, the second term of (10) is upper-bounded by

$$\mathbb{E}\left[\mathbb{1}\left[\neg\mathcal{Q}(t) \wedge \mathcal{R}(t) \wedge \mathcal{P}(t)\right]\right]$$

$$\leq (d+1)\Delta_{\max} + \sum_{i=1}^{d}\sum_{w=1}^{L_{i,2}} 2\kappa_r \sqrt{\frac{\log\left(2^d|\mathcal{A}|T\right)}{w}} + \sum_{i=1}^{d}\sum_{w=L_{i,2}+1}^{L_{i,1}} \frac{1}{w} \frac{2\kappa_r \log\left(2^d|\mathcal{A}|T\right)}{\min\limits_{\boldsymbol{a}:i\in I_{\boldsymbol{a}(t)}}\left(\frac{\Delta_{\boldsymbol{a}}}{\kappa_r} - \left(M^{*2} + 2\right)\epsilon\right)}$$

$$\leq (d+1)\Delta_{\max} + \sum_{i=1}^{d} 4\sqrt{\log\left(2^d|\mathcal{A}|T\right)L_{i,2}} + \sum_{i=1}^{d}\left(1 + \log\left(\frac{L_{i,1}}{L_{i,2}}\right)\right) \frac{2\kappa_r \log\left(2^d|\mathcal{A}|T\right)}{\min\limits_{\boldsymbol{a}:i\in I_{\boldsymbol{a}(t)}}\left(\frac{\Delta_{\boldsymbol{a}}}{\kappa_r} - \left(M^{*2} + 2\right)\epsilon\right)}$$

$$\tag{18}$$

Here, Eq (18) is because that $\sum_{w=1}^{\kappa_r}\sqrt{\frac{1}{w}} \leq 2\sqrt{\kappa_r}$ (by as simple inductive proof on $N$) and $\sum_{w=N_1}^{N_2}\frac{1}{w} \leq 1 + \log\frac{N_2}{N_1}$.

The value $\sqrt{\log\left(2^d|\mathcal{A}|T\right)L_{i,2}}$ equals to $\frac{\log\left(2^d|\mathcal{A}|T\right)}{\min\limits_{\boldsymbol{a}\in\mathcal{A}:i\in I_{\boldsymbol{a}}}\left(\frac{\Delta_{\boldsymbol{a}}}{\kappa_r} - (M^{*2}+2)\epsilon\right)}$, and $\log\frac{L_{i,1}}{L_{i,2}} = \log m$, and therefore, the total regret in the second term is

$$\mathcal{O}\left(\sum_{i=1}^{d} \frac{\kappa_r \log m \log\left(T\right)}{\min\limits_{\boldsymbol{a}:i\in I_{\boldsymbol{a}}}\left(\frac{\Delta_{\boldsymbol{a}}}{\kappa_r} - \left(M^{*2} + 2\right)\epsilon\right)}\right), \tag{19}$$

and if set $\epsilon = \frac{\Delta}{(M-^{*2}+2)\epsilon}$, the second term is

$$\mathcal{O}\left(\sum_{i=1}^{d} \frac{\kappa_r{}^2 \log m}{\Delta_i} \log T\right) \tag{20}$$

### B.3.3 The Third Term of the RHS of (10)

Let $\boldsymbol{\theta} = (\theta_1, \ldots, \theta_d)$ be a vector of parameters, $I \subseteq [d]$ and $I \neq \emptyset$ be some arm set, and $\boldsymbol{V}^c$ be the complement of $\boldsymbol{V}$. Recall that $\boldsymbol{\theta}_V$ is a vector whose $i$-th element is $\theta_i$ if $i \in V$ and 0 if $i \notin V$. Also, we use the notation $(\boldsymbol{\theta}'_{\boldsymbol{V}}, \boldsymbol{\theta}_{\boldsymbol{V}^c})$ to denote replacing $\theta_i$'s for $i \in V$ and keeping the values $\theta_i$ for $i \in V^c$ unchanged.

Given a subset $I \subseteq I_{\boldsymbol{a}^*}$, we consider the following property for $\boldsymbol{\theta}_{I^c}$. For any $\boldsymbol{\theta}'_Z$ such that $\|\boldsymbol{\theta}'_Z - \boldsymbol{\ell}_Z\|_\infty \leq \epsilon$, let $\boldsymbol{\theta}' = (\boldsymbol{\theta}'_Z, \boldsymbol{\theta}_{I^c})$, then:

- $I \subseteq I_{\mathsf{Oracle}(\boldsymbol{\theta}')}$
- Either $\mathsf{Oracle}(\boldsymbol{\theta}') = \boldsymbol{a}^*$ or $\|\mathsf{Oracle}\left(\boldsymbol{\theta}'\right) \cdot \left(\boldsymbol{\theta}' - \boldsymbol{\ell}\right)\|_1 \geq \Delta_{\mathsf{Oracle}(\boldsymbol{\theta}')} - \left(M^{*2} + 2\right)\epsilon$

The first one is to make sure that if we have normal samples in $I$ at time $t$ (i.e., the samples value $\theta_i(t)$ is within $\epsilon$ neighborhood of $\ell_i$ for all $i \in Z$), then all the arms in $I$ will be played and observed. These observations would update the beta distributions of these base arms to be more accurate, such that the probability of the next time that the samples from these base arms are also within $\epsilon$ neighborhood of their true mean value becomes larger. This fact would be used later in the quantitative regret analysis. The second one says that if the samples in $I$ are normal, then $\neg\mathcal{R}(t) \wedge \mathcal{P}(t)$ can not happen. We use $\mathcal{E}_{Z,1}\left(\boldsymbol{\theta}\right)$ to denote the event that the vector $\boldsymbol{\theta}_{I^c}$ has such a property, and emphasize that this event only depends on the values in vector $\boldsymbol{\theta}_{I^c}$.

What we want to do is to find some exact $I$ such that $\mathcal{E}_{Z,1}\left(\boldsymbol{\theta}(t)\right)$ happens when $\neg\mathcal{R}(t) \wedge \mathcal{P}(t)$ happens. If such $I$ exists, then for any $t$ such that $\mathcal{E}_{Z,1}\left(\boldsymbol{\theta}(t)\right)$ happens, there are two possible cases: i) the samples of all arms $i \in Z$ are normal, which means $\neg\mathcal{R}(t) \wedge \mathcal{P}(t)$ cannot happen, and will update the posterior distributions of all the arms $i \in Z$ to increase the probability that the samples of all the arms $i \in Z$ are normal; ii) the samples of some arms $i \in Z$ are not normal, and $\neg\mathcal{R}(t) \wedge \mathcal{P}(t)$ may happen in this case. As time goes on, the probability that the samples in $I$ are normal becomes

larger and larger, and therefore the probability that $\neg \mathcal{R}(t) \wedge \mathcal{P}(t)$ happens becomes smaller and smaller. Thus, $\sum_{t=1}^{T} \mathbb{E}\left[\mathbf{1}\left[\neg \mathcal{R}(t) \wedge \mathcal{P}(t)\right]\right]$ has a constant upper bound.

The following lemma shows that such $I$ must exist, and it is the key lemma in the analysis of the third term.

**Lemma B.2.** *Suppose that $\neg \mathcal{R}(t) \wedge \mathcal{P}(t)$ happens, then there exists $I \subseteq I_{\boldsymbol{a}^*}$ and $I \neq \emptyset$ such that $\mathcal{E}_{Z,1}(\boldsymbol{\theta}(t))$ holds.*

*Proof.* Firstly, consider the case that we choose $I = I_{\boldsymbol{a}^*}$, i.e., we change $\boldsymbol{\theta}_{I_{\boldsymbol{a}^*}}(t)$ to some $\boldsymbol{\theta}'_{I_{\boldsymbol{a}^*}}$ with $\|\boldsymbol{\theta}'_{I_{\boldsymbol{a}^*}} - \boldsymbol{\ell}_{I_{\boldsymbol{a}^*}}\|_\infty \leq \epsilon$ and get a new vector $\boldsymbol{\theta}' = \left(\boldsymbol{\theta}'_{I_{\boldsymbol{a}^*}}, \boldsymbol{\theta}_{I_{\boldsymbol{a}^*}^c}(t)\right)$. We claim that for any $\boldsymbol{a}'$ such that $I_{\boldsymbol{a}'} \wedge I_{\boldsymbol{a}^*} = \emptyset$, $\mathsf{Oracle}\,(\boldsymbol{\theta}') \neq \boldsymbol{a}'$. This is because

$$\langle \boldsymbol{a}', \boldsymbol{\theta}' \rangle = \langle \boldsymbol{a}', \boldsymbol{\theta}(t) \rangle \tag{21}$$

$$\leq \langle \boldsymbol{a}(t), \boldsymbol{\theta}(t) \rangle \tag{22}$$

$$\leq \langle \boldsymbol{a}(t), \boldsymbol{\ell} \rangle + \left(\Delta_{\boldsymbol{a}(t)} - \left(M^{*2} + 1\right)\epsilon\right) \tag{23}$$

$$\leq \langle \boldsymbol{a}^*, \boldsymbol{\ell} \rangle - \left(M^{*2} + 1\right)\epsilon \tag{24}$$

$$< \langle \boldsymbol{a}^*, \boldsymbol{\ell} \rangle - M^* \epsilon \tag{25}$$

$$\leq \langle \boldsymbol{a}^*, \boldsymbol{\theta}' \rangle \tag{26}$$

Eq (21) is because $\boldsymbol{\theta}'$ and $\boldsymbol{\theta}(t)$ only differs on arms in $I_{\boldsymbol{a}^*}$ but $I_{\boldsymbol{a}'} \cap I_{\boldsymbol{a}^*} \neq \emptyset$. Eq (22) is by the optimality of $\boldsymbol{a}(t)$ on input $\boldsymbol{\theta}(t)$. Eq (23) is by the event $\neg \mathcal{R}(t)$. Eq (24) is by the definition of $\Delta_{\boldsymbol{a}(t)}$. Eq (26) again uses the Lipschitz continuity. Thus, the claim holds.

We have two possibilities for $\mathsf{Oracle}\,(\boldsymbol{\theta}')$:

1a) for all $\boldsymbol{\theta}'_{I_{\boldsymbol{a}^*}}$ with $\|\boldsymbol{\theta}'_{I_{\boldsymbol{a}^*}} - \boldsymbol{\ell}_{I_{\boldsymbol{a}^*}}\|_\infty \leq \epsilon$, $I_{\boldsymbol{a}^*} \subseteq I_{\mathsf{Oracle}(\boldsymbol{\theta}')}$

1b) for some $\boldsymbol{\theta}'_{I_{\boldsymbol{a}^*}}$ with $\|\boldsymbol{\theta}'_{I_{\boldsymbol{a}^*}} - \boldsymbol{\ell}_{I_{\boldsymbol{a}^*}}\|_\infty \leq \epsilon$, $\mathsf{Oracle}\,(\boldsymbol{\theta}') = \boldsymbol{a}^1$ where $I_{\boldsymbol{a}^1} \cap I_{\boldsymbol{a}^*} = I_1$ and $I_1 \neq I_{\boldsymbol{a}^*}$, $I_1 \neq \emptyset$.

In 1a), let $\boldsymbol{a}^0 = \mathsf{Oracle}\,(\boldsymbol{\theta}')$. Then, we have $\langle \boldsymbol{a}^0, \boldsymbol{\theta}' \rangle \geq \langle \boldsymbol{a}^*, \boldsymbol{\theta}' \rangle \geq \langle \boldsymbol{a}^*, \boldsymbol{\ell} \rangle - M^* \epsilon$. If $\boldsymbol{a}^0 \notin \mathsf{OPT}$, we have $\langle \boldsymbol{a}^*, \boldsymbol{\ell} \rangle = \langle \boldsymbol{a}^0, \boldsymbol{\ell} \rangle + \Delta_{\boldsymbol{a}^0}$. Together, we have $\langle \boldsymbol{a}^0, \boldsymbol{\theta}' \rangle \geq \langle \boldsymbol{a}^0, \boldsymbol{\ell} \rangle + \Delta_{\boldsymbol{a}^0} - M^* \epsilon$. This implies that $\|\boldsymbol{a}^0 \cdot (\boldsymbol{\theta}' - \boldsymbol{\ell})\|_1 \geq \Delta_{\boldsymbol{a}^0} - M^* \epsilon > \Delta_{\boldsymbol{a}^0} - \left(M^{*2} + 1\right)\epsilon > \Delta_{\boldsymbol{a}^0} - \left(M^{*2} + 1\right)\epsilon$. That is, we conclude that either $\boldsymbol{a}^0 \in \mathsf{OPT}$ or $\|\boldsymbol{a}^0 \cdot (\boldsymbol{\theta}' - \boldsymbol{\ell})\|_1 \geq \Delta_{\boldsymbol{a}^0} - M^* \epsilon > \Delta_{\boldsymbol{a}^0} - \left(M^{*2} + 1\right)\epsilon$, which means that $\mathcal{E}_{\boldsymbol{a}^*,1}(\boldsymbol{\theta}'(t)) = \mathcal{E}_{\boldsymbol{a}^*,1}(\boldsymbol{\theta}(t))$ holds.

Next, we consider 1b). Fix a $\boldsymbol{\theta}'_{I_{\boldsymbol{a}^*}}$ with $\|\boldsymbol{\theta}'_{I_{\boldsymbol{a}^*}} - \boldsymbol{\ell}_{I_{\boldsymbol{a}^*}}\|_\infty \leq \epsilon$. Let $\boldsymbol{a}^1 = \mathsf{Oracle}\,(\boldsymbol{\theta}')$ which does not equal to $\boldsymbol{a}^*$. Then $\langle \boldsymbol{a}^1, \boldsymbol{\theta}' \rangle \geq \langle \boldsymbol{a}^*, \boldsymbol{\theta}' \rangle \geq \langle \boldsymbol{a}^*, \boldsymbol{\ell} \rangle - M^* \epsilon$.

Now we try to choose $I = I_1$. For all $\boldsymbol{\theta}'_{I_1}$ with $\|\boldsymbol{\theta}'_{I_1} - \boldsymbol{\ell}_{I_1}\|_\infty \leq \epsilon$, consider $\boldsymbol{\theta}' = \left(\boldsymbol{\theta}'_{I_1}, \boldsymbol{\theta}_{I_1^c}(t)\right)$. We see that $\|\boldsymbol{a}^1\,(\boldsymbol{\theta} - \boldsymbol{\ell})\| \leq 2\,(M^* - 1)\,\epsilon$. Thus,

$$\langle \boldsymbol{a}^1, \boldsymbol{\theta}' \rangle \geq \langle \boldsymbol{a}^*, \boldsymbol{\ell} \rangle - M^* \epsilon - 2(M^* - 1)\epsilon$$
$$= \langle \boldsymbol{a}^*, \boldsymbol{\ell} \rangle - (3M^* - 2)$$

Similarly, we have the following inequalities for any $I_{\boldsymbol{a}'} \cap Z_1 = \emptyset$:

$$\langle \boldsymbol{a}', \boldsymbol{\theta}' \rangle = \langle \boldsymbol{a}', \boldsymbol{\theta}(t) \rangle$$
$$\leq \langle \boldsymbol{a}(t), \boldsymbol{\theta}(t) \rangle$$
$$\leq \langle \boldsymbol{a}(t), \boldsymbol{\ell} \rangle + \left(\Delta_{\boldsymbol{a}(t)} - \left(M^{*2} + 1\right)\epsilon\right)$$
$$\leq \langle \boldsymbol{a}^*, \boldsymbol{\ell} \rangle - (M^{*2} + 1)\epsilon \tag{27}$$
$$< \langle \boldsymbol{a}^*, \boldsymbol{\ell} \rangle - (3M^* - 2) \tag{28}$$
$$\leq \langle \boldsymbol{a}^1, \boldsymbol{\theta}' \rangle$$

That is, $I_{\mathsf{Oracle}(\boldsymbol{\theta}')} \cap I_1 \neq \emptyset$. Thus, we will also have two possibilities:

2a) for all $\boldsymbol{\theta}_{I_1}$ with $\|\boldsymbol{\theta}'_{I_1} - \boldsymbol{\ell}_{I_1}\|_\infty \le \epsilon$, $I_1 \subseteq I_{\mathsf{Oracle}(\boldsymbol{\theta})}$

2b) for some $\boldsymbol{\theta}'_{I_1}$ with $\|\boldsymbol{\theta}'_{I_1} - \boldsymbol{\ell}_{I_1}\|_\infty \le \epsilon$, $\mathsf{Oracle}(\boldsymbol{\theta}') = \boldsymbol{a}^2$ where $I_{\boldsymbol{a}^2} \cap I_1 = I_2$ and $I_2 \ne I_1$, $I_2 \ne \emptyset$.

We could repeat the above argument and the size of $I_i$ is decreased by at least 1. In the first step, the terms contain $\epsilon$ (in Eq (25)) is $M^*\epsilon$, and in the second step, the terms contain $\epsilon$ (in Eq (28)) becomes $M^*\epsilon + 2(M^* - 1)\epsilon = (3M^* - 2)\epsilon$. Thus, after at most $|I_{\boldsymbol{a}^*}| - 1$ steps, this terms is at most

$$M^* + 2\left(M^* - 1\right) + 2\left(M^* - 2\right) + \cdots + 2 \times 1 = M^{*2}, \tag{29}$$

which is still less than $(M^{*2} + 1)\epsilon$ (in Eq (24) or (27) ). This means that the above analysis works for any steps in the induction procedure. When we reach the end, we could find a $I_i \subseteq I_{\boldsymbol{a}^*}$ and $I_i \ne \emptyset$ such that $\mathcal{E}_{Z_i,1}(\boldsymbol{\theta}(t))$ occurs. $\qquad\square$

By Lemma B.2, for some nonempty $I$, $\mathcal{E}_{Z,1}(\boldsymbol{\theta}(t))$, occurs when $\neg\mathcal{R}(t) \wedge \mathcal{P}(t)$ happens. Another fact is that $\|\boldsymbol{\theta}_Z(t) - \boldsymbol{\ell}_Z\|_\infty > \epsilon$. The reason is that if $\|\boldsymbol{\theta}_Z(t) - \boldsymbol{\ell}_Z\|_\infty \le \epsilon$, by definition of the property, either $\boldsymbol{a}(t) \in \mathsf{OPT}$ or $\|\boldsymbol{a}(t) \cdot (\boldsymbol{\theta} - \boldsymbol{\ell})\|_1 > \Delta_{\boldsymbol{S}(t)} - (M^{*2} + 2)\epsilon$, which means $\neg\mathcal{R}(t) \wedge \mathcal{P}(t)$ can not happen. Let $\mathcal{E}_{Z,2}(\boldsymbol{\theta})$ be the event $\{\|\boldsymbol{\theta}_Z - \boldsymbol{\ell}_Z\|_\infty > \epsilon\}$. Then, $\neg\mathcal{R}(t) \wedge \mathcal{P}(t) \Rightarrow \vee_{Z \subseteq I_{\boldsymbol{a}^*}, Z \ne \emptyset} (\mathcal{E}_{Z,1}(\boldsymbol{\theta}(t)) \wedge \mathcal{E}_{Z,2}(\boldsymbol{\theta}(t)))$.

Following a similar discussion as that of Wang & Chen (2018), we know that $\sum_{Z \subset I_{\boldsymbol{a}}, Z \ne \emptyset} \left(\sum_{t=1}^{T} \mathbb{E}\left[\mathbf{1}\{\mathcal{E}_{Z,1}\boldsymbol{\theta}(t)) \wedge \mathcal{E}_{Z,2}(\boldsymbol{\theta}(t))\}\right]\right)$, and therefore, the third term of the RHS of (10) does not depend on $t$.

### B.3.4 SUM OF ALL TERMS IN THE RHS OF (10)

The regret upper bound of GenCTS is the sum of these three terms, i.e.,

$$\mathcal{O}\left(\sum_{i=1}^{d} \frac{\kappa_r \log m \log\left(2^d |\mathcal{A}| T\right)}{\min_{\boldsymbol{a}: i \in I_{\boldsymbol{a}}}\left(\frac{\Delta_{\boldsymbol{a}}}{\kappa_r} - \left(M^{*2} + 2\right)\epsilon\right)}\right), \tag{30}$$

where $\epsilon \le \frac{\Delta_{\min}}{2\kappa_r(M^{*2}+2)}$.

## C PROOF OF THEOREMS IN SECTION 4

Here, provide proof for theorems in Section 4. We define

$$\alpha_i(t) = \left(\frac{a_i(t)}{n_i}\right)^2 (k_i(t) - q_i(t))^2 \cdot \min\left\{1, \frac{2\left(1 - \frac{x_i(t)}{n_i}\right)}{\left(\frac{x_i(t)}{n_i}\right)^2 \gamma}\right\} \tag{31}$$

### C.1 PREPARATORY LEMMA

We first show a preparatory lemma.

**Lemma C.1.** *Let $D_i^{(1)}$ and $D_i^{(2)}$ denote the Bregman divergence associated with $\phi_i^{(1)}(x) = -n_i \log \frac{x}{n_i}$ and $\phi_i^{(2)} = n_i(1 - \frac{y}{n_i}) \log\left(1 - \frac{y}{n_i}\right)$, respectively. Then, for any $x \in (0, n_i)$, we have*

$$\max_{y \in \mathbb{R}} f_i^{(1)}(y) = \max_{y \in \mathbb{R}}\left\{a(x - y) - D_i^{(1)}(y, x)\right\}$$

$$= n_i g\left(a\frac{x}{n_i}\right) \tag{32}$$

$$\max_{y \in \mathbb{R}} f_i^{(2)}(y) = \max_{y \in \mathbb{R}}\left\{a(x - y) - D_i^{(2)}(y, x)\right\}$$

$$= n_i\left(1 - \frac{x}{n_i}\right) h(a), \tag{33}$$

*where g and h are defined as*

$$g(x) = x - \log(x+1), h(x) = \exp(x) - x - 1. \tag{34}$$

*Proof.* The derivative of $f_i^{(1)}$ is expressed as

$$\frac{df_i^{(1)}(y)}{dx} = -a + \frac{n_i}{y} - \frac{n_i}{x}$$

As $f_i^{(1)}$ is a concave function with respect to $y$, the maximizer $y^*$ of $f_i^{(1)}$ satisfies $a = \frac{n_i}{y^*} - \frac{n_i}{x}$. Hence, the maximum value is expressed as

$$\max_{y \in \mathbb{R}} f_i^{(1)}(y) = f_i^{(1)}(y^*)$$

$$= a(x - y^*) + n_i \log \frac{y^*}{n_i} - n_i \log \frac{x}{n_i} + n_i \frac{x - y^*}{x}$$

$$= -n_i \log \frac{x}{y^*} + n_i \left(\frac{x - y^*}{y^*}\right)$$

$$= -n_i \left(\log \left(1 + a\frac{x}{n_i}\right) + a\frac{x}{n_i}\right)$$

$$= n_i g \left(a\frac{x}{n_i}\right)$$

which proves (32). Similarly, as $f_i^{(2)}$ is a concave function with respect to $y$, the maximizer $y^* \in \mathbb{R}$ of $f_i^{(2)}$ satisfies

$$\frac{df_i^{(2)}}{dy}(y^*) = -a + \log \left(1 - \frac{y^*}{n_i}\right) + 1 - \log \left(1 - \frac{x}{n_i}\right) - 1$$

$$= 0 \tag{35}$$

Hence, we have

$$f_i^{(2)}(y^*) = a(x - y^*) - n_i \left(1 - \frac{y^*}{n_i}\right) \log \left(1 - \frac{y^*}{n_i}\right) + n_i \left(1 - \frac{x}{n_i}\right) \log \left(1 - \frac{x}{n_i}\right)$$

$$- n_i (y^* - x) \left(\log \left(1 - \frac{x}{n_i}\right) + \frac{1}{n_i}\right)$$

$$= n_i \left(1 - \frac{y^*}{n_i}\right) - n_i \left(1 - \frac{x}{n_i}\right) - n_i \left(1 - \frac{x}{n_i}\right) \log \left(1 - \frac{y^*}{n_i}\right)$$

$$+ n_i \left(1 - \frac{x}{n_i}\right) \log \left(1 - \frac{x}{n_i}\right)$$

$$= n_i \left(1 - \frac{x}{n_i}\right) (e^a - a - 1)$$

$$= n_i \left(1 - \frac{x}{n_i}\right) h(a)$$

which proves (33). □

## C.2 COMMON ANALYSIS

### C.2.1 GENERAL REGRET UPPER BOUND

Let $D_t$ be the Bregman divergence induced by $\psi_t$, i.e.,

$$D_t(\boldsymbol{y}, \boldsymbol{x}) = \psi_t(\boldsymbol{y}) - \psi_t(\boldsymbol{x}) - \langle \nabla \psi_t(\boldsymbol{x}), \boldsymbol{y} - \boldsymbol{x} \rangle. \tag{36}$$

Then, the regret for OFTRL is bounded as follows.

**Lemma C.2.** *If $\boldsymbol{x}(t)$ is given by the OFTRL update (2), for any $\boldsymbol{x}^* \in \mathcal{X} \cap \mathbb{R}_+^d$, we have*

$$\sum_{t=1}^{T} \left\langle \hat{\boldsymbol{\ell}}(t), \boldsymbol{x}(t) - \boldsymbol{x}^* \right\rangle \leq \underbrace{\psi_{T+1}\left(\boldsymbol{x}^*\right) - \psi_1(\boldsymbol{y}(1)) + \sum_{t=1}^{T} \left(\psi_t\left(\boldsymbol{y}(t+1)\right) - \psi\left(\boldsymbol{y}(t+1)\right)\right)}_{\text{penalty term}}$$

$$+ \underbrace{\sum_{t=1}^{T} \left( \left\langle \hat{\boldsymbol{\ell}}(t) - \boldsymbol{q}(t), \boldsymbol{x}(t) - \boldsymbol{y}(t+1) \right\rangle - D_t\left(\boldsymbol{y}(t+1), \boldsymbol{x}(t)\right) \right)}_{\text{stability term}}, \quad (37)$$

*where we define $\boldsymbol{y}(t) \in \arg\min_{\boldsymbol{x} \in \mathcal{X}} \left\{ \left\langle \sum_{s=1}^{t-1} \hat{\boldsymbol{\ell}}(s), \boldsymbol{x} \right\rangle + \psi_t(\boldsymbol{x}) \right\}.$*

In the RHS of the above inequality (37), we refer to the sum of the first three terms as the *penalty term* and the remaining term as the *stability term*.

First, we prove the following lemma.

**Lemma C.3.** *The regret of the proposed algorithm is bounded as*

$$R_T \leq \gamma \sum_{i=1}^{d} n_i \mathbb{E}\left[2\beta_i(T+1) - \beta_i(1) + 2\delta_i \log \frac{\beta_i(T+1)}{\beta_i(1)}\right] + dW + 2\sum_{i=1}^{d} \delta_i n_i \delta_i, \quad (38)$$

*where $\delta_i > 0$ is defined by*

$$\delta_i = \frac{1}{3\left(1 - \frac{1}{\beta_i(1)}\right)}$$

*Proof.* Using $\bar{\boldsymbol{x}} \in \mathcal{X}$ such that $\bar{x}_i \geq \frac{n_i}{d}$ for all $i \in [d]$, let

$$\boldsymbol{x}^* = \left(1 - \frac{d}{T}\right) \boldsymbol{a}^* + \frac{d}{T}\bar{\boldsymbol{x}}.$$

Using this and the equality $\mathbb{E}\left[\hat{\boldsymbol{\ell}}(t)|\boldsymbol{x}(t)\right] = \boldsymbol{\ell}$, we have

$$R_T = \mathbb{E}\left[\sum_{t=1}^{T} \left\langle \hat{\boldsymbol{\ell}}(t), \boldsymbol{x}(t) - \boldsymbol{a}^* \right\rangle\right]$$

$$= \mathbb{E}\left[\sum_{t=1}^{T} \left\langle \hat{\boldsymbol{\ell}}(t), \boldsymbol{x}(t) - \boldsymbol{x}^* \right\rangle + \sum_{t=1}^{T} \left\langle \hat{\boldsymbol{\ell}}(t), \boldsymbol{x}^* - \boldsymbol{a}^* \right\rangle\right]$$

$$= \mathbb{E}\left[\sum_{t=1}^{T} \left\langle \hat{\boldsymbol{\ell}}(t), \boldsymbol{x}(t) - \boldsymbol{x}^* \right\rangle + \frac{d}{T} \sum_{t=1}^{T} \left\langle \hat{\boldsymbol{\ell}}(t), \bar{\boldsymbol{x}} - \boldsymbol{a}^* \right\rangle\right]$$

$$\leq \mathbb{E}\left[\langle \hat{\boldsymbol{\ell}}(t), \boldsymbol{x}(t) - \boldsymbol{x}^* \rangle\right] + dW, \quad (39)$$

where in the last inequality, we used $\sum_{t=1}^{T} \langle \hat{\boldsymbol{\ell}}(t), \bar{\boldsymbol{x}} - \boldsymbol{a} \rangle \leq T\|\bar{\boldsymbol{x}} - \boldsymbol{a}^*\|_1 \leq T \sum_{i=1}^{d} n_i = TW$.

The first term in (39) is bounded by (37) in Lemma C.2, the components of which we will bound in the following. We first consider the penalty term. The remaining part of the proof follows a similar argument as that in Ito et al. (2022a) and Tsuchiya et al. (2023), and we include the argument for completeness.

**Bounding the penalty term in (37)** Using the definition of the regularizer $\psi_t(\boldsymbol{x}) = \sum_{i=1}^{d} \beta_i(t) \varphi_i(x_i)$, we have

$$\psi_t(\boldsymbol{x}^*) = \sum_{i=1}^{d} \beta_i(t) \varphi_i(x_i^*)$$

$$\leq \sum_{i=1}^{d} \beta_i(t) \max_{\boldsymbol{x} \in \left[\frac{n_i}{T}, n_i\right]} \varphi_i(\boldsymbol{x})$$

$$\leq \sum_{i=1}^{d} \beta_i(t) \max\left\{\varphi_i\left(\frac{n_i}{T}\right), \varphi_i(n_i)\right\}, \tag{40}$$

where the first inequality follows since the definition of $\boldsymbol{x}^*$ implies $x_i^* \geq \frac{d}{T} \bar{x}_i \geq \frac{n_i}{T}$ for $i \in [d]$ and the second inequality holds since $\varphi_i$ is a convex function. Further, from the definition of $\varphi_i$, we have

$$\max\left\{\varphi_i\left(\frac{n_i}{T}\right), \varphi_i(n_i)\right\} = n_i \cdot \max\left\{\frac{1}{T} - 1 + \log T + \gamma\left(\frac{1}{T} + \left(1 - \frac{1}{T}\right)\log\left(1 - \frac{1}{T}\right)\right), \gamma\right\}$$

$$\leq n_i \cdot \max\left\{\frac{1+\gamma}{T} - 1 + \log T, \gamma\right\}$$

$$= n_i \gamma, \tag{41}$$

where the last inequality follows from $\gamma = \log T$. From this and (40), we have

$$\psi_{T+1}(\boldsymbol{x}^*) \leq \gamma \sum_{i=1}^{d} n_i \beta_i(T+1). \tag{42}$$

Further, as we have $\beta_i(t) \leq \beta_i(t+1)$ from (6) and $\varphi_i(x) \geq 0$ for any $x \in (0, n_i]$, we have

$$-\psi_1(\boldsymbol{y}(1)) + \sum_{t=1}^{T} \left(\psi_t(\boldsymbol{y}(t+1)) - \psi_{t+1}(\boldsymbol{y}(t+1))\right)$$

$$= -\sum_{i=1}^{d} \left(\beta_i(1)\varphi_i(y_i(1)) + \sum_{t=1}^{T} \left(\beta_i(t+1) - \beta_i(t)\right)\varphi_i(y_i(t+1))\right)$$

$$\leq 0. \tag{43}$$

Combining (42) and (43), we can bound the penalty term in (37) as

$$\psi_{T+1}(\boldsymbol{x}^*) - \psi_1(\boldsymbol{y}(1)) + \sum_{t=1}^{T} \left(\psi_t(\boldsymbol{y}(t+1)) - \psi_{t+1}(\boldsymbol{y}(t+1))\right)$$

$$\leq \gamma \sum_{i=1}^{d} n_i \beta_i(T+1). \tag{44}$$

**Bounding the stability term in (37)** The Bregman divergence $D_t(\boldsymbol{x}, \boldsymbol{y})$ is expressed as

$$D_t(\boldsymbol{x}, \boldsymbol{y}) = \sum_{i=1}^{d} \left(\beta_i(t) D_i^{(1)}(x_i, y_i) + \beta_i(t)\gamma D_i^{(2)}(x_i, y_i)\right)$$

$$\geq \sum_{i=1}^{d} \max\left\{\beta_i(t) D_i^{(1)}(x_i, y_i), \beta_i(t)\gamma D_i^{(2)}(x_i, y_i)\right\} \tag{45}$$

where $D_i^{(1)}$ and $D_i^{(2)}$ are Bregman divergence induced by $\varphi_i(x) = -n_i \log\left(\frac{x}{n_i}\right)$ and $\varphi_i^{(2)}(x) = n_i\left(1 - \frac{x}{n_i}\right)\log\left(1 - \frac{x}{n_i}\right)$, respectively. Let $g = x - \log(x+1)$ and $h = \exp(x) - x - 1$. Since,

$\delta_i \geq \frac{1}{3\left(1 - \frac{1}{\beta_i(1)}\right)}$ for all $i \in [d]$, from a simple calculation, we have

$$g(x) = x - \log(x+1) \leq \frac{1}{2}x^2 + \delta_i |x|^3 \quad \left(x \geq -\frac{1}{\beta_i(1)}\right) \tag{46}$$

and

$$h(x) = \exp(x) - x - 1 \leq x^2 \quad (x \leq 1) \tag{47}$$

for all $i \in [d]$. Then, we have

$$\left\langle \hat{\boldsymbol{\ell}}(t) - \boldsymbol{q}(t), \boldsymbol{x}(t) - \boldsymbol{y}(t+1) \right\rangle - D_t(\boldsymbol{y}(t+1), \boldsymbol{x}(t))$$

$$\leq \sum_{i=1}^{d} \left(\hat{l}_i(t) - q_i(t)\right)(x_i(t) - y_i(t+1)) - \beta_i(t) \max\left\{ D_i^{(1)}(y_i(t+1), x_i(t)), \gamma D_i^{(2)}(y_i(t+1), x_i(t)) \right\}$$

$$= \sum_{i=1}^{d} \beta_i(t) \left\{ \frac{\hat{l}_i(t) - q_i(t)}{\beta_i(t)}(x_i(t) - y_i(t+1)) - \max\left\{ D_i^{(1)}(y_i(t+1), x_i(t)), \gamma D_i^{(2)}(y_i(t+1), x_i(t)) \right\} \right\}$$

$$\leq \sum_{i=1}^{d} \beta_i(t) \min\left\{ n_i g_i\left(\frac{\hat{l}_i(t) - q_i(t)}{\beta_i(t)} \frac{x_i(t)}{n_i}\right), \gamma n_i\left(1 - \frac{x}{n_i}\right) h\left(\frac{\hat{l}_i(t) - q_i(t)}{\gamma \beta_i(t)}\right) \right\}, \tag{48}$$

where the last inequality follows from Lemma C.1.

Note that $g(0) = h(0) = 0$ and it holds that

$$\hat{l}_i(t) - q_i(t) = \begin{cases} \frac{a_i(t)}{x_i(t)}(k_i(t) - q_i(t)) & \text{if } a_i(t) \geq 1 \\ 0 & \text{if } a_i = 0 \end{cases}. \tag{49}$$

Therefore, the LHS of (48) is further bounded as

$$\left\langle \hat{\boldsymbol{\ell}}(t) - \boldsymbol{q}(t), \boldsymbol{x}(t) - \boldsymbol{y}(t+1) \right\rangle - D_t(\boldsymbol{y}(t+1), \boldsymbol{x}(t))$$

$$\leq \sum_{i=1}^{d} \beta_i(t) \min\left\{ n_i g\left(\frac{\frac{a_i(t)}{x_i(t)}(k_i(t) - q_i(t))}{\beta_i(t)} \frac{x_i(t)}{n_i}\right), \gamma n_i\left(1 - \frac{x}{n_i}\right) h\left(\frac{\frac{a_i(t)}{x_i(t)}(k_i(t) - q_i(t))}{\gamma \beta_i(t)}\right) \right\}$$

$$\leq \begin{cases} \sum_{i=1}^{d} \frac{1}{n_i}\left(\frac{a_i^2(t)(k_i(t) - q_i(t))^2}{2\beta_i(t)} + \frac{\delta_i a_i^3(t)|k_i(t) - q_i(t)|^3}{n_i \beta_i^2(t)}\right) & \text{if } \gamma \frac{x_i(t)}{a_i(t)} \leq 1 \\[3mm] \sum_{i=1}^{d} \frac{1}{n_i} \min\left\{ \frac{a_i^2(t)(k_i(t) - m_j(t))^2}{2\beta_i(t)} + \frac{\delta_i a_i^3(t)|k_i(t) - q_i(t)|^3}{n_i \beta_i^2(t)}, \frac{\left(1 - \frac{x_i(t)}{n_i}\right) a_i^2(t)(k_i(t) - q_i(t))^2}{\gamma\left(\frac{x_i(t)}{n_i}\right)^2 \beta_i(t)} \right\} & \text{otherwise} \end{cases}$$

$$\leq \sum_{i=1}^{d} \frac{1}{n_i} \min\left\{ \frac{\left(a_i^2(t)(k_i(t) - q_i(t))\right)^2}{2\beta_i(t)} + \delta_i \frac{a_i^3(t)|(k_i(t) - q_i(t))|^3}{n_i \beta_i^2(t)}, \left(1 - \frac{x}{n_i}\right) \frac{a_i^2(t)(k_i(t) - q_i(t))^2}{\gamma \beta_i(t)\left(\frac{x_i(t)}{n_i}\right)^2} \right\}$$

$$\leq \sum_{i=1}^{d} \frac{1}{n_i}\left(\frac{1}{2\beta_i(t)} + \frac{\delta}{n_i \beta_i^2(t)}\right) \cdot a_i^2(t)(k_i(t) - q_i(t))^2 \min\left\{ 1, \frac{2\left(1 - \frac{x_i(t)}{n_i}\right)}{\gamma\left(\frac{x_i(t)}{n_i}\right)^2} \right\}$$

$$= \sum_{i=1}^{d} n_i\left(\frac{1}{2\beta_i(t)} + \frac{1}{\beta_i^2(t)}\right) \alpha_i(t) \tag{50}$$

where the first inequality follows from (48) and (49), the second inequality follows from (46), (47), and the fact that $\left|\frac{(k_i(t) - q_i(t))}{\beta_i(t)}\right| \leq \frac{1}{\beta_i(1)} \leq 1$, and third inequality holds since $\gamma \frac{x_i(t)}{a_i(t)} \leq 1$ means

$\frac{1-\frac{x_i(t)}{n_i}}{\gamma\left(\frac{x_i(t)}{n_i}\right)^2} \geq \frac{1-\frac{x_i(t)}{a_i(t)}}{\gamma\left(\frac{x_i(t)}{a_i(t)}\right)^2} \geq \frac{1-\frac{1}{\gamma}}{\gamma\left(\frac{1}{\gamma}\right)^2} = \gamma - 1 \geq \frac{1}{2} + \delta_i$, which implies

$$\frac{(k_i(t) - q_i(t))^2}{2\beta_i(t)} + \frac{\delta_i a_i(t) |k_i(t) - q_i(t)|^3}{n_i \beta_i^2(t)} \leq \frac{(k_i(t) - q_i(t))^2}{2\beta_i(t)} + \frac{\delta_i |k_i(t) - q_i(t)|^3}{\beta_i(t)}$$

$$= \frac{1}{\beta_i(t)}\left(\frac{1}{2} + \delta_i\right)(k_i(t) - q_i(t))^2$$

$$\leq \frac{1}{\beta_i(t)} \frac{1 - \frac{x_i(t)}{n_i}}{\gamma\left(\frac{x_i(t)}{n_i}\right)^2}(k_i(t) - q_i(t))^2.$$

We hence have

$$\sum_{t=1}^{T}\left(\left\langle\hat{\boldsymbol{\ell}}(t) - \boldsymbol{q}(t), \boldsymbol{x}(t) - \boldsymbol{y}(t+1)\right\rangle - D_t\left(\boldsymbol{y}(t+1), \boldsymbol{x}(t)\right)\right)$$

$$\leq \sum_{i=1}^{d} n_i \sum_{t=1}^{T}\left(\frac{1}{2\beta_i(t)} + \frac{\delta_i}{\beta_i^2(t)}\right)\alpha_i(t). \tag{51}$$

We can show that a part of (51) is bounded as

$$\sum_{t=1}^{T}\frac{\alpha_i(t)}{2\beta_i(t)}$$

$$\leq \gamma\left(\sqrt{\beta_i^2(1) - \frac{1}{\gamma} + \frac{1}{\gamma}\sum_{t=1}^{T}\alpha_i(t)} - \sqrt{\beta_i^2(1) - \frac{1}{\gamma}}\right)$$

$$\leq \gamma\left(\beta_i\left(T+1\right) - \beta_i(1)\right). \tag{52}$$

The first inequality in (52) holds since

$$\sqrt{\beta_i^2(1) - \frac{1}{\gamma} + \frac{1}{\gamma}\sum_{t=1}^{t}\alpha_i(t)} - \sqrt{\beta_i^2(1) - \frac{1}{\gamma} + \frac{1}{\gamma}\sum_{t=1}^{t-1}\alpha_i(t)}$$

$$= \frac{1}{\gamma}\cdot\frac{\alpha_i(t)}{\sqrt{\beta_i^2(1) - \frac{1}{\gamma} + \frac{1}{\gamma}\sum_{s=1}^{t}\alpha_i(s)} + \sqrt{\beta_i^2(1) - \frac{1}{\gamma} + \gamma\sum_{s=1}^{t-1}\alpha_i(s)}} \tag{53}$$

$$\geq \frac{\alpha_i(t)}{2\gamma\sqrt{\beta_i^2(1) + \frac{1}{\gamma}\sum_{s=1}^{t-1}\alpha_i(s)}} \tag{54}$$

$$= \frac{\alpha_i(t)}{2\gamma\beta_i(t)}, \tag{55}$$

where the inequality follows by $\alpha_i(t) \leq 1$. The second inequality in (52) follows since

$$\sqrt{\beta_i^2(1) - \frac{1}{\gamma} + \frac{1}{\gamma}\sum_{t=1}^{T}\alpha_i(t)} - \sqrt{\beta_i^2(1) - \frac{1}{\gamma}} \tag{56}$$

$$\leq \sqrt{\beta_i^2(1) - \frac{1}{\gamma} + \frac{1}{\gamma}\sum_{t=1}^{T}\alpha_i(t) - \beta_i(1) + \frac{1}{\gamma}} \tag{57}$$

$$\leq \beta_i(T+1) - \beta_i(1) + \frac{1}{\gamma}, \tag{58}$$

where the first inequality follows from $\sqrt{x} - \sqrt{x-y} \leq \frac{y}{\sqrt{x}}$ for $x \geq y \geq 0$ and $\beta_i(1) \geq 1$.

Similarly, we can show

$$\sum_{t=1}^{T} \frac{\alpha_i(t)}{\beta_i^2(t)} = \sum_{t=1}^{T} \frac{\alpha_i(t)}{\beta_i^2(1) + \frac{1}{\gamma} \sum_{s=1}^{t-1} \alpha_i(s)}$$

$$= \gamma \sum_{t=1}^{T} \frac{\alpha_i(t)}{\gamma \beta_i^2(1) + \sum_{s=1}^{t-1} \alpha_i(s)}$$

$$\leq \gamma \log \left( 1 + \frac{1}{\gamma \beta_i^2(1) - 1} \sum_{t=1}^{T} \alpha_i(t) \right) \tag{59}$$

$$\leq 2\gamma \log \frac{\beta_i(T+1)}{\beta_i(1)} + 2. \tag{60}$$

The first inequality in (60) follows since

$$\log \left( 1 + \frac{1}{\gamma \beta_i^2(1) - 1} \sum_{s=1}^{t} \alpha_i(s) \right) - \log \left( 1 + \frac{1}{\gamma \beta_i^2(1) - 1} \sum_{s=1}^{t-1} \alpha_i(s) \right)$$

$$= -\log \left( 1 - \frac{\alpha_i(t)}{\gamma \beta_i^2(1) - 1 + \sum_{s=1}^{t} \alpha_i(s)} \right) \tag{61}$$

$$\geq -\log \left( 1 - \frac{\alpha_i(t)}{\gamma \beta_i^2(1) + \sum_{s=1}^{t-1} \alpha_i(s)} \right) \tag{62}$$

$$\geq \frac{\alpha_i(t)}{\gamma \beta_i^2(1) + \sum_{s=1}^{t-1} \alpha_i(s)}, \tag{63}$$

where the first inequality follows from $\alpha_i(t) \leq 1$ and the last inequality follows from $-\log(1-x) \geq x$ for $x < 1$. The second inequality in (60) follows from

$$\log \left( 1 + \frac{1}{\gamma \beta_i^2(1) - 1} \sum_{t=1}^{T} \alpha_i(t) \right)$$

$$< \log \left( 1 + \frac{1}{\gamma \beta_i^2(1)} \sum_{t=1}^{T} \alpha_i(t) \right) + \log \frac{\gamma \beta_i^2(1)}{\gamma \beta_i^2(1) - 1} \tag{64}$$

$$= \log \left( \frac{\beta_i(T+1)}{\beta_i(1)} \right) + \log \left( 1 + \frac{1}{\gamma \beta_i^2(1) - 1} \right) \tag{65}$$

$$\leq 2 \log \frac{\beta_i(T+1)}{\beta_i(1)} + \frac{2}{\gamma} \tag{66}$$

where the last inequality follows from $\log(1 + \frac{1}{x-1}) \geq \frac{2}{x}$ for $x \geq 3/2$. Bounding the RHS of (50) with (52) and (60) yields

$$\sum_{t=1}^{T} \left\langle \hat{\boldsymbol{\ell}}(t) - \boldsymbol{q}(t), \boldsymbol{x}(t) - \boldsymbol{y}(t+1) \right\rangle - D_t \left( \boldsymbol{y}(t+1), \boldsymbol{x}(t) \right)$$

$$\leq \gamma \sum_{i=1}^{d} n_i \left( \beta_i(T+1) - \beta_i(1) + 2\delta_i \log \frac{\beta_i(T+1)}{\beta_1} \right) + 2 \sum_{i=1}^{d} n_i \delta_i \tag{67}$$

Finally, by bounding the RHS of (37) and sequentially using (39), (44) and (67), we have

$$R_T \leq \gamma \sum_{i=1}^{d} n_i \mathbb{E}\left[2\beta_i(T+1) - \beta_i(1) + 2\delta_i \log \frac{\beta_i(T+1)}{\beta_i(1)}\right] + dW + 2\sum_{i=1}^{d} n_i \delta_i, \qquad (68)$$

which completes the proof. $\qquad \square$

### C.2.2 A *Lower Bound*

Below, we define

$$\Delta'_{i,\min} = \min_{\boldsymbol{a} \in \mathcal{A} \setminus \{\boldsymbol{a}^*\}} \left\{ \boldsymbol{a}^\top \boldsymbol{\ell} - \boldsymbol{a}^{*\top} \boldsymbol{\ell} : a_i = 0 \right\}. \qquad (69)$$

To obtain the regret upper bound depending on $\Delta_i$ in the stochastic regime and the stochastic regime with adversarial corruptions, we prove the following regret *lower bound*.

**Lemma C.4.** *In the stochastic regime with adversarial corruptions, for any algorithm and any action set $\mathcal{A}$, the regret is bounded as*

$$R_T \geq \mathbb{E}\left[\sum_{t=1}^{T} \left(\frac{1}{\lambda'_{\mathcal{A}}} \sum_{i \in I^*} \Delta'_{i,\min} (a_i^* - a_i(t)) + \frac{1}{\lambda_{\mathcal{A}}} \sum_{i \in J^*} \Delta_{i,\min} a_i(t)\right)\right] - 2CM, \qquad (70)$$

*where $\lambda'_{\mathcal{A}} = \min\{W_{I^*}, W - M\}$.*

*Proof.* We can bound the regret as

$$R_T$$
$$= \mathbb{E}\left[\sum_{t=1}^{T}\left(\sum_{i=1}^{d}\sum_{j=1}^{a_i(t)} L_{i,j}(t) - \sum_{i=1}^{d}\sum_{j=1}^{a_i^*(t)} L_{i,j}(t)\right)\right]$$
$$= \mathbb{E}\left[\sum_{t=1}^{T}\left(\sum_{i=1}^{d}\sum_{j=1}^{a_i(t)} L'_{i,j}(t) - \sum_{i=1}^{d}\sum_{j=1}^{a_i^*(t)} L'_{i,j}(t)\right) + \sum_{t=1}^{T}\left(\sum_{i=1}^{d}\sum_{j=1}^{a_i(t)} L'_{i,j}(t) - \sum_{i=1}^{d}\sum_{j=1}^{a_i^*(t)} L'_{i,j}(t)\right)\right]$$
$$\geq \mathbb{E}\left[\sum_{t=1}^{T}\left(\sum_{i=1}^{d}\sum_{j=1}^{a_i(t)} L'_{i,j}(t) - \sum_{i=1}^{d}\sum_{j=1}^{a_i^*(t)} L'_{i,j}(t)\right)\right] - \sum_{t=1}^{T}\left|\max_{i\in[d], j\in[n_i]} L_{i,j}(t) - L'_{i,j}(t)\right| \|\boldsymbol{a}(t) - \boldsymbol{a}^*\|_1$$
$$\geq \mathbb{E}\left[\sum_{t=1}^{T}\left(\sum_{i=1}^{d}\sum_{j=1}^{a_i(t)} L'_{i,j}(t) - \sum_{i=1}^{d}\sum_{j=1}^{a_i^*(t)} L'_{i,j}(t)\right)\right] - 2MC, \qquad (71)$$

where the first inequality follows from the Hölder's inequality, the second inequality follows since $\|\boldsymbol{a}(t) - \boldsymbol{a}^*\|_1 \leq 2M$, and the last inequality follows from the definition of $C = \mathbb{E}\left[\sum_{t=1}^{T} \max_{i\in[d]} \max_{j\in[n_i]} |L_{i,j}(t) - L'_{i,j}(t)|\right] \geq 0$. We then bound $\mathbb{E}\left[\sum_{t=1}^{T}\left(\sum_{i=1}^{d}\sum_{j=1}^{a_i(t)} L'_{i,j}(t) - \sum_{i=1}^{d}\sum_{j=1}^{a_i^*(t)} L'_{i,j}(t)\right)\right]$.

Below, we write $\langle \boldsymbol{L}, \boldsymbol{a} \rangle = \sum_{i=1}^{d}\sum_{j=1}^{a_i} L_{i,j}(t)$. We have $\langle \boldsymbol{L}, \boldsymbol{a} \rangle - \langle \boldsymbol{L}, \boldsymbol{a}' \rangle = \langle \boldsymbol{L}, \boldsymbol{a} - \boldsymbol{a}' \rangle$.

We consider the case of general action sets and recall that $I^* := \{i \in [d] : a_i^* \geq 1\}$ and $J^* = [d] \setminus I^*$. Since $\sum_{i \in I^*} (a_i^* - a_i(t)) \leq M^*$ and $\sum_{i \in J^*} a_i(t) \leq M$, we have

$$\langle \boldsymbol{L}, \boldsymbol{a}(t) - \boldsymbol{a}^* \rangle$$

$$= \frac{1}{2} \langle \boldsymbol{L}, \boldsymbol{a}(t) - \boldsymbol{a}^* \rangle + \frac{1}{2} \langle \boldsymbol{L}, \boldsymbol{a}(t) - \boldsymbol{a}^* \rangle$$

$$\geq \frac{1}{2 \min \{W_{I^*}, W - M\}} \sum_{i \in I^*} (n_i - a_i(t)) \langle \boldsymbol{L}, \boldsymbol{a}(t) - \boldsymbol{a}^* \rangle$$

$$+ \frac{1}{2 \min \{W_{J^*}, M\}} \sum_{i \in J^*} a_i(t) \langle \boldsymbol{L}, \boldsymbol{a}(t) - \boldsymbol{a}^* \rangle$$

$$\geq \frac{1}{2 \min \{W_{I^*}, W - M\}} \sum_{i \in I^*} \Delta'_{i,\min} (n_i - a_i(t)) + \frac{1}{2 \min \{W_{J^*}, M\}} \sum_{i \in J^*} \Delta_{i,\min} a_i(t).$$

Combining this inequality with (71) completes the proof. $\qquad \square$

Note that in the stochastic regime with adversarial corruptions, from Lemma C.4, it holds that

$$R_T \geq \mathbb{E} \left[ \sum_{t=1}^T \left( \frac{1}{2M^*} \sum_{i \in I^*} \Delta'_{i,\min} (a_i^* - a_i(t)) + \frac{1}{2M} \sum_{i \in J^*} \Delta_{i,\min} a_i(t) \right) \right] - 2CM$$

$$= \frac{1}{2M^*} \sum_{i \in I^*} \Delta'_{i,\min} Q_i + \frac{1}{2M} \sum_{i \in J^*} \Delta_{i,\min} P_i - 2CM, \tag{72}$$

where equality follows from the law of iterated expectations.

## C.3 PROOF FOR THE LS METHOD

In this section, we provide proof for the results of the LS Method.

### C.3.1 PRELIMINARIES

We use the following lemma to bound $\sum_{t=1}^T \alpha_i(t)$ for suboptimal arms $i \in J^*$.

**Lemma C.5.** *It holds for any $i \in [d]$ and $q_i^* \in [0, 1]$ that*

$$\sum_{t=1}^T \alpha_i(t) \leq \sum_{t=1}^T \left( \frac{a_i(t)}{n_i} \right)^2 (k_i(t) - q_i(t))^2$$

$$\leq \sum_{t=1}^T \left( \frac{a_i(t)}{n_i} \right)^2 (k_i(t) - m^*)^2 + \log \left( 1 + \sum_{t=1}^T a_i(t) \right) + \frac{5}{4} \tag{73}$$

To prove this lemma, we use the following lemma.

**Lemma C.6.** *Suppose $k_i(s) \in [0, 1]$ for any $s \in [t]$, and define $q_i(t) \in [0, 1]$ by*

$$q_i(t) = \frac{1}{1 + \sum\limits_{s=1}^{t-1} a_i(s)} \left( \frac{1}{2} + \sum_{s=1}^{t-1} a_i(s) k_i(s) \right). \tag{74}$$

*We then have*

$$\sum_{t=1}^T a_i(t) \left( (k_i(t) - q_i(t))^2 - (k_i(t) - m^*)^2 \right) \leq \frac{5}{4} + \log \left( 1 + \sum_{t=1}^T a_i(t) \right) \tag{75}$$

*for any $m^* \in [0, 1]$.*

*Proof.* From the definition of $q_i(t)$, $q_i(t)$ is expressed as

$$q_i(t) \in \arg\min_{m \in \mathbb{R}} \left\{ \left( m - \frac{1}{2} \right)^2 + \sum_{s=1}^{t-1} a_i(s) \left( m - k_i(s) \right)^2 \right\}, \tag{76}$$

which implies

$$q_i(t) - \frac{1}{2} + \sum_{s=1}^{t-1} a_i(s) \left( q_i(t) - k_i(s) \right) = 0. \tag{77}$$

We have

$$\left( m - \frac{1}{2} \right)^2 + \sum_{s=1}^{t-1} a_i(s) \left( m - k_i(s) \right)^2$$

$$= \left( m - q_i(t) + q_i(t) - \frac{1}{2} \right)^2 + \sum_{s=1}^{t-1} a_i(s) \left( m - q_i(t) + q_i(t) - k_i(s) \right)^2$$

$$= \left( m - q_i(t) \right)^2 + 2 \left( m - q_i(t) \right) \left( q_i(t) - \frac{1}{2} \right) + \left( q_i(t) - \frac{1}{2} \right)^2 + \sum_{s=1}^{t-1} a_i(s) \left( m - q_i(t) \right)^2$$

$$+ 2 \left( m - q_i(t) \right) \sum_{s=1}^{t-1} a_i(s) \left( q_i(t) - k_i(s) \right) + \sum_{s=1}^{t-1} a_i(s) \left( q_i(t) - k_i(s) \right)^2$$

$$= \left( q_i(t) - \frac{1}{2} \right)^2 + \left( \sum_{s=1}^{t-1} a_i(s) + 1 \right) \left( m - q_i(t) \right)^2 + \sum_{s=1}^{t-1} a_i(s) \left( q_i(t) - k_i(s) \right)^2 \tag{78}$$

for any $m \in \mathbb{R}$. The third equality follows from (77). Using this, for any $m^*$, we obtain

$$
\left(m^* - \frac{1}{2}\right)^2 + \sum_{t=1}^{T} a_i(t) \left(k_i(t) - m^*\right)^2
$$

$$
= \left(q_i(T+1) - \frac{1}{2}\right)^2 + \left(\sum_{t=1}^{T} a_i(t) + 1\right) \left(m^* - q_i(T+1)\right)^2 + \sum_{t=1}^{T} a_i(t) \left(q_i(T+1) - k_i(t)\right)^2
$$

$$
\geq \left(q_i(T+1) - \frac{1}{2}\right)^2 + \sum_{t=1}^{T} a_i(t) \left(q_i(T+1) - k_i(t)\right)^2
$$

$$
= \left(q_i(T+1) - \frac{1}{2}\right)^2 + \sum_{t=1}^{T-1} a_i(t) \left(q_i(T+1) - k_i(t)\right)^2 + a_i(T) \left(q_i(T+1) - k_i(T)\right)^2
$$

$$
= \left(q_i(T+1) - q_i(T) + q_i(T) - \frac{1}{2}\right)^2 + \sum_{t=1}^{T-1} a_i(t) \left(q_i(T+1) - q_i(T) + q_i(T) - k_i(t)\right)^2
$$

$$
\quad + a_i(T) \left(q_i(T+1) - k_i(T)\right)^2
$$

$$
= \left(q_i(T+1) - q_i(T)\right)^2 + 2 \left(q_i(T+1) - q_i(T)\right)\left(q_i(T) - \frac{1}{2}\right) + \left(q_i(T) - \frac{1}{2}\right)^2
$$

$$
\quad + \sum_{t=1}^{T-1} a_i(t) \left(q_i(T+1) - q_i(T)\right)^2 + 2 \left(q_i(T+1) - q_i(T)\right) \sum_{t=1}^{T-1} a_i(t) \left(q_i(t) - l_{ij}(t)\right)
$$

$$
\quad + \sum_{t=1}^{T-1} a_i(t) \left(q_i(T) - k_i(t)\right)^2 + a_i(T) \left(q_i(T+1) - k_i(T)\right)^2
$$

$$
= \left(q_i(T) - \frac{1}{2}\right)^2 + \left(\sum_{t=1}^{T-1} a_i(t) + 1\right) \left(q_i(T+1) - q_i(T)\right)^2 + \sum_{t=1}^{T-1} a_i(t) \left(q_i(T) - k_i(t)\right)^2
$$

$$
\quad + a_i(T) \left(q_i(T+1) - k_i(T)\right)^2
$$

$$
= \left(q_i(1) - \frac{1}{2}\right)^2 + \sum_{t=1}^{T} a_i(t) \left(k_i(t) - q_i(t+1)\right)^2 + \sum_{t=1}^{T} \left(1 + \sum_{s=1}^{t-1} a_i(s)\right) \left(q_i(t+1) - q_i(t)\right)
$$

$$
\tag{79}
$$

where the first and fifth inequalities follow from (78), and the last equality can be shown by repeating the same transformation $T$ times. Hence, for any $m^* \in \mathbb{R}$, we have

$$\sum_{t=1}^{T} \left(\frac{a_i(t)}{n_i}\right)^2 \left((q_i(t) - k_i(t))^2 - (k_i(t) - m^*)^2\right)$$

$$\leq \frac{1}{n_i} \sum_{t=1}^{T} a_i(t) \left((q_i(t) - k_i(t))^2 - (k_i(t) - m^*)^2\right)$$

$$\leq \frac{1}{n_i} \sum_{t=1}^{T} a_i(t) (q_i(t) - k_i(t))^2$$

$$- \frac{1}{n_i} \left(\sum_{t=1}^{T} a_i(t) (q_i(t+1) - k_i(t))^2 + \sum_{t=1}^{T} \left(\sum_{s=1}^{t-1} a_i(s) + 1\right) (q_i(t+1) - q_i(t))^2\right)$$

$$+ \frac{1}{n_i} \left(m^* - \frac{1}{2}\right)^2$$

$$= \frac{1}{n_i} \sum_{t=1}^{T} \left(a_i(t) (q_i(t) - k_i(t))^2 - a_i(t) (q_i(t+1) - k_i(t))^2 + \left(\sum_{s=1}^{t-1} a_i(s) + 1\right) (q_i(t+1) - q_i(t))^2\right)$$

$$+ \frac{1}{n_i} \left(m^* - \frac{1}{2}\right)^2$$

$$\leq \frac{1}{n_i} \sum_{t=1}^{T} \left(a_i(t) (2k_i(t) - q_i(t) - q_i(t+1)) (q_i(t+1) - q_i(t)) - \left(\sum_{s=1}^{t-1} a_i(s) + 1\right) (q_i(t+1) - q_i(t))\right)$$

$$+ \frac{1}{n_i} \left(m^* - \frac{1}{2}\right)^2$$

$$\leq \frac{1}{n_i} \sum_{t=1}^{T} \frac{a_i^2(t)}{4 \left(1 + \sum\limits_{s=1}^{t-1} a_i(s)\right)} (2k_i(t) - q_i(t) - q_i(t+1))^2 + \frac{1}{n_i} \left(m^* - \frac{1}{2}\right)^2$$

$$\leq \sum_{t=1}^{T} \frac{a_i(t)}{\left(1 + \sum\limits_{s=1}^{t-1} a_i(s)\right)} + \left(m^* - \frac{1}{2}\right)^2$$

$$\leq \log\left(1 + \sum_{t=1}^{T} a_i(t)\right) + \frac{5}{4} \tag{80}$$

where the second equality follows from $ax - tx^2 = \frac{a^2}{4t} - \left(\frac{a}{2\sqrt{t}} - \sqrt{t}x\right)^2 \leq \frac{a^2}{4t}$ that holds for any $a, x \in \mathbb{R}$, and the forth inequality holds since $|2k_i(t) - m(t) - m(t+1)| \leq 2$, which follows from $k_i(t), m(t) \in [0, 1]$. $\qquad\square$

*Proof of Lemma C.5.*

$$\sum_{t=1}^{T} \alpha_i(t)$$

$$= \sum_{t=1}^{T} \left(\frac{a_i(t)}{n_i}\right)^2 \left(k_i(t) - q_i(t)\right)^2 \cdot \min\left\{1, \frac{2\left(1 - \frac{x_i(t)}{n_i}\right)}{\left(\frac{x_i(t)}{n_i}\right)^2 \gamma}\right\}$$

$$\leq \sum_{t=1}^{T} \left(\frac{a_i(t)}{n_i}\right)^2 \left(k_i(t) - q_i(t)\right)^2$$

$$\leq \sum_{t=1}^{T} \left(\frac{a_i(t)}{n_i}\right)^2 \left(k_i(t) - m^*\right)^2 + \log\left(1 + \sum_{t=1}^{T} a_i(t)\right) + \frac{5}{4} \tag{81}$$

where the second inequality follows from Lemma C.6. $\qquad\square$

From Lemma C.5, in the stochastic regime, it holds that

$$\mathbb{E}\left[\sum_{t=1}^{T} \alpha_i(t)\right]$$

$$\leq \mathbb{E}\left[\sum_{t=1}^{T} \frac{a_i^2(t)}{n_i^2} \cdot \frac{\sigma_i^2}{a_i(t)} + \log\left(1 + \sum_{t=1}^{T} a_i(t)\right)\right] + \frac{5}{4}$$

$$\leq \frac{\sigma_i^2}{n_i^2} P_i + \log\left(1 + P_i\right) + \frac{5}{4}, \tag{82}$$

where the first inequality follows from Lemma C.5 with $m_i^* = \mu_i$ and in the last inequality, we define

$$P_i = \mathbb{E}\left[\sum_{t=1}^{T} a_i(t)\right] = \mathbb{E}\left[\sum_{t=1}^{T} x_i(t)\right]. \tag{83}$$

We give a bound on $\sum_{t=1}^{T} \alpha_i(t)$ using the following lemma.

**Lemma C.7.** *It holds for any $i \in [d]$ that*

$$\mathbb{E}\left[\alpha_i(t)\right] \leq 2\mathbb{E}\left[\min\left\{x_i(t), \frac{n_i - x_i(t)}{\sqrt{\gamma}}\right\}\right]$$

$$\leq 2\left(\frac{\sigma_i}{n_i}\right)^2 \mathbb{E}\left[\frac{n_i - x_i(t)}{\sqrt{\gamma}}\right]. \tag{84}$$

*Proof.* From the definition of $\alpha_i(t)$, we have

$$\mathbb{E}\left[\alpha_i(t)|x_i(t)\right]$$

$$=\mathbb{E}\left[\left(\frac{a_i(t)}{n_i}\right)^2 (k_i(t)-q_i(t))^2 \cdot \min\left\{1, \frac{2\left(1-\frac{x_i(t)}{n_i}\right)}{\gamma\left(\frac{x_i(t)}{n_i}\right)^2}\right\}\bigg|x_i(t)\right]$$

$$\leq\mathbb{E}\left[\left(\frac{a_i(t)}{n_i}\right)^2 \cdot \frac{\sigma_i^2}{a_i(t)}\bigg|x_i(t)\right]\min\left\{1, \frac{2\left(1-\frac{x_i(t)}{n_i}\right)}{\gamma\left(\frac{x_i(t)}{n_i}\right)^2}\right\}$$

$$\leq\left(\frac{\sigma_i}{n_i}\right)^2 \min\left\{x_i(t), \frac{2x_i(t)\left(1-\frac{x_i(t)}{n_i}\right)}{\gamma\left(\frac{x_i(t)}{n_i}\right)^2}\right\}$$

$$=\left(\frac{\sigma_i}{n_i}\right)^2 n_i \min\left\{\frac{x_i(t)}{n_i}, \frac{2\left(1-\frac{x_i(t)}{n_i}\right)}{\gamma\frac{x_i(t)}{n_i}}\right\}$$

$$\leq\begin{cases}\frac{\sigma_i^2}{n_i}\frac{x_i(t)}{n_i} & \text{if } \frac{x_i(t)}{n_i} < \frac{1}{\sqrt{\gamma}} \\ \frac{\sigma_i^2}{n_i}\frac{2\left(1-\frac{x_i(t)}{n_i}\right)}{\sqrt{\gamma}} & \text{if } \frac{x_i(t)}{n_i} \geq \frac{1}{\sqrt{\gamma}}\end{cases}$$

$$\leq\frac{\sigma_i^2}{n_i}\frac{2\left(1-\frac{x_i(t)}{n_i}\right)}{\sqrt{\gamma}}$$

$$=\left(\frac{\sigma_i}{n_i}\right)^2 \frac{2}{\sqrt{\gamma}}\left(n_i - x_i(t)\right), \tag{85}$$

where the first inequality follows from the condition of $k_i(t), q_i(t) \in [0,1]$ and the last inequality is due to $\sqrt{\gamma} \geq 2$ that follows from the assumption $T \geq 55 \geq e^4$. $\qquad\square$

### C.3.2 PROOF OF THE STOCHASTIC REGIME

**Proof for the Stochastic Regime.** We call base arms in $I^*$ *optimal arms* and $J^*$ *suboptimal arms*. We bound the RHS of (38) separately considering sub-optimal and optimal base arms.

**Sub-optimal base arms side** From (82), the component of the RHS of (38) is bounded by

$$\mathbb{E}\left[2\beta_i(T+1) - \beta_i(1) + 2\delta_i \log\left(\frac{\beta_i(T+1)}{\beta_i(1)}\right)\right]$$

$$=\mathbb{E}\left[2\sqrt{\beta_i(1)^2 + \frac{1}{\gamma}\sum_{t=1}^{T}\alpha_i(t)} - \beta_i(1) + \delta_i \log\left(1 + \frac{1}{\gamma\beta_i(1)^2}\sum_{t=1}^{T}\alpha_i(t)\right)\right]$$

$$\leq 2\sqrt{\beta_i(1)^2 + \frac{1}{\gamma}\left(\frac{\sigma_i^2}{n_i^2}P_i + \log(1+P_i) + \frac{5}{4}\right)} - \beta_i(1)$$

$$\quad + \delta_i \log\left(1 + \frac{1}{\gamma\beta_i(1)^2}\left(\frac{\sigma_i^2}{n_i^2}P_i + \log(1+P_i) + \frac{5}{4}\right)\right)$$

$$\leq 2\sqrt{\beta_i(1)^2 + \frac{\sigma_i^2}{n_i^2}\frac{P_i}{\gamma} + \frac{1}{\gamma\beta_i(1)}\left(\log(1+P_i) + \frac{5}{4}\right)} - \beta_i(1) + \delta_i \log\left(1 + \frac{\sigma_i^2}{n_i^2}\frac{P_i}{\gamma\beta_i(1)^2}\right)$$

$$\quad + \frac{\delta}{\gamma\beta_i(1)^2}\left(\log(1+P_i) + \frac{5}{4}\right)$$

$$=2\sqrt{\beta_i(1)^2 + \frac{\sigma_i^2}{n_i^2}\frac{P_i}{\gamma}} - \beta_i(1) + \delta_i \log\left(1 + \frac{\sigma_i^2}{n_i^2}\frac{P_i}{\gamma\beta_i(1)^2}\right) + \frac{\xi_i}{\gamma}\left(\log(1+P_i) + \frac{5}{4}\right), \tag{86}$$

where the first inequality follows from (82), the second inequality follows from $\sqrt{x+y} \leq \sqrt{x} + \frac{y}{2\sqrt{x}}$ that holds for any $x > 0$ and $y > 0$, $\log(1 + x + y) \leq \log(1 + x) + y$ that holds for any $x, y \geq 0$, and in the last equality we define $\xi_i = \frac{1}{\beta_i(1)} + \frac{\delta_i}{\beta_i(1)^2}$.

**Optimal base-arm side** Next, we let $i \in I^*$ be an optimal base-arm. We define the complement version of $P_i$ by

$$Q_i = \mathbb{E}\left[\sum_{t=1}^{T}(n_i - x_i(t))\right] \tag{87}$$

for $i \in [d]$. Then, from Lemma C.7, we have

$$\mathbb{E}\left[2\beta_i(T+1) - \beta_i(1) + 2\delta_i \log \frac{\beta_i(T+1)}{\beta_i(1)}\right]$$

$$= \mathbb{E}\left[2\sqrt{\beta_i(1)^2 + \frac{1}{\gamma}\sum_{t=1}^{T}\alpha_i(t)} - \beta_i(1) + \delta_i \log\left(1 + \frac{1}{\gamma\beta_i(1)^2}\sum_{t=1}^{T}\alpha_i(t)\right)\right]$$

$$\leq \mathbb{E}\left[2\sqrt{\beta_i(1)^2 + \frac{1}{\gamma}\sum_{t=1}^{T}\alpha_i(t)} - \beta_i(1) + 2\delta_i\left(\sqrt{1 + \frac{1}{\gamma\beta_i(1)^2}\sum_{t=1}^{T}\alpha_i(t)} - 1\right)\right]$$

$$= 2(\beta_i(1) + \delta_i)\mathbb{E}\left[\sqrt{1 + \frac{1}{\gamma\beta_i(1)^2}\sum_{t=1}^{T}\alpha_i(t)} - 1\right] + \beta_i(1)$$

$$\leq 2(\beta_i(1) + \delta_i) \cdot \left(\sqrt{1 + \frac{2}{\gamma^{\frac{3}{2}}\beta_i(1)^2}\left(\frac{\sigma_i}{n_i}\right)^2\sum_{t=1}^{T}\mathbb{E}\left[(n_i - x_i(t))\right]} - 1\right) + \beta_i(1)$$

$$\leq 2(\beta_i(1) + \delta_i)\sqrt{\mathbb{E}\left[\frac{2}{\gamma^{\frac{3}{2}}\beta_i(1)^2}\left(\frac{\sigma_i}{n_i}\right)^2\sum_{t=1}^{T}\mathbb{E}\left[n_i - x_i(t)\right]\right]} + \beta_i(1)$$

$$\leq 2(1 + \delta_i)\frac{\sigma_i}{n_i}\sqrt{\frac{2}{\gamma^{\frac{3}{2}}}Q_i} + \beta_i(1) \tag{88}$$

where the first inequality follows from the inequality of $\log(1 + x) \leq 2\left(\sqrt{1 + x} - 1\right)$ for $x > 0$, the second inequality follows from Lemma C.7, the third inequality follows from $\sqrt{1 + x} - 1 \leq \sqrt{x}$ for $x \geq 0$, and the last inequality follows from $\beta_i(1) \geq 1$ for any $i \in [d]$.

**Putting together the upper bound and lower bounds and applying a self-bounding technique** Bounding the RHS of (38) using (86) and (88) yields the regret upper bound depending on $(P_i)_{i \in J^*}$

and $(Q_i)_{i \in I^*}$ as

$$
\begin{aligned}
\frac{R_T}{\gamma} \\
\leq \sum_{i \in J^*} n_i &\left( \sqrt{\beta_i(1)^2 + \frac{\sigma_i^2}{n_i^2}\frac{P_i}{\gamma}} - \beta_i(1) + \delta_i \log\left(1 + \frac{\sigma_i^2}{n_i^2}\frac{P_i}{\gamma \beta_i(1)^2}\right) + \frac{\xi_i}{\gamma}\left(\log(1 + P_i) + \frac{5}{4}\right)\right) \\
&+ \sum_{i \in I^*} n_i \left(2(1 + \delta_i)\frac{\sigma_i}{n_i}\sqrt{\frac{2}{\gamma^{\frac{3}{2}}}Q_i} + \beta_i(1)\right) + \frac{dW + 2\sum\limits_{i=1}^{d}\delta_i n_i}{\gamma} \\
= \sum_{i \in J^*} &\left(\sqrt{(n_i\beta_i(1))^2 + \sigma_i^2\frac{P_i}{\gamma}} - n_i\beta_i(1) + n_i\delta_i\log\left(1 + \frac{\sigma_i^2}{n_i^2}\frac{P_i}{\gamma\beta_i(1)^2}\right) + \frac{n_i\xi_i}{\gamma}\left(\log(1 + P_i) + \frac{5}{4}\right)\right) \\
&\hspace{11cm}(89) \\
&+ \sum_{i \in I^*}\left(2(1 + \delta_i)\sigma_i\sqrt{\frac{2}{\gamma^{\frac{3}{2}}}Q_i} + n_i\beta_i(1)\right) + \frac{dW + 2\sum\limits_{i=1}^{d}\delta_i n_i}{\gamma} \\
= \sum_{i \in J^*} &\bar{f}_i\left(\frac{P_i}{\gamma}\right) + 2\sum_{i \in I^*}(1 + \delta_i)\sigma_i\sqrt{\frac{2}{\gamma^{\frac{3}{2}}}Q_i} + \sum_{i \in I^*}n_i\beta_i(1) + \frac{1}{\gamma}\left(dW + 2\sum_{i=1}^{d}\delta_i n_i + \frac{5}{4}\sum_{i \in J^*}n_i\xi_i\right), \\
&\hspace{11cm}(90)
\end{aligned}
$$

where we define convex function $\bar{f}_i : \mathbb{R} \to \mathbb{R}$ by

$$
\bar{f}_i(x) = 2\sqrt{(n_i\beta_i(1))^2 + \sigma_i^2 x} + n_i\delta_i\log\left(1 + \frac{\sigma_i^2 x}{\gamma(n_i\beta_i(1))^2}\right) + \frac{n_i\xi_i}{\gamma}\log(1 + \gamma x) - n_i\beta_i(1). \tag{91}
$$

In the stochastic regime, setting $C = 0$ in (72) yields the regret lower bound depending on $(P_i)_{i \in J^*}$ and $(Q_i)_{i \in I^*}$ as

$$
R_T \geq \frac{1}{\lambda'_{\mathcal{A}}}\sum_{i \in I^*}\Delta'_{i,\min}Q_i + \frac{1}{\lambda_{\mathcal{A}}}\sum_{i \in J^*}\Delta_{i,\min}P_i. \tag{92}
$$

Combining (90) and (92), we have

$$\frac{R_T}{\log T} = \frac{R_T}{\gamma} = 2\frac{R_T}{\gamma} - \frac{R_T}{\gamma}$$

$$\leq 2\frac{R_T}{\gamma} - \frac{1}{\gamma}\left(\frac{1}{\lambda'_{\mathcal{A}}}\sum_{i\in I^*}\Delta_{i,\min}Q_i + \frac{1}{\lambda_{\mathcal{A}}}\sum_{i\in J^*}\Delta_{i,\min}P_i\right)$$

$$\leq \sum_{i\in J^*}\left(2\bar{f}_i\left(\frac{P_i}{\gamma}\right) - \frac{\Delta_{i,\min}}{\lambda_{\mathcal{A}}}\frac{P_i}{\gamma}\right) + \sum_{i\in I^*}\left(4\left(1+\delta_i\right)\sigma_i\sqrt{\frac{2}{\gamma^{\frac{1}{2}}}\frac{Q_i}{\gamma}} - \frac{\Delta_{i,\min}}{\lambda'_{\mathcal{A}}}\frac{Q_i}{\gamma}\right) \quad (93)$$

$$+ 2\sum_{i=1\in I^*}n_i\beta_i(1) + \frac{2}{\gamma}\left(dW + 2\sum_{i=1}^{d}\delta_i n_i\delta_i + \frac{5}{4}\sum_{i\in J^*}n_i\xi_i\right)$$

$$\leq \sum_{i\in J^*}\max_{x\geq 0}\left\{2\bar{f}_i\left(x\right) - \frac{\Delta_{i,\min}}{\lambda_{\mathcal{A}}}x\right\} + \sum_{i\in I^*}\max_{x\geq 0}\left\{4\left(1+\delta_i\right)\sigma_i\sqrt{\frac{2}{\gamma^{\frac{1}{2}}}x} - \frac{\Delta_{i,\min}}{\lambda'_{\mathcal{A}}}x\right\} \quad (94)$$

$$+ 2\sum_{i\in I^*}n_i\beta_i(1) + \frac{2}{\gamma}\left(dW + 2\sum_{i=1}^{d}\delta_i n_i + \frac{5}{4}\sum_{i\in J^*}n_i\xi_i\right)$$

$$\leq \sum_{i\in J^*}\max_{x\geq 0}\left\{2\bar{f}_i\left(x\right) - \frac{\Delta_{i,\min}}{\lambda_{\mathcal{A}}}x\right\} + \sum_{i\in I^*}\frac{16\left(1+\delta_i\right)^2\lambda'_{\mathcal{A}}\sigma_i^2}{\sqrt{\gamma}\Delta_{i,\min}}$$

$$+ 2\sum_{i\in I^*}n_i\beta_i(1) + \frac{2}{\gamma}\left(dW + 2\sum_{i=1}^{d}\delta_i n_i + \frac{5}{4}\sum_{i\in J^*}n_i\xi_i\right) \quad (95)$$

where the second inequality follows from (90) and the last inequality follows from $a\sqrt{x} - bx \leq \frac{a^2}{2b}$ for $a, b, x \geq 0$. In the following, we evaluate the first term of (95).

**Bounding the first term of (95)** We will prove the following statement:

$$\max_{x\geq 0}\left\{2\bar{f}_i\left(x\right) - \frac{\Delta_{i,\min}}{\lambda_{\mathcal{A}}}x\right\} \leq h\left(\lambda_{\mathcal{A}}\frac{\sigma_i^2}{\Delta_{i,\min}}\right) + \mathcal{O}\left(\frac{\log(1+\gamma)}{\gamma}\right), \quad (96)$$

where $h : \mathbb{R}_+ \to \mathbb{R}$ is defined as

$$h_i(z) = \begin{cases} 4n_i\beta_i(1) & \text{if } 0 \leq z \leq \frac{n_i\beta_i(1)}{2\left(1+\frac{\delta_i}{n_i\beta_i(1)}\right)}, \\ 2z\left(1 + \sqrt{1+2\frac{\delta_i}{z}}\right) - 2\delta_i \\ \quad +4\delta_i\left(\log\frac{z}{n_i\beta_i(1)} + \log\left(1+\sqrt{1+2\frac{\delta_i}{z}}\right)\right) \\ \quad +\frac{(n_i\beta_i(1))^2}{z} - 2n_i\beta_i(1) & \text{if } z > \frac{n_i\beta_i(1)}{2\left(1+\frac{\delta_i}{n_i\beta_i(1)}\right)}. \end{cases} \quad (97)$$

Let $\bar{\Delta}_i = \frac{\Delta_{i,\min}}{\lambda_{\mathcal{A}}}$ for the notational simplicity. As $\bar{f}_i$ is concave, the maximum of $2\bar{f}_i(x) - \bar{\Delta}_{i,\min}x$ is attained by $x_i^* \in \mathbb{R}$ satisfying $2\bar{f}'_i(x_i^*) = \bar{\Delta}_i$. Define $\tilde{x}_i \geq 0$ by

$$\tilde{x}_i := \max\left\{\left(\frac{4\sigma_i}{\bar{\Delta}_i}\right)^2, \frac{8\delta_i n_i}{\bar{\Delta}_i}, \frac{16\xi_i n_i}{\gamma\bar{\Delta}_i}\right\}. \quad (98)$$

We then have

$$2\bar{f}'_i(\tilde{x}_i) \leq \frac{2\sigma_i}{\sqrt{\left(\frac{4\sigma_i}{\bar{\Delta}_i}\right)^2}} + \frac{2\delta_i n_i\sigma_i^2}{(n_i\beta_i(1))^2 + \sigma_i^2\frac{8\delta_i n_i}{\bar{\Delta}_i}} + \frac{2n_i\xi_i}{1 + \gamma\frac{16n_i\xi_i}{\gamma\bar{\Delta}_i}}$$

$$\leq \frac{\bar{\Delta}_i}{2} + \frac{\bar{\Delta}_i}{4} + \frac{\bar{\Delta}_i}{8} \leq \bar{\Delta}_i,$$

which implies $\tilde{x}_i \geq x_i^*$. Hence, we have

$$\max_{x \geq 0}\{2f_i(x) - \bar{\Delta}_i x\} = 2f_i(x_i^*) - \bar{\Delta}_i x_i^*$$

$$=4\sqrt{(n_i\beta_i(1))^2 + \sigma_i^2 x_i^*} + 2\delta_i \log\left(1 + \frac{\sigma_i^2 x_i^*}{(n_i\beta_i(1))^2}\right)$$

$$+ 2\frac{n_i\xi_i}{\gamma}\log\left(1 + \gamma x_i^*\right) - \bar{\Delta}_i x_i^* - 2n_i\beta_i(1)$$

$$\leq \max_{x \geq 0}\left\{4\sqrt{(n_i\beta_i(1))^2 + \sigma_i^2 x} + 2\delta_i \log\left(1 + \frac{\sigma_i^2}{(n_i\beta_i(1))^2}x\right) - \bar{\Delta}_i x\right\}$$

$$+ 2\frac{n_i\xi_i}{\gamma}\log\left(1 + \gamma \tilde{x}_i\right) - 2n_i\beta_i(1)$$

$$=\max_{x \geq 0}\left\{g_i(x) - \bar{\Delta}_i x\right\} - 2n_i\beta_i(1) + \mathcal{O}\left(\frac{(\log(1+\gamma))}{\gamma}\right), \tag{99}$$

where we define

$$g_i(x) = 4\sqrt{(n_i\beta_i(1))^2 + \sigma_i^2 x} + 2\delta_i \log\left(1 + \frac{\sigma_i^2 x}{(n_i\beta_i(1))^2}\right). \tag{100}$$

From (99) and (95), we have

$$\limsup_{T \to \infty} \frac{R_T}{\log T} \leq \sum_{i \in J^*} \left(\max_{x \geq 0}\left\{g_i(x) - \bar{\Delta}_i x\right\} - 2n_i\beta_i(1)\right) + 2\sum_{i \in I^*} n_i\beta_i(1). \tag{101}$$

In the following, we write $z_i = \frac{\sigma_i^2}{\bar{\Delta}_i}$. As we have

$$g_i'(x) = \frac{2\sigma_i^2}{\sqrt{(n_i\beta_i(1))^2 + \sigma_i^2 x}} + \frac{2\delta_i\sigma_i^2}{(n_i\beta_i(1))^2 + \sigma_i^2 x} \leq 2\sigma_i^2\left(\frac{1}{n_i\beta_i(1)} + \frac{\delta}{(n_i\beta_i(1))^2}\right), \tag{102}$$

if $z_i = \frac{\sigma_i^2}{\bar{\Delta}_i} \leq \frac{1}{2\left(\frac{1}{n_i\beta_i(1)} + \frac{\delta}{(n_i\beta_i(1))^2}\right)} = \frac{n_i\beta_i(1)}{2\left(1 + \frac{\delta}{n_i\beta_i(1)}\right)}$, the maximum of $g_i(x) - \bar{\Delta}_i x$ is attained by $x = 0$, implying

$$\max\left\{g_i(x) - \bar{\Delta}_i x\right\} = g_i(0) = 4n_i\beta_i(1) \quad \text{if} \quad z_i := \frac{\sigma_i^2}{\bar{\Delta}_i} \leq \frac{n_i\beta_i(1)}{2\left(1 + \frac{\delta}{n_i\beta_i(1)}\right)}. \tag{103}$$

Otherwise, we have

$$g_i(x) - \bar{\Delta}_i x = 4n_i\beta_i(1)\sqrt{1 + \frac{\sigma_i^2}{(n_i\beta_i(1))^2}x} + 2\delta_i \log\left(1 + \frac{\sigma_i^2}{(n_i\beta_i(1))^2}x\right)$$

$$- \frac{(n_i\beta_i(1))^2 \bar{\Delta}_i}{\sigma_i^2}\left(1 + \frac{\sigma_i^2 x}{(n_i\beta_i(1))^2}\right) + \frac{(n_i\beta_i(1))^2}{z_i}$$

$$= 4n_i\beta_i(1)\sqrt{1 + \frac{\sigma_i^2}{(n_i\beta_i(1))^2}x} + 4\delta_i \log\left(\sqrt{1 + \frac{\sigma_i^2}{(n_i\beta_i(1))^2}x}\right)$$

$$- \frac{(n_i\beta_i(1))^2}{z_i}\left(\sqrt{1 + \frac{\sigma_i^2 x}{(n_i\beta_i(1))^2}}\right)^2 + \frac{(n_i\beta_i(1))^2}{z_i}.$$

From this, by setting $y = \sqrt{1 + \frac{\sigma_i^2 x}{(n_i\beta_i(1))^2}}$, we obtain

$$\max_{x \geq 0}\left\{g_i(x) - \bar{\Delta}_i x\right\} \leq \max_{y \geq 0}\left\{4n_i\beta_i(1)y + 4\delta_i \log y - \frac{(n_i\beta_i(1))^2}{z_i}y^2\right\} + \frac{(n_i\beta_i(1))^2}{z_i}. \tag{104}$$

We here use the following:

$$\max_{y \geq 0} \left\{ ay + b \log y - cy^2 \right\} = \frac{1}{2} \left( \frac{a}{4c} \left( a + \sqrt{a^2 + 8bac} - b \right) \right) + b \log \frac{a + \sqrt{a^2 + 8bc}}{4c}, \quad (105)$$

which holds for any $a, b, c > 0$. We hence have

$$\max_{y \geq 0} \left\{ 4n_i\beta_i(1)y + 4\delta_i \log y - \frac{(n_i\beta_i(1))^2}{z_i} y^2 \right\}$$

$$= \frac{1}{2} \left( \frac{4n_i\beta_i(1)z_i}{4(n_i\beta_i(1))^2} \left( 4n_i\beta_i(1) + \sqrt{(4n_i\beta_i(1))^2 + 32\frac{\delta_i(n_i\beta_i(1))^2}{z_i}} \right) - 4\delta_i \right)$$

$$+ 4\delta \log \frac{4n_i\beta_i(1) + \sqrt{(4n_i\beta_i(1))^2 + 32\frac{\delta_i(n_i\beta_i(1))^2}{z_i}}}{4\frac{(n_i\beta_i(1))^2}{z_i}}$$

$$= 2 \left( z_i \left( n_i\beta_i(1) + \sqrt{1 + 2\frac{\delta}{z_i}} \right) - \delta_i \right) + 4\delta_i \left( \log \frac{z_i}{n_i\beta_i(1)} + \log \left( 1 + \sqrt{1 + 2\frac{\delta_i}{z_i}} \right) \right). \quad (106)$$

Combining (99) with (103), (104), (106), we obtain

$$\max_{x \geq 0} \left\{ 2f_i(x) - \bar{\Delta}_i x \right\} \leq h_i \left( \frac{\sigma_i^2}{\bar{\Delta}_i} \right) + \mathcal{O} \left( \frac{\log(1 + \gamma)}{\gamma} \right)$$

$$= h_i \left( \lambda_\mathcal{A} \frac{\sigma_i^2}{\Delta_{i,\min}} \right) + \mathcal{O} \left( \frac{\log(1 + \gamma)}{\gamma} \right) \quad (107)$$

where $h_i : \mathbb{R}_+ \to \mathbb{R}$ is defined by (97). From (95) and (107), we complete the proof of (96).

**Bouding $h$** For $z > \frac{n_i\beta_i(1)}{2\left(1 + \frac{\delta_i}{n_i\beta_i(1)}\right)}$, $h(z)$ in (97) is bounded as

$$h_i(z) \leq 2z \left( 1 + 1 + \frac{\delta_i}{z} \right) - 2\delta_i + 4\delta_i \left( \log z + \log \left( 1 + \sqrt{1 + 2\frac{\delta_i}{z}} \right) \right)$$

$$+ \frac{(n_i\beta_i(1))^2}{n_i\beta_i(1)} \cdot 2 \left( 1 + \frac{\delta_i}{n_i\beta_i(1)} \right) - 2n_i\beta_i(1)$$

$$= 4z + 4\delta_i \left( \log z + \log \left( 1 + \sqrt{1 + 2\frac{\delta_i}{z}} \right) + \frac{1}{2} \right)$$

$$\leq 4z + c_i \log(1 + z) \quad \left( c = \mathcal{O}\left(\delta_i^2\right) \right), \quad (108)$$

where the last inequality follows from $\log(1 + z) = \Omega\left(\frac{1}{\delta}\right)$ that holds for $z > \frac{n_i\beta_i(1)}{2\left(1 + \frac{\delta_i}{n_i\beta_i(1)}\right)}$. Hence, for any $z > 0$, $h_i(z)$ is bounded as

$$h_i(z) = \max \left\{ 4z + c_i \log(1 + z), 2n_i\beta_i(1) \right\}. \quad (109)$$

From this and (107), we obtain

$$R_T \leq \left( \sum_{i \in J^*} \max \left\{ 4\frac{\lambda_\mathcal{A}\sigma_i^2}{\Delta_{i,\min}} + c_i \log \left( 1 + \frac{\lambda_\mathcal{A}\sigma_i^2}{\Delta_{i,\min}} \right), 2n_i\beta_i(1) \right\} + 2 \sum_{i \in I^*} n_i\beta_i(1) \right) \log T \quad (110)$$

$$+ \sum_{i \in I^*} \frac{16(1 + \delta_i)^2 \lambda'_\mathcal{A}\sigma_i^2}{\Delta'_{i,\min}} \sqrt{\log T} + o\left(\sqrt{\log T}\right) \quad (111)$$

which completes the proof of upper bound of the LS method under the stochastic regime for the stochastic regime.

### C.3.3 PROOF FOR THE STOCHASTIC REGIME WITH ADVERSARIAL CORRUPTIONS

We here show a regret bound for the stochastic regime with adversarial corruptions, which is the following regret bound:

$$R_T \leq \mathcal{R}^{\mathrm{LS}} + \mathcal{O}\left(CMR^{\mathrm{LS}}\right), \tag{112}$$

where $\mathcal{R}^{\mathrm{LS}}$ is $\mathcal{O}\left(\sum_{i \in J^*} \frac{\lambda_{\mathcal{A}} \sigma_i^2}{\Delta_i} \log T\right)$ and $C$ is the corruption level defined in Section 2.

*Proof.* In the stochastic regime with adversarial corruptions, using Lemma C.5 with $m_i^* = \ell_i$ we have

$$
\mathbb{E}\left[\sum_{t=1}^{T} \alpha_i(t)\right]
$$

$$
\leq \mathbb{E}\left[\sum_{t=1}^{T}\left(\frac{a_i(t)}{n_i}\right)^2 (k_i(t) - \mu_i)^2 + \log\left(1 + \sum_{t=1}^{T} a_i(t)\right)\right] + \frac{5}{4}
$$

$$
= \mathbb{E}\left[\sum_{t=1}^{T}\left(\frac{a_i(t)}{n_i}\right)^2 (k_i(t) - l_i'(t) + l_i'(t) - \mu_i)^2 + \log\left(1 + \sum_{t=1}^{T} a_i(t)\right)\right] + \frac{5}{4}
$$

$$
\leq \frac{\sigma_i^2}{n_i^2} P_i + \log\left(1 + P_i\right) + \frac{5}{4} + P_i', \tag{113}
$$

where we define

$$
P_i' = \mathbb{E}\left[\sum_{t=1}^{T}\left(\frac{a_i(t)}{n_i}\right)^2 (k_i(t) - l_i'(t))^2\right]. \tag{114}
$$

Hence, in a similar argument to that of showing (86), by using (113) instead of (82), we obtain

$$\mathbb{E}\left[2\beta_i(T+1) - \beta_i(1) + 2\delta_i \log \frac{\beta_i(T+1)}{\beta_i(1)}\right]$$

$$= \mathbb{E}\left[2\sqrt{\beta_i^2(1) + \frac{1}{\gamma}\sum_{t=1}^T \alpha_i(t)} - \beta_i(1) + \delta_i \log\left(1 + \frac{1}{\gamma\beta_i^2(1)}\sum_{t=1}^T \alpha_i(t)\right)\right]$$

$$\leq 2\sqrt{(\beta_i(1))^2 + \frac{1}{\gamma}\left(\frac{\sigma_i^2}{n_i^2}P_i + \log(1+P_i) + \frac{5}{4} + P_i'\right)} - \beta_i(1)$$

$$+ \delta_i \log\left(1 + \frac{1}{\gamma\beta_i(1)^2}\left(\frac{\sigma_i^2}{n_i^2}P_i + \log(1+P_i) + \frac{5}{4} + P_i'\right)\right)$$

$$\leq 2\sqrt{(\beta_i(1))^2 + \frac{\sigma_i^2}{n_i^2}\frac{P_i}{\gamma} + \frac{1}{\gamma\beta_i(1)}\left(\log(1+P_i) + \frac{5}{4}\right) + 2\sqrt{\frac{P_i'}{\gamma}}} - \beta_i(1)$$

$$+ \delta_i \log\left(1 + \frac{1}{\gamma\beta_i(1)^2}\left(\frac{\sigma_i^2}{n_i^2}P_i + P_i'\right)\right) + \frac{\delta_i}{\gamma(\beta_i(1))^2}\left(\log(1+P_i) + \frac{5}{4}\right)$$

$$\leq 2\sqrt{(\beta_i(1))^2 + \frac{\sigma_i^2}{n_i^2}\frac{P_i}{\gamma} + \frac{1}{\gamma\beta_i(1)}\left(\log(1+P_i) + \frac{5}{4}\right) + 2\sqrt{\frac{P_i'}{\gamma}}} - \beta_i(1)$$

$$+ \delta_i \log\left(\left(1 + \frac{1}{\gamma\beta_i(1)^2}\frac{\sigma_i^2}{n_i^2}P_i\right) \cdot \left(1 + \frac{1}{\gamma\beta_i(1)^2}P_i'\right)\right) + \frac{\delta_i}{\gamma(\beta_i(1))^2}\left(\log(1+P_i) + \frac{5}{4}\right)$$

$$\leq 2\sqrt{(\beta_i(1))^2 + \frac{\sigma_i^2 P_i}{n_i^2\gamma}} - \beta_i(1) + \delta_i \log\left(1 + \frac{\sigma_i^2 P_i}{\gamma(n_i\beta_i(1))^2}\right) + \frac{\xi_i}{\gamma}\left(\log(1+P_i) + \frac{5}{4}\right)$$

$$+ 2\sqrt{\frac{P_i'}{\gamma}} + \delta_i \log\left(1 + \frac{P_i'}{\gamma\beta_i^2(1)}\right)$$

$$\leq 2\sqrt{(\beta_i(1))^2 + \frac{\sigma_i^2}{n_i^2}\frac{P_i}{\gamma}} - \beta_i(1) + \delta_i \log\left(1 + \frac{\sigma_i^2 P_i}{\gamma(n_i\beta_i(1))^2}\right) + \frac{\xi_i}{\gamma}\left(\log(1+P_i) + \frac{5}{4}\right)$$

$$+ \left(2 + \frac{\delta_i}{\beta_i(1)}\right)\sqrt{\frac{P_i'}{\gamma}},$$

where the last inequality follows from $\log(1+x) \leq \sqrt{x}$ for $x \geq 0$. Combining this with (38) and (88), via a similar argument to that of showing (90), we have

$$\frac{R_T}{\gamma} \leq \sum_{i\in J^*} \bar{f}_i\left(\frac{P_i}{\gamma}\right) + 2\sum_{i\in I^*}(1+\delta_i)\sigma_i\sqrt{\frac{2}{\gamma^{\frac{3}{2}}}Q_i} + \sum_{i\in I^*} n_i\beta_i(1) + \frac{1}{\gamma}\left(dW + 2\sum_{i=1}^d \delta_i n_i + \frac{5}{4}\sum_{i\in J^*} n_i\xi_i\right)$$

$$+ \sum_{i\in J^*}\left(2 + \frac{\delta_i}{\beta_i(1)}\right)n_i\sqrt{\frac{P_i'}{\gamma}}$$

$$\leq \sum_{i\in J^*} \bar{f}_i\left(\frac{P_i}{\gamma}\right) + 2\sum_{i\in I^*}(1+\delta_i)\sigma_i\sqrt{\frac{2}{\gamma^{\frac{3}{2}}}Q_i} + \sum_{i\in I^*} n_i\beta_i(1) + \frac{1}{\gamma}\left(dW + 2\sum_{i=1}^d \delta_i n_i + \frac{5}{4}\sum_{i\in J^*} n_i\xi_i\right)$$

$$+ 2\sum_{i\in J^*} \frac{n_i\delta_i}{\beta_i(1)}\sqrt{\frac{P_i'}{\gamma}}, \tag{115}$$

where $2 \leq \frac{\delta_i}{\beta_i(1)}$ and $\bar{f}_i$ is defined by

$$\bar{f}_i(x) = 2\sqrt{(n_i\beta_i(1))^2 + \sigma_i^2 x} + n_i\delta_i \log\left(1 + \frac{\sigma_i^2 x}{\gamma(n_i\beta_i(1))^2}\right) + \frac{n_i\xi_i}{\gamma}\log(1+\gamma x) - n_i\beta_i(1).$$

We further have

$$
\begin{aligned}
&\frac{\sum\limits_{i \in J^*} \frac{n_i \delta_i}{\beta_i(1)} \sqrt{\frac{P'_i}{\gamma}}}{} \\
&\leq \sqrt{\frac{\sum\limits_{i \in J^*} \left(\frac{n_i \delta_i}{\beta_i(1)}\right)^2}{\gamma} \sum_{i \in J^*} P'_i} \\
&= \sqrt{\frac{\sum\limits_{i \in J^*} \left(\frac{n_i \delta_i}{\beta_i(1)}\right)^2}{\gamma} \mathbb{E}\left[\sum_{t=1}^{T} \sum_{i \in J^*} \left(\frac{a_i(t)}{n_i}\right)^2 (k_i(t) - l'_i(t))^2\right]} \\
&\leq \sqrt{\frac{|J^*| \sum\limits_{i \in J^*} \left(\frac{n_i \delta_i}{\beta_i(1)}\right)^2}{\gamma} \mathbb{E}\left[\sum_{t=1}^{T} \max_{i \in [d], j \in [n_i]} \left|L_{i,j}(t) - L'_{i,j}(t)\right|^2\right]} \\
&= \sqrt{\frac{|J^*| \sum\limits_{i \in J^*} \left(\frac{n_i \delta_i}{\beta_i(1)}\right)^2}{\gamma} C}
\end{aligned}
\tag{116}
$$

where the first inequality follows from the Cauchy-Shwarz inequality, the first equality follows from the definition of $P'_i$ in (114), and the second inequality follows from the fact that $\sum\limits_{i \in J^*} \left(\frac{a_i(t)}{n_i}\right)^2 \leq |J^*|$. Combining (115) and (116), we obtain

$$
\begin{aligned}
\frac{R_T}{\gamma} \leq{}& \sum_{i \in J^*} \bar{f}_i \left(\frac{P_i}{\gamma}\right) + 2 \sum_{i \in I^*} (1 + \delta_i) \sigma_i \sqrt{\frac{2}{\gamma^{\frac{3}{2}}} Q_i} + \sum_{i \in I^*} n_i \beta_i(1) \\
&+ \frac{1}{\gamma}\left(dW + d + 2W\delta_i + \frac{5}{4} \sum_{i \in J^*} n_i \xi_i\right) \\
&+ 2\sqrt{\frac{|J^*| \sum\limits_{i \in J^*} \left(\frac{n_i \delta_i}{\beta_i(1)}\right)^2}{\gamma} C}
\end{aligned}
\tag{117}
$$

From (117) and Lemma C.4, for any $\chi \in (0, 1]$, we have

$$\frac{R_T}{\log T}$$

$$= (1 + \chi)\frac{R_T}{\gamma} - \lambda \frac{R_T}{\gamma}$$

$$\leq (1 + \chi)\frac{R_T}{\gamma} - \frac{\lambda}{\gamma}\left(\frac{1}{\lambda(\mathcal{A})}\sum_{i \in I^*}\Delta'_{i,\min}Q_i + \frac{1}{\lambda_{\mathcal{A}}}\sum_{i \in J^*}\Delta_{i,\min}P_i - 2CM\right)$$

$$\leq \sum_{i \in J^*}\left((1+\chi)\bar{f}_i\left(\frac{P_i}{\gamma}\right) - \lambda\frac{\Delta_{i,\min}}{\lambda_{\mathcal{A}}}\frac{P_i}{\gamma}\right) + \sum_{i \in I^*}\left(2(1+\chi)(1+\delta_i)\sqrt{\frac{2}{\gamma^{1/2}}\frac{Q_i}{\gamma}} - \lambda\frac{\Delta'_{i,\min}}{\lambda'_{\mathcal{A}}}\frac{Q_i}{\gamma}\right)$$

$$+ 2(1+\chi)\sqrt{\frac{|J^*|\sum_{i \in J^*}\left(\frac{n_i\delta_i}{\beta_i(1)}\right)^2}{\gamma}}C + \frac{2\lambda CM}{\gamma}$$

$$+ (1+\chi)\left(\sum_{i \in I^*}n_i\beta_i(1) + \frac{1}{\gamma}\left(dW + \sum_{i=1}^d n_i\delta_i + \frac{5}{4}\sum_{i \in J^*}\xi_i n_i\right)\right)$$

$$\leq \sum_{i \in J^*}\max_{x \geq 0}\left\{(1+\chi)\bar{f}_i(x) - \lambda\frac{\Delta_{i,\min}}{\lambda_{\mathcal{A}}}x\right\} + \sum_{i \in I^*}\max_{x \geq 0}\left\{2(1+\chi)(1+\delta)\sqrt{\frac{2}{\gamma^{1/2}}x} - \lambda\frac{\Delta'_{i,\min}}{\lambda'_{\mathcal{A}}}x\right\}$$

$$+ 2(1+\chi)\sqrt{\frac{|J^*|\sum_{i \in J^*}\left(\frac{n_i\delta_i}{\beta_i(1)}\right)^2}{\gamma}}C + \frac{2\lambda CM}{\gamma}$$

$$+ (1+\chi)\left(\sum_{i \in I^*}n_i\beta_i(1) + \frac{1}{\gamma}\left(dW + \sum_{i=1}^d n_i\delta_i + \frac{5}{4}\sum_{i \in J^*}\xi_i n_i\right)\right)$$

$$\leq \sum_{i \in J^*}\max_{x \geq 0}\left\{(1+\chi)\bar{f}_i(x) - \lambda\frac{\Delta_{i,\min}}{\lambda_{\mathcal{A}}}x\right\} + \sum_{i \in I^*}\frac{4(1+\lambda)^2(1+\delta)^2\lambda'_{\mathcal{A}}}{\lambda\sqrt{\gamma}\Delta'_{i,\min}}$$

$$+ 2(1+\chi)\sqrt{\frac{|J^*|\sum_{i \in J^*}\left(\frac{n_i\delta_i}{\beta_i(1)}\right)^2}{\gamma}}C$$

$$+ (1+\chi)\left(\sum_{i \in I^*}n_i\beta_i(1) + \frac{1}{\gamma}\left(dW + \sum_{i=1}^d n_i\delta_i + \frac{5}{4}\sum_{i \in J^*}\xi_i n_i\right)\right). \tag{118}$$

Further, letting $\bar{\Delta}_i = \frac{\Delta_{i,\min}}{\lambda_{\mathcal{A}}}$, we have

$$\max_{x \geq 0}\left\{(1+\chi)\bar{f}_i(x) - \chi\bar{\Delta}_i x\right\}$$

$$= \frac{1+\chi}{2}\max_{x \geq 0}\left\{2\bar{f}_i(x) - \frac{2\chi\bar{\Delta}_i}{1+\chi}x\right\}$$

$$\leq \frac{1+\chi}{2}h\left(\frac{(1+\chi)\sigma_i^2}{2\chi\bar{\Delta}_i}\right) + \mathcal{O}\left(\frac{\log(1+\gamma)}{\gamma}\right)$$

$$\leq \max\left\{\frac{(1+\chi)^2}{\chi}\frac{\sigma^2}{\bar{\Delta}_i} + c_i\log\left(1 + \frac{\sigma_i^2}{\chi\bar{\Delta}_i}\right), (1+\chi)n_i\beta_i(1)\right\} + \mathcal{O}\left(\frac{\log(1+\gamma)}{\gamma}\right)$$

$$\leq \max\left\{4\frac{\sigma_i^2}{\bar{\Delta}_i} + c_i\log\left(1 + \frac{\sigma_i^2}{\bar{\Delta}_i}\right), 2n_i\beta_i(1)\right\} + (1+c_i)\left(\frac{1}{\chi} - 1\right)\frac{\sigma_i^2}{\bar{\Delta}_i} + \mathcal{O}\left(\frac{\log(1+\gamma)}{\gamma}\right) \tag{119}$$

where $h(z)$ is defined as (97), the first inequality follows from (107), the second inequality comes from (109) and $\chi \in (0, 1]$, and the last inequality follows from

$$\frac{(1+\chi)^2}{\chi} = \chi + 2 + \frac{1}{\chi} \leq 3 + \frac{1}{\chi} = 4 + \left(\frac{1}{\chi} - 1\right)$$

and

$$\log\left(1 + \frac{\sigma_i^2}{\chi \bar{\Delta}_i}\right) \leq \frac{1}{\chi} \log\left(1 + \frac{\sigma_i^2}{\bar{\Delta}_i}\right)$$

$$\leq \log\left(1 + \frac{\sigma_i^2}{\bar{\Delta}_i}\right) + \left(\frac{1}{\chi} - 1\right)\frac{\sigma_i^2}{\bar{\Delta}_i}.$$

Using (118), (119), and $\chi \leq 1$, we obtain

$$\frac{R_T}{\log T} \leq \sum_{i \in J^*} \max\left\{4\frac{\sigma_i^2}{\bar{\Delta}_i} + c_i \log\left(1 + \frac{\sigma_i^2}{\bar{\Delta}_i}\right), 2n_i\beta_i(1)\right\} + 2\sum_{i \in I^*} n_i\beta_i(1) + 2\sqrt{\frac{|J^*| \sum_{i \in J^*} \left(\frac{n_i\delta_i}{\beta_i(1)}\right)^2}{\gamma}}C$$

$$+ 2\chi\frac{CM}{\gamma} + \left(\frac{1}{\chi} - 1\right)\sum_{i \in J^*}(1+c_i)\frac{\sigma_i^2}{\bar{\Delta}_i} + \sum_{i \in I^*}\frac{4(1+\chi)^2(1+\delta)^2\lambda'_{\mathcal{A}}}{\chi\sqrt{\gamma}\Delta'_{i,\min}} + \mathcal{O}\left(\frac{\log(1+\gamma)}{\gamma}\right)$$

$$\tag{120}$$

By choosing $\chi = \sqrt{\frac{\gamma \sum_{i \in J^*}\left(\frac{\sigma_i^2}{\bar{\Delta}_i} + \left(\frac{n_i\delta_i}{\beta_i(1)}\right)^2\right)}{\gamma \sum_{i \in J^*}\left(\frac{\sigma_i^2}{\bar{\Delta}_i} + \left(\frac{n_i\delta_i}{\beta_i(1)}\right)^2\right) + 2CM}}$, we have

$$\chi \leq \sqrt{\frac{\gamma \sum_{i \in J^*}\left(\frac{\sigma_i^2}{\bar{\Delta}_i} + \left(\frac{n_i\delta_i}{\beta_i(1)}\right)^2\right)}{\gamma \sum_{i \in J^*}\left(\frac{\sigma_i^2}{\bar{\Delta}_i} + \left(\frac{n_i\delta_i}{\beta_i(1)}\right)^2\right) + 2CM}} \tag{121}$$

and

$$\frac{1}{\chi} - 1 = \sqrt{1 + \frac{2CM}{\gamma \sum_{i \in J^*}\left(\frac{\sigma_i^2}{\bar{\Delta}_i} + \left(\frac{n_i\delta_i}{\beta_i(1)}\right)^2\right)}} - 1$$

$$\leq \sqrt{\frac{2CM}{\gamma \sum_{i \in J^*}\left(\frac{\sigma_i^2}{\bar{\Delta}_i} + \left(\frac{n_i\delta_i}{\beta_i(1)}\right)^2\right)}}, \tag{122}$$

which implies that

$$2\sqrt{\frac{|J^*| \sum_{i \in J^*}\left(\frac{n_i\delta_i}{\beta_i(1)}\right)^2}{\gamma}}C + \frac{2\chi CM}{\gamma} + (1+c)\left(\frac{1}{\chi} - 1\right)\sum_{j \in J^*}\frac{\sigma_i^2}{\bar{\Delta}_i}$$

$$= \mathcal{O}\left(\sqrt{\frac{C\max\{M, |J^*|\}}{\gamma}\sum_{i \in J^*}\left(\frac{\sigma_i^2}{\bar{\Delta}_i} + \left(\frac{n_i\delta_i}{\beta_i(1)}\right)^2\right)}\right). \tag{123}$$

From this and (120), recalling that $\gamma = \log T$ and $\bar{\Delta}_i = \frac{\Delta_{i,\min}}{\lambda_{\mathcal{A}}}$, we obtain

$$
\begin{aligned}
R_T &\leq \left( \sum_{i \in J^*} \max\left\{ 4\lambda_{\mathcal{A}} \frac{\sigma_i^2}{\Delta_{i,\min}} + c \log\left( 1 + \lambda_{\mathcal{A}} \frac{\sigma_i^2}{\Delta_{i,\min}} \right), 2n_i \beta_i(1) \right\} + 2 \sum_{i \in I^*} n_i \beta_i(1) \right) \log T \\
&\quad + \mathcal{O}\left( \sqrt{ C \max\{M, |J^*|\} \sum_{i \in J^*} \left( \lambda_{\mathcal{A}} \frac{\sigma_i^2}{\Delta_{i,\min}} + \left( \frac{n_i \delta_i}{\beta_i(1)} \right)^2 \right) \log T } \right) \\
&\quad + \sum_{i \in I^*} \frac{(1+\chi)^2}{\chi} \frac{4(1+\delta)^2 \lambda'_{\mathcal{A}}}{\Delta'_{i,\min}} \sqrt{\log T} + o\left( \sqrt{\log T} \right)
\end{aligned} \tag{124}
$$

which completes the proof of the stochastic regime with adversarial corruption. $\qquad\square$

### C.3.4   PROOF FOR THE ADVERSARIAL REGIME

**Proof for the adversarial regime.** First, we prove $R_T \leq \sqrt{4W Q_2 \log T} + \mathcal{O}(W \log T) + dW + d + 2W\delta$. For any $\boldsymbol{q}^* \in [0,1]^d$, bounding the RHS of Lemma C.3, we have

$$
\begin{aligned}
R_T & \\
&\leq \gamma \sum_{i=1}^{d} n_i \mathbb{E}\left[ 2\beta_i(T+1) - \beta_i(1) + 2\delta \log\left( \frac{\beta_i(T+1)}{\beta_i(1)} \right) \right] + dW + 2 \sum_{i=1}^{d} \delta_i n_i \delta_i \\
&\leq 2\gamma \sum_{i=1}^{d} n_i \mathbb{E}\left[ \beta_i(T+1) \right] + \mathcal{O}\left( W\gamma + dW + \sum_{i=1}^{d} \delta_i n_i \delta_i \right) \\
&= 2\gamma \sum_{i=1}^{d} n_i \mathbb{E}\left[ \sqrt{ \beta_i^2(1) + \frac{1}{\gamma} \sum_{t=1}^{T} \alpha_i(t) } \right] + \mathcal{O}\left( W\gamma + dW + \sum_{i=1}^{d} \delta_i n_i \delta_i \right) \\
&\leq 2\gamma \sum_{i=1}^{d} n_i \mathbb{E}\left[ \sqrt{ \beta_i^2(1) + \frac{1}{\gamma} \left( \sum_{t=1}^{T} \left( \frac{a_i(t)}{n_i} \right)^2 (k_i(t) - m_i^*)^2 + \log(1 + a_i(t)) + \frac{5}{4} \right) } \right] \\
&\quad + \mathcal{O}\left( W\gamma + dW + \sum_{i=1}^{d} \delta_i n_i \delta_i \right) \\
&\leq 2 \sum_{i=1}^{d} n_i \mathbb{E}\left[ \sqrt{ \gamma \sum_{t=1}^{T} \left( \frac{a_i(t)}{n_i} \right)^2 (k_i(t) - m_i^*)^2 } \right] + \mathcal{O}\left( W\gamma + dW + \sum_{i=1}^{d} \delta_i n_i \delta_i \right) \\
&\leq 2 \sqrt{ \sum_{i=1}^{d} n_i^2 \gamma \sum_{i=1}^{d} \left( \mathbb{E}\left[ \sqrt{ \sum_{t=1}^{T} \left( \frac{a_i(t)}{n_i} \right)^2 (k_i(t) - m_i^*)^2 } \right] \right)^2 } + \mathcal{O}\left( W\gamma + dW + \sum_{i=1}^{d} \delta_i n_i \delta_i \right) \\
&\leq 2 \sqrt{ \sum_{i=1}^{d} n_i^2 \gamma \sum_{i=1}^{d} \mathbb{E}\left[ \sum_{t=1}^{T} \left( \frac{a_i(t)}{n_i} \right)^2 (k_i(t) - m_i^*)^2 \right] } + \mathcal{O}\left( W\gamma + dW + \sum_{i=1}^{d} \delta_i n_i \delta_i \right) \tag{125} \\
&\leq 2 \sqrt{ \sum_{i=1}^{d} n_i^2 \gamma \mathbb{E}\left[ \sum_{t=1}^{T} \| \boldsymbol{k}(t) - \boldsymbol{q}^* \|_2^2 \right] } + \mathcal{O}\left( W\gamma + dW + \sum_{i=1}^{d} \delta_i n_i \delta_i \right)
\end{aligned}
$$

where the second inequality follows from $\beta_i(T+1) = \mathcal{O}(T)$, the third inequality follows from Lemma C.5, and the fifth inequality follows from the Cauchy-Schwarz inequality. Since $\boldsymbol{q}^*$ is arbitrary, we obtain the desired results by $\boldsymbol{q}^* = \bar{l}$.

Next, we prove $R_T \leq \sqrt{4 \sum_{i=1}^{d} n_i{}^2 L^* \log T} + \mathcal{O}(d \log T) + dW + d + 2W\delta$. By setting $\boldsymbol{q}^* = 0$ in (125), we have

$$R_T$$

$$\leq 2\sqrt{\sum_{i=1}^{d} n_i{}^2 \gamma \sum_{i \in \{j \in [d] | a_j(t) \geq 1\}} \mathbb{E}\left[\sum_{t=1}^{T} \left(\frac{a_i(t)}{n_i}\right)^2 k_i(t)^2\right]} + \mathcal{O}\left(W\gamma + dW + \sum_{i=1}^{d} \delta_i n_i \delta_i\right)$$

$$\leq 2\sqrt{\sum_{i=1}^{d} n_i{}^2 \gamma \sum_{i \in \{j \in [d] | a_j(t) \geq 1\}} \mathbb{E}\left[\sum_{t=1}^{T} \left(\frac{a_i(t)}{n_i}\right)^2 k_i(t)\right]} + \mathcal{O}\left(W\gamma + dW + \sum_{i=1}^{d} \delta_i n_i \delta_i\right)$$

$$\leq 2\sqrt{\sum_{i=1}^{d} n_i{}^2 \gamma \sum_{i \in \{j \in [d] | a_j(t) \geq 1\}} \mathbb{E}\left[\sum_{t=1}^{T} \left(\frac{a_i(t) - a_i^*}{n_i}\right) k_i(t) + \frac{a_i^*}{n_i} k_i(t)\right]} + \mathcal{O}\left(W\gamma + dW + \sum_{i=1}^{d} \delta_i n_i \delta_i\right)$$

$$\leq 2\sqrt{\sum_{i=1}^{d} n_i{}^2 \gamma (R_T + L^*)} + \mathcal{O}\left(W\gamma + dW + \sum_{i=1}^{d} \delta_i n_i \delta_i\right)$$

where the third inequality follows from Jensen's inequality. By solving this equation in $R_T$, we obtain

$$R_T \leq 2\sqrt{\sum_{i=1}^{d} n_i{}^2 \gamma L^*} + \mathcal{O}\left(W\gamma + dW + \sum_{i=1}^{d} \delta_i n_i \delta_i\right) \tag{126}$$

which is the desired bound.

Finally, we prove $R_T \leq \sqrt{\sum_{i=1}^{d} n_i{}^2 (mT - L^*) \log T} + \mathcal{O}\left(W\gamma + dW + \sum_{i=1}^{d} \delta_i n_i \delta_i\right)$. By setting $\boldsymbol{q}^* = \mathbf{1}$ in (125) and repeating a similar argument as for proving $R_T \leq \sqrt{4 \sum_{i=1}^{d} n_i{}^2 L^* \log T} + \mathcal{O}\left(W\gamma + dW + \sum_{i=1}^{d} \delta_i n_i \delta_i\right)$, we have

$$R_T \leq 2\sqrt{\sum_{i=1}^{d} n_i{}^2 \gamma \sum_{i=1}^{d} \mathbb{E}\left[\sum_{t=1}^{T} \left(\frac{a_i(t)}{n_i}\right)^2 (k_i(t) - 1)^2\right]} + \mathcal{O}\left(W\gamma + dW + \sum_{i=1}^{d} \delta_i n_i \delta_i\right)$$

$$= 2\sqrt{\sum_{i=1}^{d} n_i{}^2 \gamma \sum_{i=1}^{d} \mathbb{E}\left[\sum_{t=1}^{T} \left(\frac{a_i(t)}{n_i}\right)^2 (1 - k_i(t))^2\right]} + \mathcal{O}\left(W\gamma + dW + \sum_{i=1}^{d} \delta_i n_i \delta_i\right)$$

$$\leq 2\mathbb{E}\left[\sqrt{\sum_{i=1}^{d} n_i{}^2 \gamma \sum_{i=1}^{d} \mathbb{E}\left[\sum_{t=1}^{T} a_i(t) (1 - k_i(t))\right]}\right] + \mathcal{O}\left(W\gamma + dW + \sum_{i=1}^{d} \delta_i n_i \delta_i\right)$$

$$\leq 2\sqrt{\sum_{i=1}^{d} n_i{}^2 \gamma \left(MT - \sum_{t=1}^{T} a_i^* k_i(t) - \sum_{t=1}^{T} k_i(t) (a_i(t) - a_i^*)\right)} + \mathcal{O}\left(W\gamma + dW + \sum_{i=1}^{d} \delta_i n_i \delta_i\right)$$

$$\leq 2\sqrt{\sum_{i=1}^{d} n_i{}^2 \gamma (MT - L^* - R_T)} + \mathcal{O}\left(W\gamma + dW + \sum_{i=1}^{d} \delta_i n_i \delta_i\right)$$

where the third inequality follows since $\sum_{i=1}^{d} a_i(t) \leq M$ and the forth inequality follows from Jensen's inequality. By solving this inequation in $R_T$, we obtain

$$R_T \leq 2\sqrt{\sum_{i=1}^{d} n_i{}^2 \gamma \left(MT - L^*\right) + \mathcal{O}\left(W\gamma + dW + \sum_{i=1}^{d} \delta_i n_i \delta_i\right)}, \tag{127}$$

which completes the proof.

### C.4 PROOF FOR THE GD METHOD

Here, we provide proofs for the GD method. We can prove it by a similar discussion to that for the LS method. We first discuss the key lemma for this argument.

### C.4.1 PRELIMINARIES

**Lemma C.8.** *Assume that $q_i(t)$ is given by (4). Then, for any $i \in [d]$ and $u_i(1), \ldots, u_i(T) \in [0, 1]$, we have*

$$\sum_{t=1}^{T} \alpha_i(t) \leq \sum_{t=1}^{T} a_i(t) \left(k_i(t) - q_i(t)\right)^2$$

$$\leq \frac{1}{1 - \eta} \sum_{t=1}^{T} a_i(t) \left(k_i(t) - u_i(t)\right)^2 + \frac{1}{\eta\left(1 - 2\eta\right)} \left(\frac{1}{4} + 2\sum_{t=1}^{T} |u_i(t+1) - u_i(t)|\right) \tag{128}$$

*Proof.* Take $i \in [d]$ satisfying $a_i(t) = 1$. Then, it holds that

$$\left(k_i(t) - q_i(t)\right)^2 - \left(k_i(t) - u_i(t)\right)^2$$
$$\leq 2\left(k_i(t) - q_i(t)\right)\left(u_i(t) - q_i(t)\right)$$
$$= 2(k_i(t) - q_i(t))(q_i(t+1) - q_i(t)) + 2(k_i(t) - q_i(t))(u_i(t) - q_i(t+1))$$
$$= 2\eta(k_i(t) - q_i(t))^2 + \frac{2}{\eta}(q_i(t+1) - q_i(t))(u_i(t) - q_i(t+1))$$
$$\leq 2\eta(k_i(t) - q_i(t))^2 + \frac{1}{\eta}(u_i(t) - q_i(t))^2 - (u_i(t) - q_i(t+1)^2),$$

where the inequalities follow from $y^2 - x^2 = 2y(y - x) - (x - y)^2 \leq 2y(y - x)$ for $x, y \in \mathbb{R}$ and the last inequality follows from the definition of $q_i(t)$ in (4). Hence, we have

$$\left(k_i(t) - q_i(t)\right)^2 \leq \frac{1}{1 - 2\eta} \left(\left(k_i(t) - u_i(t)\right)^2 + \frac{1}{\eta}\left(\left(u_i(t) - q_i(t)\right)^2 - \left(u_i(t) - q_i(t+1)\right)^2\right)\right). \tag{129}$$

From the definition of $\alpha_i(t)$ and (129), we have

$$
\begin{aligned}
\sum_{t=1}^{T} \alpha_i(t) \leq & \sum_{t=1}^{T} \left(\frac{a_i(t)}{n_i}\right)^2 (k_i(t) - q_i(t))^2 \\
\leq & \frac{1}{1-2\eta} \sum_{t=1}^{T} \left(\frac{a_i(t)}{n_i}\right)^2 (k_i(t) - u_i(t))^2 \\
& + \frac{1}{\eta(1-2\eta)} \sum_{t=1}^{T} \left(\frac{a_i(t)}{n_i}\right)^2 \left((u_i(t) - q_i(t))^2 - (u_i(t) - q_i(t+1))^2\right) \\
= & \frac{1}{1-2\eta} \sum_{t=1}^{T} \left(\frac{a_i(t)}{n_i}\right)^2 (k_i(t) - u_i(t))^2 \\
& + \frac{1}{\eta(1-2\eta)} \left(\sum_{t=1}^{T} \left((u_i(t+1) - q_i(t+1))^2 - (u_i(t) - q_i(t+1))^2\right) + (u_i(1) - q_i(1))^2\right) \\
\leq & \frac{1}{1-2\eta} \sum_{t=1}^{T} \left(\frac{a_i(t)}{n_i}\right)^2 (k_i(t) - u_i(t))^2 \\
& + \frac{1}{\eta(1-2\eta)} \left(\sum_{t=1}^{T} (u_i(t+1) + u_i(t) - 2q_i(t+1))(u_i(t+1) - u_i(t)) + \frac{1}{4}\right) \\
\leq & \frac{1}{1-2\eta} \sum_{t=1}^{T} \left(\frac{a_i(t)}{n_i}\right)^2 (k_i(t) - u_i(t))^2 \\
& + \frac{1}{\eta(1-2\eta)} \left(2 \sum_{t=1}^{T} |u_i(t+1) - u_i(t)| + \frac{1}{4}\right),
\end{aligned}
$$

which completes the proof. $\qquad\square$

### C.4.2 PROOF FOR THE STOCHASTIC REGIME

From Lemma C.8, setting $u_i(t) = k_i$ for all $i \in [d]$ and $t \in [T]$ and taking the expectation yield that

$$
\begin{aligned}
\mathbb{E}\left[\sum_{t=1}^{T} \alpha_i(t)\right] \leq & \frac{1}{1-2\eta} \mathbb{E}\left[\sum_{t=1}^{T} \left(\frac{a_i(t)}{n_i}\right)^2 (k_i(t) - k_i)^2\right] + \frac{1}{4\eta(1-2\eta)} \\
= & \frac{1}{1-2\eta} \frac{\sigma_i^2}{n_i^2} P_i + \frac{1}{4\eta(1-2\eta)},
\end{aligned}
$$

where $P_i$ is defined in (83). By using this inequality instead of (82) and repeating the same argument as that in Appendix C.3.2, we obtain that the upper bound of the regret is

$$
\mathcal{O}\left(\frac{1}{1-2\eta} \sum_{i \in J^*} \frac{\lambda_{\mathcal{A}} \sigma_i^2}{\Delta_i} \log T\right). \tag{130}
$$

### C.4.3 PROOF FOR THE STOCHASTIC REGIME WITH ADVERSARIAL CORRUPTION

Here, we show a regret upper bound of the GD method under the stochastic regime with adversarial corruptions given:

$$
R_T \leq R^{\mathrm{GD}} + \mathcal{O}(\sqrt{CMR^{\mathrm{GD}}}) \tag{131}
$$

*Proof.* Letting $u_i(t) = \mu_i$ for all $i \in [d]$ and $t \in [T]$ in Lemma C.8 and taking the expectation yield that

$$\mathbb{E}\left[\sum_{t=1}^{T} \alpha_i(t)\right] \leq \frac{1}{1-2\eta} \mathbb{E}\left[\left(\frac{a_i(t)}{n_i}\right)^2 (k_i(t) - \mu_i)^2\right] + \frac{1}{4\eta(1-2\eta)}$$

$$\leq \frac{1}{1-2\eta} \mathbb{E}\left[\left(\frac{a_i(t)}{n_i}\right)^2 (k_i(t) - l_i'(t) + l_i'(t) - \mu_i)^2\right] + \frac{1}{4\eta(1-2\eta)}$$

$$= \frac{1}{1-2\eta}\left(\frac{\sigma_i^2}{n_i^2} P_i + P_i'\right) + \frac{1}{4\eta(1-2\eta)}, \tag{132}$$

where $P_i$ is defined in (83) and the last inequality is obtained by a similar argument as for (113). By using this inequality instead of (82) and repeating a similar argument to that in the discussion of the LS method, we obtain the desired upper bound. $\qquad\square$

### C.4.4 PROOF FOR THE ADVERSARIAL REGIME

From Lemma C.8, we immediately obtain

$$\sum_{t=1}^{T}\sum_{i=1}^{d} \alpha_i(t) \leq \frac{1}{1-2\eta} \sum_{t=1}^{T}\sum_{i=1}^{d}\left(\frac{a_i(t)}{n_i}\right)^2 (k_i(t) - u_i(t))^2$$

$$+ \frac{1}{\eta(1-2\eta)}\left(\frac{d}{4} + 2\sum_{t=1}^{T} \|\boldsymbol{u}(t+1) - \boldsymbol{u}(t)\|_1\right) \tag{133}$$

for any $\boldsymbol{u}(t) = (u_1(t), \ldots, u_d(t))^\top \in [0,1]^d$.

First, we prove $R_T \leq \sqrt{\sum_{i=1}^{d} n_i^2}\sqrt{\frac{\gamma}{\eta(1-2\eta)}(d + 8V_1)} + \mathcal{O}(W\gamma)$. From (38), letting $\boldsymbol{u}(t) = \boldsymbol{k}(t)$ in (133), we can bound the regret as

$$R_T \leq 2\gamma \sum_{i=1}^{d} n_i \mathbb{E}\left[\sqrt{\beta_i(1)^2 + \frac{1}{\gamma}\sum_{t=1}^{T}\sum_{t=1}^{T} \alpha_i(t)}\right] + \mathcal{O}(W\gamma)$$

$$\leq 2\mathbb{E}\left[\sqrt{\gamma\left(\sum_{i=1}^{d} n_i^2\right)\sum_{t=1}^{T}\sum_{i=1}^{d} \alpha_i(t)}\right] + \mathcal{O}(W\gamma)$$

$$\leq \frac{2\sqrt{\sum_{i=1}^{d} n_i^2}}{\sqrt{\eta(1-2\eta)}} \mathbb{E}\left[\sqrt{\gamma\left(\frac{d}{4} + 2\sum_{t=1}^{T-1} \|\boldsymbol{k}(t+1) - \boldsymbol{k}(t)\|_1\right)}\right] + \mathcal{O}(W\gamma)$$

$$\leq \sqrt{\sum_{i=1}^{d} n_i^2}\sqrt{\frac{\gamma}{\eta(1-2\eta)}(d + 8V_1)} + \mathcal{O}(W\gamma), \tag{134}$$

where the second inequality follows from the Cauchy-Schwartz inequality, the third inequality follows by setting $u_i(t) = k_i(t)$ for all $i \in [d]$ and $t \in [T]$ in (133), and the last inequality follows from Jensen's inequality. This becomes the desired path-length bound.

Next, we prove $R_T \leq \sqrt{\frac{1}{1-2\eta} \min\{L^*, MT - L^*, Q_2\}}$. For any $\boldsymbol{q}^* \in [0,1]^d$, letting $\boldsymbol{u}(t) = \boldsymbol{q}^*$ for all $t \in [T]$ in (133), we have

$$\sum_{t=1}^{T}\sum_{i=1}^{d} \alpha_i(t) = \frac{1}{1-2\eta} \sum_{t=1}^{T}\sum_{i=1}^{d}\left(\frac{a_i(t)}{n_i}\right)^2 (k_i(t) - m_i^*)^2 + \frac{d}{4\eta(1-\eta)}. \tag{135}$$

Using this inequality, we have

$$
\mathbb{E}\left[\sum_{t=1}^{T}\sum_{i=1}^{d}\alpha_i(t)\right] \leq \frac{1}{1-2\eta}\min_{\boldsymbol{q}^*\in[0,1]^d}\left\{\mathbb{E}\left[\sum_{t=1}^{T}\sum_{i=1}^{d}\left(\frac{a_i(t)}{n_i}\right)^2(l_i(t)-m_i^*)^2\right]\right\} + \frac{d}{4\eta(1-2\eta)}
$$

$$
\leq \frac{1}{1-2\eta}\min\left\{R_T+L^*, MT-L^*-R_T, Q_2\right\}, \tag{136}
$$

where in the last inequality, we set $\boldsymbol{q}^*=\boldsymbol{0}$ (resp. $\boldsymbol{q}^*=\boldsymbol{1}$) and use the same argument as that in Appendix C.3.4 for deriving the term with $R_T+L^*$ (resp $MT-L^*-R_T$), and $\boldsymbol{q}^*=\bar{\boldsymbol{\ell}}$ for deriving the term with $Q_2$, and this completes the proof.

