# OpenReview forum: "Multi-play Combinatorial Semi-Bandit Problem"
_ICLR.cc/2026/Conference — Submitted to ICLR 2026_

### Official Review · Reviewer_YQJV · 2025-10-25

**Soundness:** 2
**Presentation:** 2
**Contribution:** 2
**Rating:** 4
**Confidence:** 3

**Summary:**

The paper studies the multi-play combinatorial bandit problem, where at each round the learner selects a subset of arms (subject to combinatorial constraints) and receives a reward and observations from the selected arms with semi-bandit feedback. Under stationary and stochastic arm rewards, a variant of combinatorial Thompson sampling is shown to achieve $O(\log T)$ regret. When arm reward distributions are general, an algorithm that can adapt to attain best-of-both-worlds type guarantees is proposed. Its regret is bounded in the stochastic and adversarial regimes as well as a spectrum of regimes in between. Simulations are performed to compare the performances of the proposed algorithms with those from the literature.

**Strengths:**

This paper bridges the gap between multi-play and combinatorial bandits. It focuses on a setup that includes stochastic and adversarial reward regimes as special cases. For different regimes, it proposes different algorithms inspired by Thompson sampling and best of both worlds algorithms from the literature. It offers rigorous theoretical analysis with regret bounds. The problem is relevant to applications where multiple actions must be selected simultaneously under constraints.

**Weaknesses:**

1. The main results extend known techniques rather than introduce fundamentally new methods. The writing of the paper does not clearly convey the technical novelty of the derivations; as a result, the technical novelty seems incremental. Please elaborate in detail why the multi-play CMAB setting is significantly more challenging to analyze than the single-play CMAB setting.


2. Since this paper is highly theoretical in nature, the derivation of regret lower bounds for algorithms that do not use the duplication technique will greatly improve the contribution and will clearly demonstrate whether the derived upper bounds are optimal in some sense.

3. The presentation is difficult to follow as the notation is very dense. While this is partly due to page limits, it negatively impacts the broader accessibility of the paper to the ICLR community.

4. Simulations are limited in scope. No evaluation was performed in realistic settings where combinatorial multi-play naturally arises.

**Questions:**

1. The authors claim that the GenCTS algorithm does not require enumerating all possible actions at the beginning of the game. However, for the stochastic setting, isn't this a property of the oracle rather than the algorithm itself? Isn't the competitor CTS or an algorithm like CUCB also computationally feasible when the action set is exponentially large in d?

2. Is the compute complexity of the oracle included in the compute complexity of GenCTS? Isn't the competitor CTS also computationally feasible when the action set is exponentially large in d? The scalability of duplication CTS becomes an issue when the batch size is large.

3. Many works in CMAB resort to approximation oracles. Combinatorial Thompson sampling may not work well with such types of oracles (See, e.g., Wang, Siwei, and Wei Chen. "Thompson sampling for combinatorial semi-bandits." International Conference on Machine Learning. PMLR, 2018). Does this mean that the results in this paper cannot be extended to learning algorithms that use approximation oracles?

4. Multi-play setting is reminiscent of the batched-bandits setting (see, e.g., Gao, Zijun, et al. "Batched multi-armed bandits problem." Advances in Neural Information Processing Systems 32 (2019)). Please elaborate on how your work is connected to batched bandits.

---

> ### Author Response · Authors · 2025-11-20
>
> We would like to thank the reviewer for the valuable and constructive comments.
>
> > The authors claim that the GenCTS algorithm does not require enumerating all possible actions at the beginning of the game. However, for the stochastic setting, isn't this a property of the oracle rather than the algorithm itself? Isn't the competitor CTS or an algorithm like CUCB also computationally feasible when the action set is exponentially large in d?
>
> We agree that computational efficiency, without enumerating the action set, is a property inherited from the Oracle. Indeed, classical algorithms like CTS or CUCB are also computationally feasible when coupled with an efficient oracle. However, the core advantage of GenCTS is that it avoids the duplicating technique required to adapt classical $\\{0,1\\}^d$ settings to the $\mathbb{Z}^d$ integer settings.
>
> > Is the compute complexity of the oracle included in the compute complexity of GenCTS? Isn't the competitor CTS also computationally feasible when the action set is exponentially large in d? The scalability of duplication CTS becomes an issue when the batch size is large.
>
> Yes, the computation time of the oracle is included in the overall complexity of GenCTS, resulting in $\mathcal{O}(\mathrm{poly}(d)T)$ or $\mathcal{O}(\mathrm{pseudo\text{-}poly}(d)T)$．The scalability issue of the duplicating CTS is primarily due to the total number of duplicated base arms ($d'$), not necessarily the batch size. Figure 4 is also helpful in this regard.
>
> > Many works in CMAB resort to approximation oracles. Combinatorial Thompson sampling may not work well with such types of oracles (See, e.g., Wang, Siwei, and Wei Chen. "Thompson sampling for combinatorial semi-bandits." International Conference on Machine Learning. PMLR, 2018). Does this mean that the results in this paper cannot be extended to learning algorithms that use approximation oracles?
>
> Thank you for your great question. Extending our framework to the cases with  approximate oracles is the most important direction for our future work. We recognize that adapting GenCTS and GenLBINFV to the approximate oracle setting is a technically challenging problem due to the difficulty in maintaining the theoretical regret guarantees.
>
> > Multi-play setting is reminiscent of the batched-bandits setting (see, e.g., Gao, Zijun, et al. "Batched multi-armed bandits problem." Advances in Neural Information Processing Systems 32 (2019)). Please elaborate on how your work is connected to batched bandits.
>
> Thank you for this helpful comment. Our setting is different from the batched bandits in Gao et al. (2019). Batched bandits, in which the agent pulls a single arm per round, restrict adaptivity by revealing feedback only once per batch, while our setting allows full per-round adaptivity but requires selecting a combinatorial multi-play action. Also, in batched bandits, the learner receives feedback only $M$ times (for example, $M = O(\log \log T))$, and the main question is how to keep the minimax regret. Even though a combinatorial action contains many components, it is not a “batch” in the sense of limited adaptivity. Thus, the two frameworks address different constraints and are not directly comparable.

---

> > ### Comment · Reviewer_YQJV · 2025-11-23
> >
> > Thank you for your response. For now, I’ll keep my score as it is.

---

### Official Review · Reviewer_uaEi · 2025-10-31

**Soundness:** 2
**Presentation:** 3
**Contribution:** 2
**Rating:** 4
**Confidence:** 2

**Summary:**

This paper states the multi-play combinatorial semi-bandit (MP-CSB) problem, a generalization of the combinatorial semi-bandit (CSB) framework to multiple pulls per each arm on one round. This setting covers stochastic versions of optimal transport and knapsack problems, where binary decision CSB can hardly be applied. Two algorithms are proposed: GenCTS, a generalized Thompson sampling algorithm with O(log T) stochastic regret and GenLBINFV, and a best-of-both-worlds algorithm achieving variance-dependent logarithmic regret in the stochastic setting. Theoretical analysis guarantees improved regret bounds, and experiments confirm superior empirical performance.

**Strengths:**

1. A novel problem setting is addressed: MP-CSB extends CSB to non-binary, integer-valued actions, what extends MABs to broader scope of stochastic combinatorial optimization problems.
2. Algorithmic Efficiency: GenCTS avoids exponential enumeration of actions and produces in polynomial time
Regret bounds: both algorithms achieve state-of-the-art regret bounds

**Weaknesses:**

1. The practical motivation for the proposed MP-CSB setting is somewhat unclear. The authors propose two particular cases of the problem for practical inspiration: stochastic versions of the Optimal Transport problem and the knapsack problem. Unfortunately, they do not cite previous papers on the proposed cases, and their practical significance remains unclear.

2. The related work section seems incomplete. There are previous papers on non-binary combinatorial bandits, e.g., “Fixed-Budget Real-Valued Combinatorial Pure Exploration of Multi-Armed Bandit” by Nakamura et al. There are papers on stochastic versions of the Optimal Transport problem with bandits, e.g., “Bandit Optimal Transport” by Lorenzo Croissant. I expect a review with a comparison of the setting and algorithms to previous literature.

**Questions:**

See Weak points.

---

> ### Author Response · Authors · 2025-11-20
>
> We would like to thank the reviewer for the valuable and constructive comments.
>
> > The practical motivation for the proposed MP-CSB setting is somewhat unclear. The authors propose two particular cases of the problem for practical inspiration: stochastic versions of the Optimal Transport problem and the knapsack problem. Unfortunately, they do not cite previous papers on the proposed cases, and their practical significance remains unclear.
>
> We thank the reviewer for the feedback. We will revise the paper to clarify that the OT and Knapsack problems serve as canonical abstractions for broad classes of real-world problems involving integer flows and budgeted allocations, rather than narrow applications. We will also include suggested citations like [Croissant et al., 2025] (Please see also the answer to W2). Our work addresses online regret minimization where the action space is the subset of $\mathbb{Z}^d$. To our knowledge, this setting is unexplored, and standard binary ($\\{0,1\\}^d$) combinatorial  bandits cannot efficiently handle it due to action space explosion.
>
> > The related work section seems incomplete. There are previous papers on non-binary combinatorial bandits, e.g., “Fixed-Budget Real-Valued Combinatorial Pure Exploration of Multi-Armed Bandit” by Nakamura et al. There are papers on stochastic versions of the Optimal Transport problem with bandits, e.g., “Bandit Optimal Transport” by Lorenzo Croissant. I expect a review with a comparison of the setting and algorithms to previous literature.
>
> Thank you for your inputs on related papers and suggestions! They are relevant and we will certainly add these papers to enrich our Related Work sections. We would like to note that there are still distinctions and we will add the following discussion as comparison.
> (Croissant 2025)  studies the Kantorovich formulation of the OT problem as an extension of stochastic linear bandits on Hilbert spaces, while our offline OT problem is based on a discrete assignment problem. The main difference is that our work also addresses the adversarial setting, in addition to the stochastic setting, and provides best-of-both-worlds regret guarantees.
> (Nakamura et al., 2024) focus on Best Arm Identification (BAI) (pure exploration), where the goal is to find the optimal arm regardless of the cost incurred during learning. Our goal is regret minimization, and BAI algorithms generally incur linear regret and are not applicable to our setting. Also, their work operates on a continuous real-valued space ($\mathbb{R}^d$), while we address strict integer constraints ($\mathbb{Z}^d$).More importantly, our work provides Best-of-Both-Worlds guarantees (handling adversarial/corrupted regimes), whereas Nakamura et al. is limited to the stochastic setting.
>
>
> [R1] Lorenzo Croissant. Bandit Optimal Transport. 2025.
>
> [R2] Nakamura et al., 2024,  Fixed-Budget Real-Valued Combinatorial Pure Exploration of Multi-Armed Bandit

---

### Official Review · Reviewer_FyN2 · 2025-11-06

**Soundness:** 3
**Presentation:** 2
**Contribution:** 3
**Rating:** 6
**Confidence:** 3

**Summary:**

This work considers an extension of the well-studied combinatorial semi-bandits (CSB) problem, where at each round the learner selects a subset of base arms a loss that is a (Lipschitz) function of the losses of the individual selected arms, while they observe all of the latter losses.
The extension considered here allows the learner to select a single arm multiple times in the same round, hence seeing the choice at any round as a vector with non-negative integer values instead of a binary vector.
While this problem can be already solved by relying on existing approaches for linear bandits or CSB, the former approach would typically require to enumerate all combinatorial arms (which is generally impractical) while the latter requires to duplicate base arms and thus leading to possibly suboptimal performance and guarantees.
The authors show both a Thompson-sampling-based algorithm that achieves a variance-dependent logarithmic regret bound in the stochastic case where the loss can be any Lipschitz function of the losses of base arms, and an INF-based algorithm with log-barrier regularization and data-dependent learning rates that provides best-of-both-worlds regret guarantees in the case of linear losses.
Experiments on synthetic data further show the improvements achieved by the proposed algorithms for this generalized CSB problem, compared to the performance of adaptation of existing algorithms for the vanilla CSB problem via base arm duplication.

**Strengths:**

The authors investigate a natural and interesting extension of the CSB problem, and clearly motivate their proposed techniques by observing crucial downsides coming from straightforward extensions of algorithms from prior work.
The generalized CSB problem considered in this submission is also well motivated by fundamental combinatorial optimization problems such as knapsack and optimal transport.
The proposed techniques work under assumptions with various degrees of generality, with the Thompson sampling algorithm in the stochastic setting with Lipschitz losses, as well as an INF-based algorithm that guarantees near-optimal regret even in adversarial environments.

**Weaknesses:**

While the extensions of the algorithms lead to advantages compared to previous approaches, as underlined by the authors, the adaptation of existing techniques appears relatively straightforward.
Nonetheless, the authors show that carefully tuning the various steps of such algorithms (e.g., the extension of the log-barrier regularizer and the time-varying learning rate) and analyzing them carefully remain nontrivial tasks, at which they succeeded.

In the introduction, the description of the problem setting itself as an generalization of CSB in simple words is somewhat unclear and could benefit from a clearer presentation.
In particular, the authors mention decisions “taking non-negative integer values” without clearly stating in words what these integers represent.
One may intuitively deduce that these represent the multiplicity in the choice of each base arm, but this does not seem to be stated so explicitly, and a reader only truly understands this fact when reading the formal definition of the problem setting (which may be heavy to parse given the many objects that require a definition).

As a side note, the definition of the stochastic regime with adversarial corruptions at lines 168-171 seems slightly different from the more established one in Ito (2021).
In particular, the expectation in your definition is external, while Ito (2021) defines the corruption level $C$ by looking at mean losses of the stochastic ones and the corrupted ones (instead of the realized ones).
It would then seem that the $C$ you consider would need to be generally larger, and thus guarantees depending on your $C$ would be worse than depending on the original definition of the corruption level.
Anyhow, this is not a major weakness but it does require some comment in my opinion.

Another side note is about the values of $b\_i$’s from the duplication approach.
It is never specified, and thus left unclear, what values these parameters should be set to.
However, at first glance it would seem that setting $b\_i=n\_i$ would work as intended.
The authors, unfortunately, never comment on that and they also leave this point unaddressed (e.g., see line 424).
This should really be looked into in order to perform a fair comparison with existing approaches adapted by duplicating base arms, especially because previous algorithms would be at a clear advantage if it suffices to set $b\_i = n\_i$.

Finally, one of the main advantages of the proposed algorithms is runtime (and space) efficiency, but experiments never compare the runtime of the proposed algorithms to the baselines.
It then feels like experiments are not really covering all relevant aspects that should be empirically verified, leaving the reader with a sense of incompleteness.

**Questions:**

1. What you describe at lines 161-164 is an adaptive (non-oblivious) adversary, right? If so, it would be better to clarify this aspect since it might not be straightforward to work with non-oblivious adversaries in partial feedback settings.
2. Could you comment on the different definition of the stochastic regime with adversarial corruption compared to the one from Ito (2021)? See my comment in **Weaknesses**.
3. The bands around the curves in the plots from the experiments section have semi-width equal to the standard deviation over the 100 trials, or don’t they? Please clarify this in the text.

**Minor comments and typos:**
- Line 137: the last $\\mathbb{R}$ is more precisely $[0,1]$ given line 133
- Line 155-156: “independent” random variables?
- Please, recall the definition of $N\_i(t-1) = \sum\_{s=1}^{t-1} a\_i(s)$ around eq. (3)

---

> ### Author Response · Authors · 2025-11-20
>
> We would like to thank the reviewer for the valuable and constructive comments.
>
> > What you describe at lines 161-164 is an adaptive (non-oblivious) adversary, right? If so, it would be better to clarify this aspect since it might not be straightforward to work with non-oblivious adversaries in partial feedback settings.
>
> Yes, the description at lines 161–164 indeed corresponds to an adaptive (non-oblivious) adversary. In the adversarial regime we assume that, at each round t, the loss $L_{i,j}(t)$ may depend on the entire past history ${(L^1(s), ..., L^d(s), a(s))}_{s < t}$, as stated in Section 2.
> To avoid confusion, we will add an explicit clarification in Section 2:
>
> > Could you comment on the different definition of the stochastic regime with adversarial corruption compared to the one from Ito (2021)?
>
> Our corruption level C is, in the sense that it is defined as the sum over all rounds of the largest per-arm difference between the losses before and after corruption in each round, the same as in Ito (2021). Because our definition takes an expectation outside this quantity, it may be somewhat more conservative than that of Ito (2021).
>
> > The bands around the curves in the plots from the experiments section have semi-width equal to the standard deviation over the 100 trials, or don’t they? Please clarify this in the text.
>
> In Figures 1 and 2, the solid lines represent the average regret over 100 independent runs, and the shaded bands around each curve correspond to ± one empirical standard deviation across these 100 runs.
> We will clarify this explicitly in Section 5 as follows:
> “In Figures 1 and 2, the solid curves show the average regret over 100 independent trials, and the shaded regions indicate ± one standard deviation across these trials.”

---

> > ### Comment · Reviewer_FyN2 · 2025-11-23
> >
> > Thank you for your response. Could you also address my comment on the values of $b\_i$ and how they relate to those of $n\_i$, as well as my remark on the computational efficiency of your algorithms compared to existing approaches?

---

> > > ### Author Response · Authors · 2025-11-29
> > > **Comments on the computational efficiency of our algorithms compared to existing approaches**
> > >
> > > We appreciate the reviewer’s comment that runtime (and space) efficiency is one of the main advertised advantages of our algorithms, and that this aspect is not directly evaluated in the experimental section.
> > >
> > > The experiments in the current version are conducted with a relatively small dimension $d$. The reason is that, for CTS with arm duplication, the dimension becomes so large that the oracle for solving the offline optimal transport problem could not finish within a reasonable amount of time. Similarly, for LBINFV, applying arm duplication makes the size of the action set very large, and the algorithm again does not terminate within a practical computation time. In this way, both CTS and LBINFV become computationally impractical when the number of arms increases due to duplication.
> > > By contrast, our algorithms do not require arm duplication, so the dimension $d$ (i.e., the number of original arms) remains unchanged, and there is still room to further increase $d$. In fact, although we did not include these results in the current version, we have confirmed that for larger $d$ our algorithms can run in much shorter time than existing algorithms that rely on the arm-duplication technique.
> > > In the revised version, we will describe these points and also report the actual running times.

---

> ### Author Response · Authors · 2025-11-29
> **Comments on $b_{i}$ and $n_{i}$**
>
> We thank the reviewer for pointing out that we did not explicitly spell out how the duplication parameter $\\{ b_{i} \\}_{i = 1, \ldots, d}$ is instantiated when adapting existing CSB algorithms to the MP-CSB setting.
>
> $b\_{i}$ is the number of duplicates required to construct the binary action set. For example, in the optimal transport problem, the number of arms from a supplier $x$ to the demand nodes is $u_{x}$. This is because supplier $x$ has $u\_{x}$ trucks, and we need to duplicate the arm as many times as the number of trucks. On the other hand, $n\_{i} = \min\\{ u\_{x}, v\_{y} \\}$, since sending more than $v\_{y}$ trucks would violate the constraints of the optimal transport problem. This suggests that we always have $b\_{i} \ge n_{i}$. A similar argument can be made for the maximum flow problem. For the knapsack problem, we actually have $b\_{i} = n\_{i}$. We will explain these points to make it clear that the regret upper bounds of our algorithms are not necessarily worse than those of existing algorithms based on the arm-duplication technique.

---

### Official Review · Reviewer_tugJ · 2025-11-06

**Soundness:** 2
**Presentation:** 1
**Contribution:** 1
**Rating:** 2
**Confidence:** 3

**Summary:**

The authors consider a variant of the combinatorial semi-bandits problem where each arm can be played multiple-times in a round.  The authors propose a Thompson sampling based method for the stochastic setting as well as a best-of-both-worlds method, obtaining regret for stochastic regret for the former and adversarial (as well as stochastic rewards with adversarial corruptions) for the latter.  The authors also run experiments, showing their work outperforms combinatorial semi-bandit method baselines where arm duplications are used.

**Strengths:**

- The authors consider a variant of the combinatorial semi-bandit problem with multiple-arm pulls (multiple base arms can each be pulled multiple times in a single round)
- The authors propose a TS based algorithm (building on CTS from Wang and Chen 2018) for the multi-play CSB with stochastic environments and analyze regret and complexities (computational and storage)
- the authors propose a best of both worlds algorithm (building on LBINFV from Tsuchiya et al 2023) for multi-play CSB analyzing regret bounds for adversarial, stochastic, and stochastic with corruptions.
- the authors include experiments, showing simple duplication and use of prior CSB methods is inefficient

**Weaknesses:**

### Major
- There is no discussion of technical challenges in the main paper in extending CSB methods to multi-play settings.  There may be some significant technical hurdles in the regret analyses, which could merit significance to the contribution, but they are not made clear in the main paper.
    - The authors do contrast regret bounds and complexities against the trivial approach of duplicating arms and blindly applying CSB methods.  It is obvious that such an approach is wasteful (all the losses from playing arm $i$ should be used together).  There may be some significant technical hurdles in the regret analyses, which could merit significance to the contribution, but they are not made clear in the main paper.
    - line 317 mentions generalizing LBINFV “is not straightforward.  We show that, by carefully leveraging $n_i$’s, LBINFV can be generalized to MP-CSB.” And terms like the regularizer (5) and parameters (6) include dependence on $n_i$, but from my reading I did not see a clear articulation of what was not straightforward.
- There is no discussion of, or even brief references to, past works on variants of “multi-play” bandits.  In terms of method design, analysis, and extent to which the changes from regular bandits is replicated here (in the change from CSB) vs extending CSB to multi-play has unique technical challenges?  from a quick search, I came across a few that sound relevant
    - “Multiple-Play Stochastic Bandits with Shareable Finite-Capacity Arms” ICML 2022 this sounds fairly close to the setup considered in the paper, where each arm can be played multiple times (up to a capacity limit).  This ICML paper, interestingly, considers unknown capacities and an aggregate semi-bandit feedback for each arm (as opposed to a per-play feedback).     That presents challenges not present in the current work, as the capacity must be estimated.

    - Komiyama et al. (2015) “Optimal regret analysis of Thompson sampling in stochastic multi-armed bandit problem with multiple plays” in ICML 2015 for multi-play MAB.  In this setting arms are not repeatable in a round).  This may have been later generalized by CTS  (there are also later multi-play TS methods like “Multiple-Plays Multi-armed Bandit with Accelerated Thompson Sampling” https://ieeexplore.ieee.org/document/10695179 )
    - There is also a large literature on multi-agent MAB (which is a generalization of multi-play MAB as the same arm can be played multiple times in the same round), including some with combinatorial action spaces (https://arxiv.org/abs/2405.05950 is one example came across, there are probably more that also consider combinatorial actions).   Works of this line would probably not generalize the problem the authors consider (e.g. the number of agents might need to match the per-round arm-play capacities $\{n_i\}$ for that to be the case), but there may still be relevant relations in terms of algorithm design and analysis that may merit discussion.
    - More broadly speaking, there is a lack of references to recent works. From quick search of the pdf, there are no citations to works published or pre-prints posted in 2024 or 2025.  This brings concern about how well characterized the state of the related literature is.  CTS is from 2018, and 2023 is the latest of the CSB references.

### Minor
- Table 1 references CTS and LBINFV but reports the regret bounds and complexities not of those works (designed for the single-play setting) but what they would be if the learner naively duplicated base arms and then applied those methods (oblivious to the duplication).  That is misleading to the reader.   It is fine to include those formulas as a simple baseline, but that should be in the text, not the table.  Instead, mark the sets of rows as being related but different problems and report CTS and LBINFV for the regret and complexity for the single-play settings they were designed for.
    - (minor) if the past works mentioned are not limited to linear, then that should be noted.
    - (minor) If there are any CSB works since 2018 that outperform CTS, they should be included (eg the table for CSB should not be limited to the two that are extended in this work)
    - (minor) I do not think linear bandit methods would be essential to include in the table
- I think using space in lines 256-311 to highlight improvements over the trivial duplication technique is not helpful (can keep it, but put it in the appendix), as duplication is obviously inefficient statistically and computationally.  Instead, it would have been much better that the authors discuss technical challenges (if any) in the regret analysis and high level descriptions of they overcame those challenges.

- For the motivating real-world applications, the descriptions need to be revised
    - line 198, for OT, a real-world application what is the “cost of edges it passes through”?
    - The knapsack problem as described is not a multi-play problem.  There are simply $d$ items.
- joint reward function non-linearity (for the stochastic setting)
    - Line 158 when does this assumption hold, other than linear?  If you want to claim your results hold for non-linear settings, you need to discuss what non-linear classes your results apply to.
    - the OT and knapsack examples have linear objectives (the authors later explicitly acknowledge that); the ad example does not explicitly state the objective function.
- Figures 1 and 2 should have labels on the axes.  The main text does not clarify exactly what is plotted (what either axis is), just referring to figures as showing “the results”

### Very Minor
- line 307-311 memory discussion looks to be specializing to OT problem without mentioning that (I was confused by notation there at first)
- line 238 “output[s]”
- There are capitalization errors in the reference list, Kveton et al 2015 ref, Schrijver, combettes et al

**Questions:**

Several questions are included along with comments in the "Weaknesses" section above.

---

> ### Author Response · Authors · 2025-11-20
> **Answer to the first major concern**
>
> We would like to thank the reviewer for the valuable and constructive comments.
>
> Response to Comment 1 (technical challenges in extending CSB / LBINFV)
>
> We appreciate the reviewer’s detailed reading and for pointing out that the technical challenges in extending CSB methods to the multi-play setting were not sufficiently highlighted in the main paper.
> Our main technical difficulty in generalizing LBINFV to MP-CSB is that both the action space and the feedback structure change in a way that breaks the original analysis:
> In ordinary CSB, each coordinate of the decision vector satisfies $x_i \in [0,1]$, and if arm $i$ is selected, it is pulled at most once per round.
>
> In MP-CSB, actions are vectors $a \in \mathbb{Z}\_{\geq 0}^d$, and each arm $i$ can be pulled up to $n\_i$ times per round. The algorithm must construct a fractional point $x(t) \in \mathrm{conv}(\mathcal{A})$, whose coordinates satisfy $0 \le x_i(t) \le n_i$,
> and for which we may observe up to $a_i(t)$ individual losses per round. Because of this, simply plugging in the original LBINFV regularizer and proof does not go through.
>
> Concretely:
>
> **1. New regularizer that respects capacities $n_i$**.
>
> The regularizer in GenLBINFV is $\psi\_t(x) = \sum\_{i=1}^d \beta_i(t),\phi_i(x_i)$, where $\phi_i(z)= n_i \Bigg( \frac{z}{n_i} - 1 -\log\frac{z}{n_i} + \log T\cdot\frac{z}{n_i} \bigg(1-\frac{z}{n_i}\bigg)\log\Big(1-\frac{z}{n_i}\Big) \Bigg)$. This is not just a cosmetic scaling of the LBINFV regularizer. We had to redesign it so that:
> - Its effective domain matches $x\_i \in [0,n\_i]$ rather than $[0,1]$,
> - The Fenchel–dual / OFTRL analysis produces regret terms that scale with $\lambda_{\mathcal{A}}$ and $\sigma_i^2$, instead of incurring an extra factor $\sum\_i n\_i$ (as would happen if we naively replicated arms).
>
> In particular, the scaling by $n_i$ and the appearance of $z/n_i$ inside the logarithms are precisely what enables the analysis to treat the multi-play case without paying for each duplicate arm separately.
>
> **2. Variance-aware regularization parameters depending on the multi-play structure.**
>
> The regularization parameter is defined as
> $\beta_i(t) = \sqrt{ \Bigl(1 + \frac{\epsilon_i}{n_i}\Bigr)^2 \frac{1}{\log T}
> \sum_{s=1}^{t-1} \Bigl(\frac{a_i(s)}{n_i}\Bigr)^2
> (k_i(s)-q_i(s))^2 \min\Bigl(1,, 2\Bigl(1-\frac{x_i(s)}{n_i}\Bigr)
> \Bigl(\frac{x_i(s)}{n_i}\Bigr)^2 \log T\Bigr)}$.
>
> This differs substantially from LBINFV. They are tailored to:
> - The fact that we have up to $a\_i(s)$ observations per round for each arm $i$, and
> - The specific unbiased estimator $\hat\ell\_i(t) = q\_i(t) + \frac{a\_i(t)}{x\_i(t)}\bigl(k\_i(t) - q\_i(t)\bigr)$, whose variance now depends on both the fraction $x\_i(t)/n\_i$ and the multiplicity $a\_i(t)$.
>
> The proof has to be reworked so that the local norms induced by $\psi_t$ still control the variance of this new estimator. In particular, our analysis introduces concentration bounds that handle the factor $(a\_i(t)/x\_i(t))^2$, which is absent in the original LBINFV setting.
>
> **3. New action-set dependent constant $\lambda\_{\mathcal{A}}$ and regret scaling.**
>
> The introduction of the constant $\lambda\_{\mathcal{A}} = \min \{ M, W\_{J^\ast} \}$, where
> $M = \max_{a\in\mathcal{A}} ||a||\_{1}, \quad W\_{J^\ast} = \sum_{i\in J^\ast} n_i$, and its role in our main theorem is a further technical difference from LBINFV.  Especially, the structure of the optimal action through $J^\ast$ is important. This is crucial to obtain the variance-dependent $O(\log T)$ bound of the form $O\Bigl(\sum_{i\in J^\ast} \lambda_{\mathcal{A}} \frac{\sigma_i^2}{\Delta_i}\log T\Bigr)$ without an extra $\sum_i n_i$ factor that would appear if we simply duplicated arms and applied LBINFV as a black box.
>
> We agree that these challenges are currently only implicit in the main text. In the revision, we will add a dedicated paragraph (at the beginning of Section 4) explaining:
> - why the LBINFV proof does not directly extend to MP-CSB,
> - how the rescaled hybrid regularizer $\phi_i$ and capacity-aware parameters $\beta_i(t)$ are designed to fix this, and
> - how this leads to the improved regret dependence compared to duplicating arms.
>
> We will also clearly cross-reference the key lemmas in the appendix that formalize these steps.

---

> ### Author Response · Authors · 2025-11-21
> **Answer to the second major concern (part 1)**
>
> Response to Comment 2 (relation to multi-play and multi-agent bandits, and recent work)
>
> We thank the reviewer for the helpful pointers and for highlighting the need for a broader discussion of multi-play bandits and recent literature. We address each sub-point and will incorporate these works in the related-work section.
>
> **1. Multiple-Play Stochastic Bandits with Shareable Finite-Capacity Arms (X. Wang et al., ICML2022)**
>
> Our MP-CSB model differs in several important respects:
>
> 1.1 General combinatorial constraints
>
> MP-CSB allows actions $a \in \mathbb{Z}_{\ge 0}^d$ constrained by a combinatorial structure (e.g., flows on a bipartite graph for OT, knapsack constraints). In contrast, the ICML 2022 paper focuses on a simpler multiple-play setting without such general combinatorial constraints.
> 1.2 Semi-bandit feedback
>
> In the ICML 2022 setting, the learner only sees a single scalar reward per arm that depends on both the unknown capacity and the number of plays. This drives their main technical challenge: estimating capacities from aggregate load-dependent rewards.
>
> 1.3  Best-of-both-worlds vs. purely stochastic guarantees.
>
> The ICML 2022 paper is set in the purely stochastic regime and its regret analysis is tailored to that setting. In contrast, one of the main strengths of our work is that we design GenLBINFV to achieve best-of-both-worlds (BOBW) guarantees: the same algorithm enjoys variance-dependent logarithmic regret in stochastic MP-CSB instances while also retaining robust guarantees in adversarial (and adversarially corrupted) settings. To the best of our knowledge, such BOBW guarantees have not been studied in the shareable finite-capacity multi-play bandit framework of Wang et al., and they are crucial in our positioning of MP-CSB as a unifying model for both stochastic and adversarial environments.
>
> We will add a paragraph clarifying that the ICML 2022 setting can be seen as a special case where the action is a “load vector” without general combinatorial structure and only aggregate feedback is observed, whereas MP-CSB targets combinatorial integer-flow problems with rich semi-bandit feedback. Extending MP-CSB to unknown capacities, in the spirit of that work, is an interesting and orthogonal direction that we will mention as future work.
>
> **2. Multi-play MAB (Komiyama et al., ICML2015)**
>
> Komiyama et al. (ICML 2015) study multiple-play MAB with binary actions $a \in \\{0,1\\}^d$ such that $\sum_i a\_i = m$ and at most one pull per arm per round. More recent works on “multiple-plays multi-armed bandit with accelerated Thompson Sampling” build on this setting with improved TS algorithms.
> MP-CSB can be seen as a strict generalization of these models:
> If we restrict the action set to $\\{0,1\\}^d$ with $\sum\_i a\_i = m$ and set $n_i = 1$, then MP-CSB reduces to the classical multiple-play MAB studied by Komiyama et al.
>
> Our algorithms (GenCTS / GenLBINFV) work in much richer domains where:
> - actions are arbitrary non-negative integer vectors satisfying combinatorial constraints (e.g., transportation polytopes, knapsack),
> - each arm can be used multiple times per round (up to n_i), and
> - we consider not only stochastic but also adversarial and corrupted regimes.
>
> We will make this relation explicit by:
> - adding citations to Komiyama et al. (2015) and recent accelerated TS variants, and
> - adding a sentence in Section 2 that classical multi-play MAB is recovered as a special case of MP-CSB with binary actions and $n_i = 1$.
>
> **3. Multi-agent MAB with combinatorial actions**
>
> Recent work on multi-agent MAB (including combinatorial variants) studies settings where:
> - there are multiple agents, each selecting a set of arms,
> - feedback is often full-bandit at the subset level (i.e., only the aggregate reward is observed), and
> - there may be communication or synchronization constraints across agents.
>
> Our MP-CSB framework is different but related:
> - We consider a single learner that selects a combinatorial multi-play action and observes semi-bandit feedback, potentially with multiple samples per arm.
> - The combinatorial coupling is expressed through constraints on $a \in \mathbb{Z}\_{\ge 0}^d$, rather than through coordination among multiple agents.
> - Our BOBW analysis (GenLBINFV) crucially exploits the per-arm multi-feedback structure and the capacities $n_i$, which do not appear in the same way in federated or decentralized multi-agent MAB.
>
> We agree that drawing this conceptual connection can help position our contribution in the broader literature, and we will add a short paragraph in the related-work section explaining the differences and commonalities along two axes: (i) combinatorial structure of actions vs. number of agents, and (ii) semi-bandit vs. aggregate feedback.

---

> ### Author Response · Authors · 2025-11-21
> **Answer to the second major concern (part 2)**
>
> **4 Lack of recent references**
>
> We appreciate this concern. After receiving the review, we conducted an additional literature search focusing on 2024–2025 works on combinatorial bandits, multi-play bandits, and BOBW algorithms. To the best of our knowledge, we did not find prior work that simultaneously:
> - allows multiple observations per arm per round with fixed capacities $n_i$,
> - works with general combinatorial action sets over non-negative integer flows (e.g., OT and knapsack),
> - provides both stochastic, adversarial, and adversarial-corruption guarantees in a single BOBW algorithm, and
> - avoids the extra $\sum_i n_i$ factor that arises from naïve arm duplication in both regret and memory complexity.
>
> We will nevertheless update the related-work section to include:
> -  the X. Wang et al. (ICML2022) work on shareable finite-capacity arms
> - Komiyama et al. (2015) and recent multi-play TS variants
> - recent multi-agent and combinatorial bandit papers, and
> - additional 2024–2025 combinatorial/BOBW references where appropriate.
>
> We will also explicitly state how MP-CSB is positioned with respect to two main axes:
> - action space: from simple multiple-play MAB to general combinatorial integer flows,
> - feedback: from full-bandit or aggregate rewards to semi-bandit with multiple observations per arm.
> - robustness across environments: rom purely stochastic guarantees to best-of-both-worlds guarantees that remain valid under adversarial or adversarially corrupted losses
>
> We will also clarify that our contribution is not only in the regret guarantees, but also in the computational and memory complexity: by working directly on MP-CSB rather than duplicating arms, our algorithms avoid an extra $sum\_i n\_i$ factor both in regret and in resource usage, while still providing best-of-both-worlds guarantees.
>
> We hope that these clarifications and additions address the reviewer’s concerns about the technical novelty and the coverage of related work.

---

### Official Review · Reviewer_N7EU · 2025-11-12

**Soundness:** 3
**Presentation:** 2
**Contribution:** 2
**Rating:** 4
**Confidence:** 4

**Summary:**

The paper studied the regret minimization problem for combinatorial bandits where each arm is allowed to be pulled for more than once during each time step. It proposed and analyzed two algorithms for stochastic and adversarial settings respectively. It also conducted numerical results to indicate the superiority of proposed algorithms.

**Strengths:**

1. The author(s) compared the proposed algorithms and existing ones with Table 1.
2. The applications are discussed.

**Weaknesses:**

1. Beginning from LINE 185, some applications of MP-CSB are discussed, where there is a contraint $\mathbf{a}\in \ldots$ for the optimal transport problem and another one for the knapsack problem. However, I failed to find a corresponding constraint in the problem formulation and hence feel a bit confused about the relation between the application scenarios and the studied formulation.
My major concerns are as below:
1. A lower bound is discussed around LINE 176. However, the two works cited there studied the combinatorial bandits where one arm can be pulled at most once at each time step. I wonder how their results imply the lower bound here? Besides, if there is one, the lower bound should be included in Table 1.
1. The GenLBINFV algorithm is claimed to be best-of-both-world. However, there is no discussion on its comparison to a lower bound. How can the algorithm be claimed to be BOBW?
1. The pseudocode of GenLBINFV algorithm should be included in the main part of the manuscript as it is quite important.
4. More discussions on numerical results are appreciated.
1. This paper \url{https://proceedings.mlr.press/v51/jun16.html} also allowed multiple pulls of one single arm but focused on the pure exploration problem. Such works should be mentioned as related works.

Minor issue:
1. Table 1: The names of existing algorithms and the corresponding references should be included altogether.

**Questions:**

Please refer to the **Weaknesses** section above.

---

> ### Author Response · Authors · 2025-11-20
>
> We thank the reviewer for the insightful comments and questions.
>
> > A lower bound is discussed around LINE 176. However, the two works cited there studied the combinatorial bandits where one arm can be pulled at most once at each time step. I wonder how their results imply the lower bound here? Besides, if there is one, the lower bound should be included in Table 1.
>
> According to previous work, the lower bound for the standard CSB setting is $O(\frac{dm \log T}{\Delta})$, where $d$ and $m$ are the number of arms and the maximum number of arms one can pull in a round, respectively. If we apply the existing CSB algorithm to MP-CSB, the number of arms and the maximum number of arms one can pull in a round are $\sum_{i=1}^{d} b_{i}$ and $M$, respectively, which implies the lower bound in line 175.
> We leave the derivation of a lower bound for algorithms that can pull a single arm multiple times in one round without duplicating arms as future work.
>
> > The GenLBINFV algorithm is claimed to be best-of-both-world. However, there is no discussion on its comparison to a lower bound. How can the algorithm be claimed to be BOBW?
>
> By duplicating the arms and applying the existing CSB algorithm to the MP-CSB setting, the regret lower bound is $O(\log⁡T)$ in the stochastic regime, and $\tilde{\mathcal{O}}(\sqrt{T})$ in the worst case for the adversarial regime. Since we expect that the lower bound for algorithms that can pull an arm multiple times in a round has the same order (up to constant factors), we refer to the GenLBINFV algorithm as a best-of-both-worlds (BOBW) algorithm. We will add a supplementary explanation of this point in the revised version of the paper.

---

> > ### Comment · Reviewer_N7EU · 2025-11-23
> >
> > Thanks for your reponse to some of my questions. I would like to keep the rating.

---

### Meta-Review · Area_Chair_4REG · 2026-01-06

**Summary:**

This is a paper that falls squarely in the reject region. One of the main criticisms of the reviewers is missing literature and the technical novelty, which appeared minor to the reviewers. The rebuttal tries to address this but reading through it I also have a hard time seeing the challenging and interesting novel technical part.

**Reviewer Concerns:**

The main concerns are that the reviewers want more comparison to existing literature and that the technical novelty appears minor.

**Reviewer Scores:**

I don't think that the reviewers would have changed their score after the rebuttal.

---

### Decision · Program_Chairs · 2026-01-26

Reject